# Pericytes are organ-specific regulators of tissue morphogenesis

Seyed Javad Rasouli [1], Kai Kruse [2], Rodrigo Diéguez-Hurtado [1], Parisa Ghanbari[3], Anusha Aravamudhan [1], Mara E. Pitulescu [1,4] & Ralf H. Adams [1] ✉

Endothelial cells lining the vessel network are indispensable for vascular transport but also provide paracrine signals controlling the behavior of nearby cell types. Pericytes are another essential component of the vessel wall, but little is known about their interactions with other cell populations during organ growth and patterning. Here, we use mouse genetics to address the function of three pericyte-derived factors in postnatal lung and brain. We find that inactivation of the gene for hepatocyte growth factor (HGF) or brain-derived neurotrophic factor (BDNF) in pericytes causes no overt alterations in postnatal brain but impairs lung development, which we attribute to defective interaction with AT2 epithelial cells and pulmonary endothelium, respectively. In contrast, pericyte expression of the growth factor Nodal is dispensable for lung morphogenesis but regulates vessel growth and barrier function in the postnatal brain through interactions with endothelial cells, astrocytes and microglia. Taken together, our findings establish that pericytes are a critical source of paracrine signals controlling morphogenetic processes in an organ-specific fashion.

Blood vessels form a highly elaborate, hierarchical network that reaches, with a few exceptions, into all parts of the vertebrate body. In addition to the essential transport function of the vasculature, endothelial cells provide critical molecular signals acting on other cell populations in their vicinity. This paracrine (also termed 'angiocrine') function of endothelial cells regulates morphogenesis, homeostasis and regeneration of different organs, including lung, liver, heart, and bone[1,2]. Endothelial cells (ECs) are also a critical component of stem cell niches and thereby play key roles in hematopoiesis, bone formation, and neurogenesis[3,4]. Thus, ECs have emerged as important signaling centers that acquire organ-specific specialization, coordinate regeneration and help to prevent deregulated, disease-promoting processes. Even though the range of EC-derived signals has been explored only partially, it is evident that the angiocrine function of capillary EC is reflected by specialized expression of certain genes, distinguishing different organs but also local environments within the same organ[5–8].

Pericytes, capillary-associated supporting cells, are another essential component of the vessel wall and help to preserve vascular integrity[9,10]. Pericytes, but also the other mural cell population, vascular smooth muscle cells (vSMCs), are fairly heterogeneous with regard to their morphology, but also developmental origin. Pericytes and vSMCs in the developing heart, for example, can either arise from epicardial mesothelial cells or, as we have shown, via endothelial-to-mesenchymal transition from the embryonic endocardium[11,12]. Neural crest cells give rise to pericytes in the retina, brain, thymus, and the head region[13–16]. In the gut, lung and liver, the mesothelium, a single-layer squamous epithelium, is a source of mural cells[17,18]. Distinct developmental origins and tissue environments may also explain the expression of specific molecular markers by pericytes in different organs, including a range of secreted molecules, as was uncovered by single-cell RNA-sequencing (scRNA-seq) and other approaches[19,20]. Likewise, scRNA-seq has revealed substantial organ-specific gene

[1]Department of Tissue Morphogenesis, Max Planck Institute for Molecular Biomedicine, Münster, Germany. [2]Max Planck Institute for Molecular Biomedicine, Bioinformatics Service Unit, Münster, Germany. [3]Department of Vascular Cell Biology, Max Planck Institute for Molecular Biomedicine, Münster, Germany. [4]Vascular Patterning Dynamics Group, Max Planck Institute for Molecular Biomedicine, Münster, Germany. ✉e-mail: ralf.adams@mpi-muenster.mpg.de

expression profiles of pericytes and other vessel-associated cells in adult brain and lung[21]. Our own previous work has established that pulmonary pericytes control the proliferation of epithelial cells and are thereby indispensable for alveologenesis during postnatal lung development[22].

Stimulated by the findings above, we have utilized scRNA-seq in combination with mouse genetics and cell culture approaches to explore whether pericytes act as organ-specific regulators of tissue morphogenesis during postnatal development. We chose lung and brain as representative model organs and explored the function of several pericyte-derived, potentially angiocrine-acting factors, namely brain-derived neurotrophic factor (BDNF), hepatocyte growth factor (HGF) and Nodal, a TGFβ family protein. Our findings establish that pericytes are indeed functionally specialized in those two organs and regulate the behavior of other cell types in their vicinity through the secretion of angiocrine signals.

## Results

### In vivo characterization of lung and brain pericytes

To investigate pericyte morphology in lung and brain, we employed a genetic labeling strategy involving *Pdgfrb-CreERT2* transgenic mice[12,23] in combination with the *R26-mT/mG* Cre reporter[24]. While vascularization of the brain is initiated around midgestation in the mouse embryo, this process continues during postnatal stages in order to expand and remodel the brain vasculature[25]. Similarly, growth of the murine pulmonary vasculature begins early after midgestation but continues during several weeks of postnatal life, accompanying major steps of lung septation and alveolarization, which results in a peak of alveolar density at postnatal day (P) 39[26–28]. Given the comparably broad expression of PDGFRβ in different embryonic cell populations[29], we selected the time window from postnatal day P0 up to P12 for the brain and up to P21 for the lung for our analyses. Previous work had already established that *Pdgfrb-CreERT2* allows efficient fate tracking and genetic manipulations in postnatal mural cells in a range of organs, including brain and lung[12,22,30,31], whereas limited recombination is observed in cells with low PDGFRβ expression, such as fibroblasts. Following the administration of a low dose of 4-hydroxytamoxifen (4-OHT) at P1, Cre-mediated recombination and irreversible expression of green fluorescent protein (GFP) labeled a limited number of mural cells in different postnatal organs. Analysis of GFP+ cells by confocal microscopy at high resolution (Fig. 1a–c) revealed substantial differences in pericyte abundance and morphology. GFP+ pericytes in the postnatal brain cortex at P12 extend numerous short cellular protrusions and are densely covering capillaries (Fig. 1a). Moreover, *Pdgfrb-CreERT2*-labeled brain pericytes are located in close proximity to astrocytes expressing glial fibrillary acidic protein (GFAP) and microglia expressing Allograft inflammatory factor 1 (AIF1) (Fig. 1b). In contrast, lung pericytes are comparably sparse and extend multiple long cellular processes contacting ECs (Fig. 1c). In both brain and lung, there is good colocalization between GFP signal and PDGFRβ immunostaining, which is most prominent on the cell body (Fig. 1a, c). In the lung, very little GFP signal is seen outside of the vasculature, consistent with our previous work showing that activation of *Pdgfrb-CreERT2*-mediated recombination with tamoxifen administration at P1–P3 does not lead to labeling of PDGFRα+ fibroblasts, alpha smooth muscle actin (αSMA)+ bronchial smooth muscle cells or myofibroblasts[22]. In addition, PDGFRβ immunostaining of pericytes is predominantly confined to the alveolar septum, the tissue between apposing alveolar walls (Fig. 1d, f). Immunostaining indicates that alveolar type 2 (AT2) epithelial cells express Prosurfactant Protein C (proSP-C), which acts as a stem cell in the pulmonary epithelium and gives rise to terminally differentiated RAGE+ (receptor for advanced glycation end products)[32,33]. AT2 cells are also enriched in the septum region (Fig. 1e, f).

As previous work has uncovered evidence for the expression of organ-specific markers both in mice and humans[19,34,35], we aimed at the identification of pericyte-derived, paracrine-acting factors that might potentially control morphogenetic processes during postnatal development. Comparison of a public scRNA-seq resource for murine pulmonary development provided by the Thébaud laboratory[36] with our own scRNA-seq data, which is introduced later in the course of this article, indicates that pericytes from wild-type postnatal lung and brain express the expected general mural cell markers, namely *Pdgfrb* (encoding Platelet-derived growth factor receptor β, PDGFRβ), *Cspg4* (Chondroitin Sulfate Proteoglycan 4), and *Notch3*, which encodes a Notch family receptor (Fig. 1g, i). In addition, pericytes from the lung and the brain show organ-specific gene expression. In particular, we identified several differentially expressed transcripts for secreted factors. *Hgf* (encoding hepatocyte growth factor) and *Bdnf* (brain-derived neurotrophic factor), a regulator of axon guidance through its receptor TrkB/Ntrk2[37,38], are found in pericytes of the lung but are not detectable in the brain (Fig. 1g–i and Supplementary Data 1–4). *Angpt1* (Angiopoietin 1), an important regulator of vascular growth and integrity, shows a similar distribution pattern of higher expression in lung relative to brain pericytes, which is consistent with previous work[22] and published scRNA-seq results from adult lung and brain[21,39,40]. In turn, *Nodal*, a member of the TGFβ superfamily, is enriched in brain pericytes relative to lung (Fig. 1g–i and Supplementary Data 1–4). Based on these findings and bioinformatic analyses pointing at potential interactions of these pericyte-derived factors and neighboring cells in brain and lung (Supplementary Fig. 1a, b and Fig. 2a–d), we used an inducible genetic strategy involving the *Pdgfrb-CreERT2* line in combination with loxP-flanked alleles for three candidate genes, namely *Hgf*, *Bdnf* and *Nodal*, for functional studies during postnatal organ development.

### Pericyte-derived BDNF controls pulmonary angiogenesis and development

To investigate the function of *Bdnf* in pulmonary pericytes in mice, conditional *Bdnf* alleles were introduced into the *Pdgfrb-CreERT2* background. Following tamoxifen administration at P1-3, ICAM2 immunostaining shows that vascularization is reduced in P21 lungs of the resulting *Bdnf*[iPCKO] mutants (i.e., inducible *Pdgfrb-CreERT2*-mediated knockouts) relative to littermate controls (Fig. 2a, f). The number of ECs, visualized by nuclear ERG immunostaining, and the number of PDGFRβ+ cells are decreased in *Bdnf*[iPCKO] lungs (Fig. 2b, c, g, h), whereas the ratio of ERG+ to PDGFRβ+ cells is not significantly changed (Fig. 2i). Lung immunostaining for EdU+ endothelial cells shows a significant reduction of proliferating ERG+ cells in *Bdnf*[iPCKO] lungs compared to controls (Fig. 2d, j). In addition to the endothelial defects, immunostaining for NKX2.1 (a transcription factor expressed by both AT1 and AT2 cells) and LAMP3 (an AT2 cell marker) shows that the number of AT1 and AT2 epithelial cells is also decreased in P21 *Bdnf*[iPCKO] lungs (Fig. 2e, k). Consequently, the mutants exhibit enlarged airspaces and a significant decrease in total lung volume (Fig. 2l–n).

Bioinformatic analysis of scRNA-seq data from P21 *Bdnf*[iPCKO] and littermate control lungs (see the "Methods" section) confirms the expected reduction of *Bdnf* transcripts in mutant pericytes (Supplementary Fig. 3a) but also reveals significant changes in gene expression in the mutant endothelium (Supplementary Fig. 3b–d). Expression of multiple endothelial markers, including *Plvap* (encoding plasmalemma vesicle-associated protein, associated with EC permeability), *Gja5* (encoding Gap Junction Protein Alpha 5 protein), *Itga1* (encoding Integrin Subunit Alpha 1), and *Aplnr* (Apelin receptor) are downregulated. The proliferation markers *Ccnd1* (Cyclin D1), *Ccna2* (Cyclin A2) and *Mki67* (Ki67) are also reduced in mutant lung ECs (Supplementary Fig. 3d, e). Arguing that pericyte-derived BDNF might act on

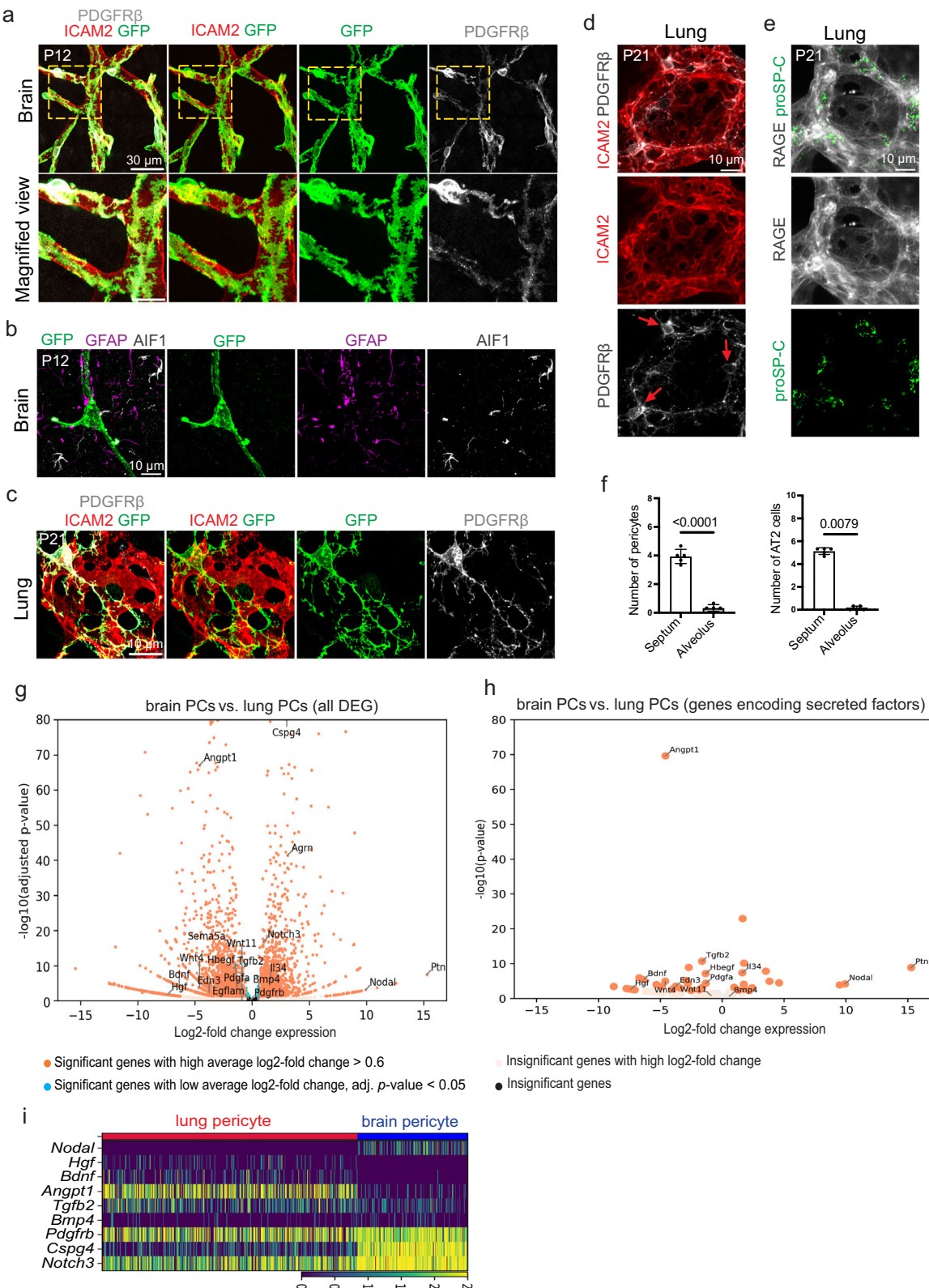

ECs, the receptor tyrosine kinase TrkB, a high affinity receptor of BDNF encoded by the gene *Ntrk2*[41], is detected in ICAM2+ ECs in P21 lungs by immunostaining (Fig. 2o). Similarly, scRNA-seq data shows that *Ntrk2* is predominantly expressed by general capillary (gCap) ECs in addition to mesenchymal cells in the P21 lung (Fig. 2p and Supplementary Fig. 3f). Further arguing for the relevance of pericyte-derived BDNF, TrkB phosphorylation at tyrosine residue 515 (Tyr515), an indicator of

receptor activation, is substantially reduced in lysates from P21 *Bdnf*[iPCKO] lungs relative to controls (Fig. 2q). Immunostaining of phospho-TrkB shows a strong reduction of the signal in *Bdnf*[iPCKO] ICAM2+ ECs relative to control (Supplementary Fig. 3g, h).

Given that vascularization is a prerequisite for normal pulmonary epithelial morphogenesis[42] and because of endothelial *Ntrk2* expression and TrkB phosphorylation (Fig. 2o, p), we used a conditional gene

**Fig. 1 | Characterization of lung and brain pericytes. a** Labeling of pericytes in the P12 brain cortex via *Pdgfrb(BAC)-CreERT2*-mediated, tamoxifen-induced GFP expression (green). Confocal images also show PDGFR (gray), a pericyte marker, and ICAM2, labeling ECs (red). Bottom panels show higher magnification of the yellow dashed boxes in the top row. **b** Confocal images showing the proximity of GFP+ pericytes (green), Glial fibrillary acidic protein (GFAP)+ astrocytes (magenta) and Allograft inflammatory factor-1 (AIF)+ microglia (gray) in the P12 brain cortex. **c** *Pdgfrb(BAC)-CreERT2*-mediated labeling of pericytes (GFP, green) in the P21 lung. Confocal images also show PDGFRβ (gray) and ICAM2 (red). **d** Maximum intensity projections showing PDGFRβ+ pericytes (gray) and ICAM2+ ECs (red) in the lung alveolus. Red arrows indicate pulmonary pericytes in the alveolar septum. **e** Confocal images showing the localization of Prosurfactant Protein C (proSP-C)-expressing type 2 alveolar epithelial cells (green) and receptor for advanced glycation end products (RAGE)-positive type 1 alveolar epithelial cells (gray). **f** Graphs showing the number (per area/82 × 82 × 6 μm) of pericytes and type 2 alveolar epithelial cells in the septal and alveolar regions of lung alveoli. Data represent mean ± s.e.m. (*n* = 5 mice per group); *P*-values, unpaired two-tailed Student *t*-test (pericytes) and two-tailed Mann–Whitney test (alveolar epithelial cells). Volcano plots showing the differential expression of genes (**g**) and differentially expressed ligands (**h**) in pericytes from P14 lung[36] versus P12 brain. Average log2-fold change > 0.6, adjusted *p*-value < 0.05). **i** Heatmap of differentially expressed ligands (padj < 0.01) in pericytes from postnatal lung[36] versus brain. Pseudobulk DE analysis uses two-sided Wald test + independent filtering as implemented by pyDESeq2 (**g–i**). Source data are provided as a Source Data file.

targeting strategy to study the function of the receptor in ECs. *Cdh5-CreERT2*-controlled inactivation of *Ntrk2*, induced by tamoxifen administration at P1–3, leads to the reduction of ICAM2+ area, ERG+ cells (Fig. 2r–u) and proliferative endothelial cells (Fig. 2s, x) in the resulting *Ntrk2*[iΔEC] mutant lungs at P21. Although the number of PDGFRβ+ cells is also decreased in *Ntrk2*[iΔEC] mutant lungs, the ratio of ECs to pericytes is comparable in *Ntrk2*[iΔEC] and littermate control lungs (Fig. 2r, v, w), consistent with the findings in *Bdnf*[iPCKO] mutants. Overall, these data indicate that pericyte-derived BDNF controls TrkB signaling in ECs and thereby pulmonary angiogenesis, which, in turn, is indispensable for lung development. These findings are consistent with earlier studies proposing roles of BDNF-TrkB signaling in the regulation of integrin or PI3K/Akt-mediated EC migration as well as cell survival[41,43,44]. Consistent with the phenotype observed in *Bdnf*[iPCKO] lungs (Fig. 2l, m), *Ntrk2*[iΔEC] lungs exhibit pronounced airspace enlargement (Fig. 2y) and a significant decrease in total lung volume (Fig. 2z). Despite these structural abnormalities, no apparent alterations in vital physiological functions such as arterial oxygen saturation and respiratory rates were detected in either *Bdnf*[iPCKO] (Supplementary Fig. 3i, j) or *Ntrk2*[iΔEC] mice (Supplementary Fig. 3k, l).

### Pericyte-derived BDNF or HGF is dispensable in the postnatal brain

In contrast to the lung phenotype, quantitative analysis of the ICAM2+ vascular area and number of proliferating ECs (ICAM2+, EdU+) in the P12 *Bdnf*[iPCKO] brain shows no significant differences relative to littermate controls (Supplementary Fig. 4a–c). In addition, combined ERG and PDGFRβ staining indicates that the number of ECs, pericytes and the ratio of endothelial cells (ERG+) to pericytes (PDGFRβ+) remains unaltered in the postnatal *Bdnf*[iPCKO] brain (Supplementary Fig. 4d–g). We also did not observe any changes in collagen IV expression, which is a major constituent of the vascular basement membrane[45] (Supplementary Fig. 4h). Impairment of the blood-brain barrier (BBB) can induce neuronal injury and neuroinflammation[46]. Double immunostaining for ICAM2 and blood-derived immunglobulin G (IgG) shows that the signal is detectable inside the vasculature but absent within the brain parenchyma (Supplementary Fig. 4i). Moreover, confocal images of ICAM2+ cortical blood vessels show no obvious defects in Aquaporin-4-expressing (AQP4+) astrocyte endfeet, indicating that this key component of the neurovascular unit is maintained in the *Bdnf*[iPCKO] brain vasculature (Supplementary Fig. 4j, m). Reactive astrogliosis, a typical response to CNS injury[47], is also absent, as indicated by normal levels of GFAP immunostaining (Supplementary Fig. 4k, n). Activation of microglia, detectable through the upregulation of the marker allograft inflammatory factor 1 (AIF1) and morphological changes, such as increased cell volume and the formation of elongated cellular processes[48–51], is also not altered in *Bdnf*[iPCKO] relative to littermate controls (Supplementary Fig. 4i, o, p). Similar to the absence of a phenotype in P12 *Bdnf*[iPCKO], examination of mutant brains at P21 reveals no significant changes in vascular density (Supplementary Fig. 5 a, b), coverage by AQP4+ astrocyte endfeet (Supplementary Fig. 5a, c), and the

abundance of GFAP+ astrocytes and AIF+ microglial cells (Supplementary Fig. 5d–h). Together, these data establish that *Pdgfrb-CreERT2*-controlled inactivation of the *Bdnf* gene causes no overt alterations in the postnatal cortex, which is consistent with the absent expression of the neurotrophic factor in brain pericytes.

As mentioned earlier, *Hgf* expression is also absent from brain pericytes. Accordingly, a similar immunostaining analysis of brain sections from P12 *Hgf*[iPCKO] mutants, generated with the *Pdgfrb-CreERT2* line and postnatal tamoxifen administration from P1-3, showed no obvious differences to control littermates. Quantitative analyses of vascular area, the number of proliferative endothelial cells, the ratio of endothelial cells to pericytes, and collagen type IV expression are comparable in the *Hgf*[iPCKO] and control brain vasculature (Supplementary Fig. 6a–h). Double immunostaining for ICAM2 and IgG shows that IgG is confined to the vasculature, with no detectable leakage into the brain parenchyma (Supplementary Fig. 6i). Furthermore, immunostaining shows no noticeable differences in AQP4+ glial endfeet, GFAP+ astrocytes, AIF1 expression by microglia, and average microglial volume (Supplementary Fig. 6j–p), a pattern that was also observed in P21 *Hgf*[iPCKO] brains (Supplementary Fig. 7a–h).

Taken together, these results indicate that *Pdgfrb-CreERT2*-controlled inactivation of *Hgf* or *Bdnf* leads to no detectable alterations in the postnatal brain, which is consistent with the absent or very low expression of these two growth factors in brain pericytes.

### Alveolarization and lung morphogenesis require pericyte-derived HGF

Given the importance of HGF signaling and its receptor c-Met for lung development[22,52–54] and based on the expression of *Hgf* in pulmonary pericytes (Fig. 1g–i), we investigated the lung phenotype of *Hgf*[iPCKO] mutants at P21 following tamoxifen administration after birth. This analysis focused on LAMP3 + AT2 cells, which, as immunostaining shows, exhibit strong c-Met protein expression in the lung (Fig. 3a and Supplementary Fig. 8a–d). Likewise, scRNA-seq data show enrichment of *Met* transcripts in AT2 cell clusters (Supplementary Fig. 8a and Supplementary Fig. 9d–g). *Pdgfrb-CreERT2*-controlled inactivation of *Hgf* (Supplementary Fig. 9a) impairs alveologenesis (Fig. 3b), which involves a significant reduction of AT2 epithelial cells as well as reduced AT2 cell proliferation, whereas apoptosis is not increased in *Hgf*[iPCKO] mutant lungs (Fig. 3c–g, i–k). Interestingly, we observed hotspots with high levels of staining for the transmembrane pattern recognition receptor RAGE, which is not only a marker for AT1 cells but has also been linked to inflammatory processes in the lung[55]. The overall RAGE + AT1 epithelial area, however, is reduced in *Hgf*[iPCKO] lungs compared to littermate controls (Fig. 3h, l), reflecting that the AT2 population serves as a progenitor pool for AT1 cells[32]. We also found that *Hgf*[iPCKO] mutant lungs exhibit markedly expanded airspaces and a significantly reduced overall lung volume (Fig. 3m, n).

To investigate the cellular and molecular alterations in *Hgf*[iPCKO] animals in greater detail, we performed scRNA-seq analysis of P21 mutant and littermate control lungs (Supplementary Fig. 9a–h).

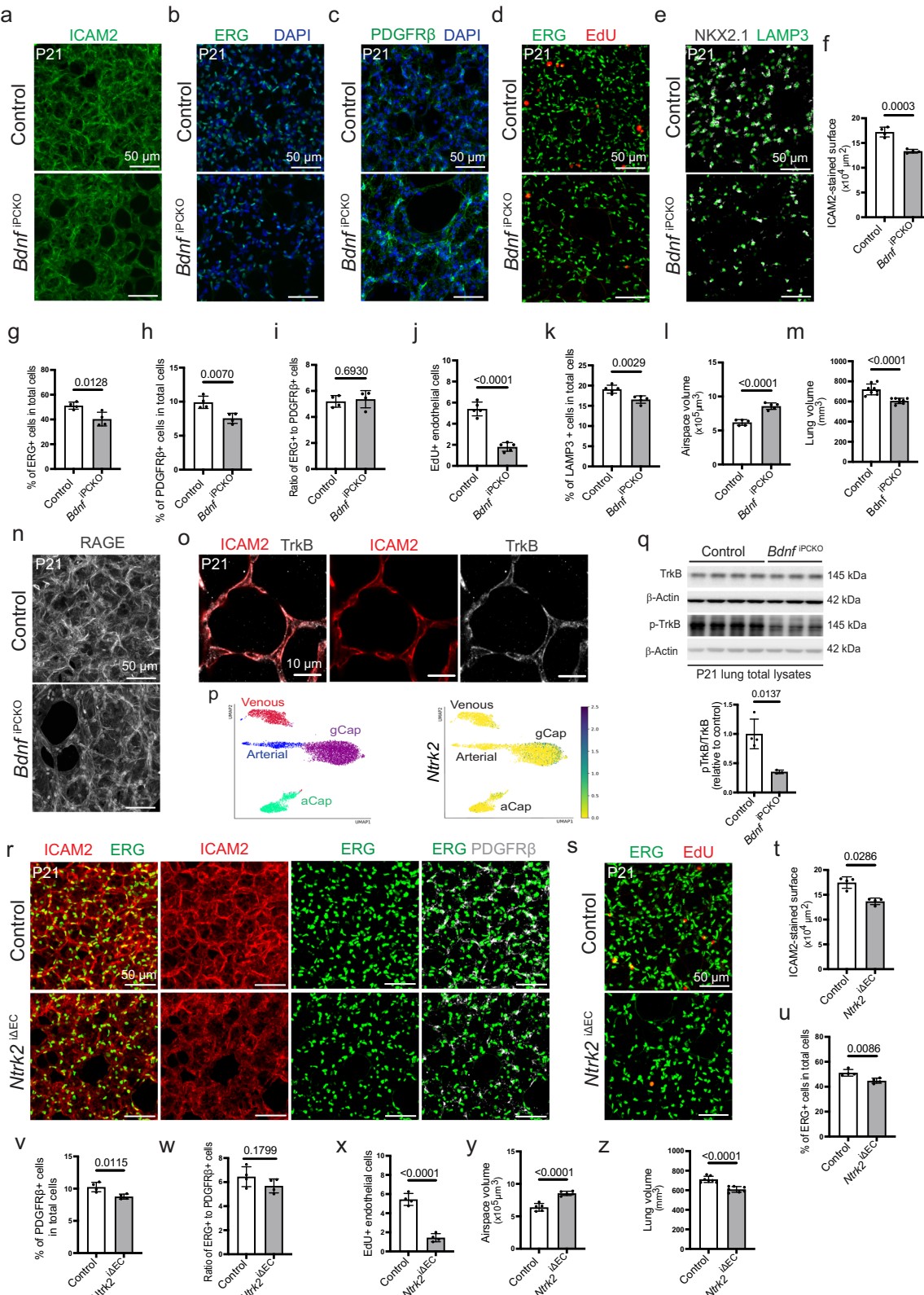

Bioinformatic analysis of the resulting data confirmed the expected strong reduction of Hgf transcripts in pulmonary pericytes relative to control (Supplementary Fig. 9a). The data also shows a reduction of AT2 cells expressing *Met*, and *Lamp3, Abca*, and *Tinag* in *Hgf*^iPCKO lungs (Fig. 3o and Supplementary Fig. 9h). Furthermore, *Hgf*^iPCKO AT2 cells show reduced expression of genes associated with cell proliferation (Fig. 3p) and markers associated with differentiation into AT1

epithelium are also affected in the AT1–AT2 transitory population (Fig. 3q). Despite pronounced morphological defects, overall respiratory function, including arterial oxygen saturation and breathing rate at the resting state, is preserved in P21 *Hgf*^iPCKO (Supplementary Fig. 8e, f).

In addition to the epithelial defects, mice lacking PDGFRβ+ cell-derived HGF show a significant decrease of PECAM1+ pulmonary

**Fig. 2 | Pulmonary vascular development is regulated by BDNF-TrkB signaling.** **a**, **b** Confocal images of *Bdnf*[iPCKO] and littermate control lungs stained for ICAM2 (green) (**a**) or ERG (green) and DAPI (blue) (**b**). **c** Confocal images of *Bdnf*[iPCKO] and littermate control lungs showing PDGFRβ-labeled pericytes (green) and DAPI-stained nuclei (blue). **d** Confocal images of *Bdnf*[iPCKO] and littermate control lungs showing EdU (red)-stained ECs (ERG+, green). **e** Confocal images of NKX2.1-stained alveolar epithelial cells (gray) and LAMP3+ type 2 alveolar epithelial cells (green) in *Bdnf*[iPCKO] and littermate control lungs. **f**–**m** Graphs showing the ICAM2-stained vascular density based on 3D reconstruction surface images, as shown in (**a**) (**f**), percentage of ERG+ EC nuclei in total cells (**g**), percentage of PDGFRβ+ cells in total cells (**h**), ratio of ERG+ cells to PDGFRβ+ cells (**i**), the number of EdU+ ECs per area (283 × 283 × 22 μm) (**j**), ratio of LAMP3+ AT2 epithelial cells to total cells (**k**), airspace volume (**l**), and lung volume measurement (**m**) in P21 *Bdnf*[iPCKO] and littermate control lungs. Data represent mean ± s.e.m. (*n* = 4 (**f**–**i**), *n* = 5 (**j**–**l**), *n* = 8 (**m**); *P*-values, unpaired two-tailed Student *t*-test. **n** Confocal images of *Bdnf*[iPCKO] and control lungs stained for RAGE (gray). **o** Confocal images showing TrkB immunostaining (gray) in ICAM2+ (red) ECs in pulmonary capillaries. **p** Validation of expression of *Ntrk2* (encoding TrkB) in ECs by scRNA-seq analysis. UMAP plot showing color-coded EC subclusters, namely arterial, venous, general capillary (gCap) and aerocytes (aCap), in P21 lung (left). *Ntrk2* expression in gCap endothelial cells (right). **q** Western blot showing TrkB and Phospho-TrkB (p-TrkB) in *Bdnf*[iPCKO] and littermate control total lung lysates. Molecular weight marker (kDa) is indicated. β-Actin is shown as a loading control. Quantitation of p-TrkB/TrkB ratio is shown in the graph below. Data represent mean ± s.e.m. (*n* = 4 control and 3 mutants); *P*-values, unpaired two-tailed Welch's test. **r** Confocal images of ICAM2+ (red) and ERG+ (green) endothelial cells, and PDGFRβ+ cells (gray) in EC-specific *Ntrk2*[iΔEC] loss-of-function mutant and littermate control lungs. **s** Confocal images of *Ntrk2*[iΔEC] and control lungs showing EdU (red)-stained ECs (ERG+, green).Graphs showing the ICAM2-stained vascular density (**t**), percentage of ERG + EC nuclei in total cells (**u**), percentage of PDGFRβ+ cells in total cells (**v**), ratio of ERG+ cells to PDGFRβ+ cells (**w**), the number of EdU+ ECs per area (283 × 283 × 22 μm) (**x**), airspace volume (**y**), and lung volume measurement (**z**) in P21 *Ntrk2*[iΔEC] and littermate control lungs. Data represent mean ± s.e.m. (*n* = 4 (**t**–**x**), *n* = 5 (**y**), *n* = 7 (**z**); *P*-values, unpaired two-tailed Mann–Whitney test in **t** and unpaired two-tailed Student *t*-test in (**u**–**z**). Source data are provided as a Source Data file.

endothelium relative to controls (Fig. 3r, t), whereas the ratio of ECs to pericytes, measured by combined ERG and PDGFRβ staining, remains unchanged (Fig. 3s, u–w). Given the absent or very low expression of *Met* in endothelial cells (Supplementary Fig. 8a, d and Supplementary Fig. 9d), the reduced vascularization could be secondary to the defects in alveolar epithelium observed in postnatal *Hgf*[iPCKO] lung. In fact, AT1 cell-derived VEGF-A was previously shown to regulate the differentiation of pulmonary ECs[26,56].

The sum of these data, together with our previous study[22] confirm the importance of pericyte-derived HGF for lung development, which we attribute to reduced AT2 epithelial cell proliferation.

## Pericyte-derived *Nodal* is not essential for postnatal lung development

Postnatal *Nodal*[iPCKO] mutants were generated with *Pdgfrb-CreERT2* mice using the same strategy as outlined above for *Hgf* and *Bdnf*. Confocal microscopy and PECAM1 immunostaining show no significant changes in the postnatal *Nodal*[iPCKO] pulmonary vasculature (Supplementary Fig. 10a, g). Likewise, combined ERG, PDGFRβ, and DAPI staining indicate that the number of ECs, pericytes and the ratio of endothelial cells (ERG+) to pericytes (PDGFRβ+) are comparable in mutants and the corresponding littermate controls (Supplementary Fig. 10b, h–j). Moreover, the number of proliferating endothelial cells is not changed in *Nodal*[iPCKO] lungs compared to control littermates (Supplementary Fig. 10c, k). The lung epithelium, stained with RAGE and NKX2.1, is also not altered in *Nodal*[iPCKO] mutants (Supplementary Fig. 10d, e, l). The percentage of AT2 cells as well as the number of proliferating AT2 cells, airspace and lung volume are not different in *Nodal*[iPCKO] and control littermate lungs (Supplementary Fig. 10e, f, m–p). Overall, respiratory function is preserved in P21 *Nodal*[iPCKO], with no measurable differences in arterial oxygen saturation or respiratory rate in the resting state (Supplementary Fig. 10q, r).

Together, these data indicate that *Pdgfrb-CreERT2*-controlled inactivation of *Nodal* leads to no overt alterations in the postnatal lung, which is consistent with the absent or very low expression of this growth factor in pulmonary pericytes.

## Postnatal *Nodal*[iPCKO] mutants show reduced brain vascularization

*Nodal* transcripts are prominently expressed by brain pericytes (Fig. 1g–i, Supplementary Fig. 11a–d), which is further supported using a public scRNA-seq resource for adolescent mouse brain (Supplementary Fig. 11e, f) provided by the Linnarsson laboratory (mousebrain.org)[40]. Inducible inactivation of *Nodal* mediated by *Pdgfrb-CreERT2* results in strongly reduced transcript levels in brain pericytes (Supplementary Fig. 11d) and impaired growth of the brain vasculature by P12 (Fig. 4a–c).

Microscopic and quantitative analyses reveal a decrease in vascular area together with a reduction of endothelial cells and pericytes in *Nodal*[iPCKO] brains relative to littermate controls, whereas the ratio of PDGFRβ+ pericytes to capillary EC remains unchanged (Fig. 4b, d, f–h).

Consistent with the reduction of *Nodal*[iPCKO] brain capillaries, EdU administration reveals a significant decrease in EC proliferation (Fig. 4e, i). In addition, confocal images of the *Nodal*[iPCKO] brain cortex reveal small, isolated areas of Texas Red-conjugated dextran (70 kDa), serum IgG and Ter119+ red blood cell extravasation (Supplementary Fig. 12a, d, f). Vascular ensheathment by AQP4+ astrocyte endfeet, however, is maintained in the *Nodal*[iPCKO] mutant cortex (Supplementary Fig. 12b, c). Together, these data indicate that pericyte-derived Nodal controls vessel growth and integrity in the postnatal brain.

Previous work has indicated that Nodal is a positive regulator of tumor angiogenesis and can promote tube-formation by cultured human umbilical vein endothelial cells (HUVECs)[57,58]. To gain insight into the regulation of EC behavior by Nodal, we stimulated murine immortalized brain endothelial (b.End3) cells with recombinant rhNodal and analyzed the phosphorylation of the downstream signal transducer SMAD2 by Western blotting. Treatment with rhNodal increases the level of phosphorylated SMAD2 (p-SMAD2) in b.End3 lysates in a dose-dependent fashion without altering total SMAD2 (Fig. 4j), which is prevented by the addition of SB431542, an inhibitor of TGFβ type I receptors. Similarly, rhNodal stimulates b.End3 proliferation and migration in a scratch wound assay and, again, these effects are suppressed by SB431542 (Fig. 4k–n). Together, these data support that Nodal can regulate features of vascular growth directly through the stimulation of ECs.

## Reactive astrogliosis is negatively modulated by Nodal signaling

The analysis of brain sections by immunostaining for GFAP and confocal microscopy revealed signs of reactive astrogliosis, which are most evident in the *Nodal*[iPCKO] cortex and brainstem (Fig. 5a). Reactive astrogliosis, a typical response to CNS injury and a range of diseases[47], involves a spectrum of molecular, cellular and functional changes in astrocytes. Remarkably, a highly increased number of reactive astrocytes, characterized by hypertrophy and an increase of GFAP+ protrusions, can be detected in both sagittal (Fig. 5a, b, d) and coronal (Fig. 5c) sections of the *Nodal*[iPCKO] brain cortex. In addition, transmission electron microscopy of mutant and littermate control brain cortex confirms the enrichment of glycogen granules and intermediate filaments in mutant astrocytes (Fig. 5e), which are two features of reactive astrocytes in pathologic conditions[59].

Confocal microscopy of the P12 *Nodal*[iPCKO] brain cortex shows that reactive GFAP+ astrocytes are present in regions of vascular leakage, as indicated by TER119 or IgG extravasation (Supplementary Fig. 12d, f), which raises the possibility that reactive astrogliosis may occur

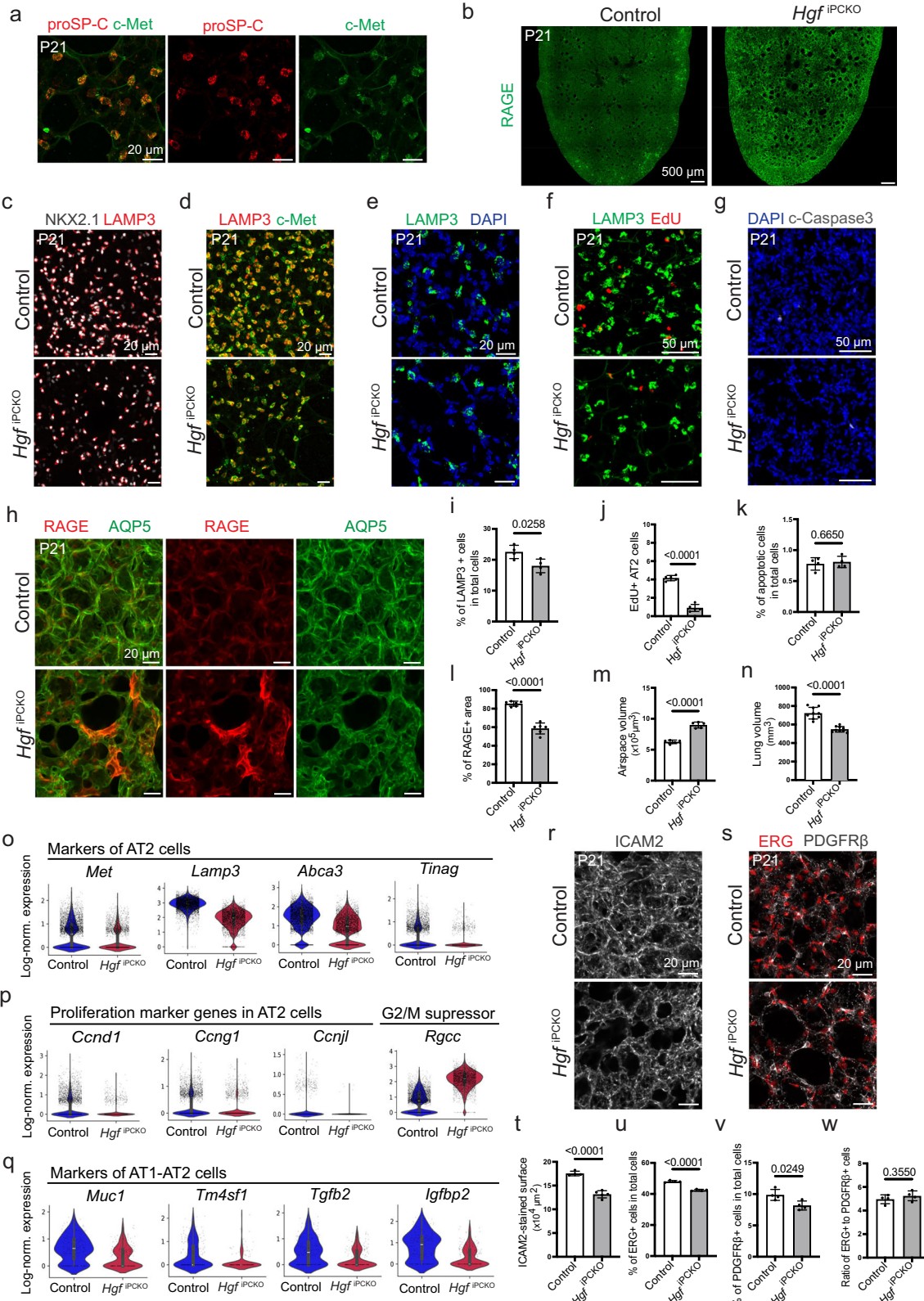

secondary to vascular defects. However, GFAP+ astrocytes are also present in regions without detectable red blood cell (Ter119+, red) (Supplementary Fig. 12e) or IgG leakage (Supplementary Fig. 12g). Notably, GFAP+ astrocytes are already increased in the P8 *Nodal*[iPCKO] brain cortex in the absence of detectable leakage, indicating that reactive astrogliosis is initiated prior to the emergence of vascular defects (Supplementary Fig. 13a–c).

For a more detailed investigation of the cellular and molecular alterations in *Nodal*[iPCKO] mutants, we performed scRNA-seq analysis of the P12 brain cortex from mutants and littermate controls (Fig. 5f, g and Supplementary Fig. 11a–d). Bioinformatic analysis of the resulting data confirms the increase of reactive astrocytes in *Nodal*[iPCKO] mutants relative to littermate controls (Fig. 5f). Moreover, our scRNA-seq data shows the upregulated expression of genes associated with reactive

**Fig. 3 | Pericyte-derived HGF promotes lung alveolarization. a** Confocal images of P21 lung section showing proSP-C+ AT2 cells (red) and co-staining by c-Met (green). **b** Tile scan confocal view of RAGE+ AT1 cells (green) in PDGFRβ cell-specific *Hgf* iPCKO loss-of-function and littermate control lungs. Confocal images of NKX2.1+ alveolar epithelial cells (gray) and LAMP3+ AT2 cells (red) (**c**), c-Met (green) and LAMP3 (red) co-staining (**d**), LAMP3+ AT2 cells (green) together with DAPI (blue) (**e**), LAMP3 (green) and EdU (red) (**f**), c-Caspase3 (gray) and nuclei (DAPI, blue) (**g**), RAGE (red) and AQP5 (green) (**h**) in *Hgf* iPCKO and littermate control lungs. **i−n** Graphs showing the percentage of LAMP3 positive cells in total cells (**i**), the number of proliferative LAMP3+ cells per area ($283 \times 283 \times 22\,\mu m$) (**j**), the percentage of apoptotic cells (**k**), the percentage of RAGE positive area (**l**), airspace volume (**m**), and lung volume measurement (**n**) in *Hgf* iPCKO and littermate control lungs. Data represent mean ± s.e.m. ($n = 4$ (**i, k**), $n = 5$ (**j, m**), $n = 6$ (**l**) and $n = 9$ (**n**)); *P*-values, unpaired two-tailed Student *t*-test. (**o-q**) Violin plots showing expression of AT2 cell-specific markers (**o**), proliferation markers in AT2 cells (**p**), and expression of type 1 and 2 alveolar epithelial cell-specific markers (**q**) in *Hgf* iPCKO and control samples ($n = 5229$ control and 3624 mutant cells; two animals per group); Boxplots in center of violins shows median (white bar), quartiles (box), and the farthest data point within 1.5 * inter-quartile range (whiskers). Confocal images of ICAM2+ pulmonary endothelium (gray) (**r**) and ERG+ ECs (red) together with PDGFRβ+ pericytes (gray) in P21 *Hgf* iPCKO and control lungs (**s**). Graphs showing the ICAM2-stained vascular density based on 3D reconstruction surface images, as shown in (**r**) (**t**), ratio of ERG+ EC nuclei to total cells (**u**), ratio of PDGFRβ+ cells to total cells (**v**), and ratio of ERG+ cells to PDGFRβ+ cells in P21 *Hgf* iPCKO and control lungs (**w**). Data represent mean ± s.e.m. ($n = 4$ for controls, $n = 5$ for mutants in **t**, and $n = 4$ in **u−w**); *P*-values, unpaired two tailed Student *t*-test. Source data are provided as a Source Data file.

astrocytes, including transcripts for glial fibrillary acidic protein (*Gfap*), AP-1 family transcription factors (*Jun, Junb, Fos, Fosb*), the amino acid transporter solute carrier family 7 member 5 (*Slc7a5*), the Wnt pathway protein Axin2 (*Axin2*), nuclear receptor 4A1 (*Nr4a1*), the transcription factors EGR-1, Sox2 and Sox9 (*Egr1, Sox2, Sox9*) and the cell cycle regulator Cyclin D1 (*Ccnd1*) in mutant relative to control brain (Fig. 5g and Supplementary Fig. 14a). Analysis of immunostained brain sections by confocal microscopy validates that the transcriptional regulators SOX2, SOX9 (Supplementary Fig. 14b−e) as well as FOS (Supplementary Fig. 15a, b) are enriched in *Nodal* iPCKO GFAP+ astrocytes.

In vitro experiments confirm that cultured mouse primary astrocytes are responsive to recombinant rhNodal protein. Western blot analysis of cell lysates from rhNodal-treated astrocytes reveals a significantly increased phosphorylation of the downstream signal transducer SMAD2 (p-SMAD2), which is abolished by the inhibitor SB431542 (Fig. 5h). These data support that pericyte-derived Nodal might directly regulate astrocyte behavior in the postnatal brain. Cell culture experiments also show that SB431542 administration leads to an increase in GFAP+ astrocytes (Supplementary Fig. 14f, g), which is consistent with the known role of TGFβ as a negative or, depending on context, positive regulator of astrocyte reactivity[60]. Supporting our findings from the scRNA-seq analysis, rhNodal stimulation reduces the nuclear immunostaining of FOS and FOSB (Supplementary Fig. 15c−f). Conversely, SB431542 treatment of astrocytes increases the nuclear signal for SOX9, FOS and FOSB (Supplementary Fig. 14f, h and Supplementary Fig. 15c−f).

Taken together, these findings indicate that the loss of Nodal can induce reactive astrogliosis directly, consistent with an angiocrine signaling role of the pericyte-derived TGFβ family ligand.

## Activation of microglia is negatively controlled by Nodal

Microglial cells are important for immune surveillance in the brain and undergo morphological changes, from ramified to ameboid-like morphology, in response to injury or pathogens[61−63]. This activation involves alterations in gene and protein expression, reflecting the role of microglia in maintaining brain homeostasis under both healthy and diseased conditions[64−69]. In our study, microglia activation was assessed by quantifying microglial morphology as well as by analyzing transcriptional changes, including chemokine expression, by scRNA-seq. AIF1 and cluster of differentiation 68 (CD68) are well-established markers of microglial cells in the CNS, which are commonly found in activated microglia. AIF1 contributes to membrane cytoskeleton rearrangement, while CD68 is a lysosomal protein that functions as a scavenger receptor[49,70−74]. Visualization of immunostained microglia by confocal microscopy shows that the expression of AIF1 and CD68 is strongly increased in P12 *Nodal* iPCKO brain sections relative to littermate controls (Fig. 6a−c). Upregulated expression of AIF1 and CD68 in *Nodal* iPCKO brain lysates is confirmed by Western blot analysis (Fig. 6d). Furthermore, *Nodal* iPCKO AIF1+ microglial cells acquire thick and elongated cellular processes, consistent with exposure to acute stress[75] (Fig. 6e−g).

The increased number of activated microglial cells in *Nodal* iPCKO mutant brains relative to littermate controls is confirmed by the analysis of our scRNA-seq data (Fig. 6h, i). Notably, multiple markers associated with microglial activation, including transcripts for the chemokines tumor necrosis factor (*Tnf*), Interleukin-1 alpha (*Il1a*), CD52 (*Cd52*), CD74 (*Cd74*), colony stimulating factor 1 (*Csf1*), C-X-C motif chemokine ligand 10 (*Cxcl10*) and 16 (*Cxcl16*), C-C motif chemokine 12 (*Ccl12*) and 4 (*Ccl4*), are upregulated in *Nodal* iPCKO microglia. Other markers of microglia activation, namely the AXL receptor tyrosine kinase (*Axl*)[76] and the intermediate filament protein Nestin (*Nes*)[77], are also increased after loss of pericyte-derived Nodal (Fig. 6j and Supplementary Fig. 16a). Immunostaining confirms the increase of Nestin, which is known to be elevated in microglia during inflammation[77], in *Nodal* iPCKO brain sections relative to littermate controls (Supplementary Fig. 16b, e). Confocal microscopy of P12 *Nodal* iPCKO brain cortex shows that activated AIF1+ microglia are present in regions of vascular leakage (Supplementary Fig. 17a), which raises the possibility that microglial activation may occur secondary to vascular defects. However, similar to activated astrocytes, AIF1+ microglia are also present in regions without extravasation of IgG or Texas Red-dextran (Supplementary Fig. 17b, c). Notably, enlarged microglia are already present in the P8 *Nodal* iPCKO brain cortex, a stage in which vascular leakage is not yet detected, indicating that the TGFβ family ligand can suppress microglia activation directly (Supplementary Fig. 18a−e).

For a more detailed investigation of the role of Nodal, we treated cultured primary murine brain-derived microglial cells with recombinant rhNodal. This treatment significantly increases the phosphorylation of SMAD2, which is blocked by administration of SB431542 (Fig. 6k). Consistent with previous reports, TGF-β signaling through SMAD2 regulates microglial activity and maintains a quiescent phenotype in the CNS[78,79]. Cell stimulation experiments also confirm the suppression of Nestin expression by rhNodal in cultured murine brain-derived microglia cells, whereas SB431542 treatment leads to strongly enhanced Nestin immunostaining (Supplementary Fig. 16c, f). Expression of the chemokine CXCL10, which has been linked to microglial activation and migration[51,80,81], is also reduced by rhNodal treatment and strongly increased by SB431542 (Supplementary Fig. 16d, g).

Based on a simplified classification, one can distinguish M1 microglia, which are pro-inflammatory and neurotoxic, and M2 microglia with anti-inflammatory and neuroprotective roles[64,82,83]. Computational analyses of our scRNA-seq data suggest that the loss of pericyte-derived Nodal signaling shifts microglial polarization towards the M1 type, indicating that pericyte-derived Nodal may have both anti-inflammatory and neuroprotective effects (Fig. 6l). However, this simple classification does not entirely capture the complexity of microglial responses in various neurodegenerative conditions[64,84].

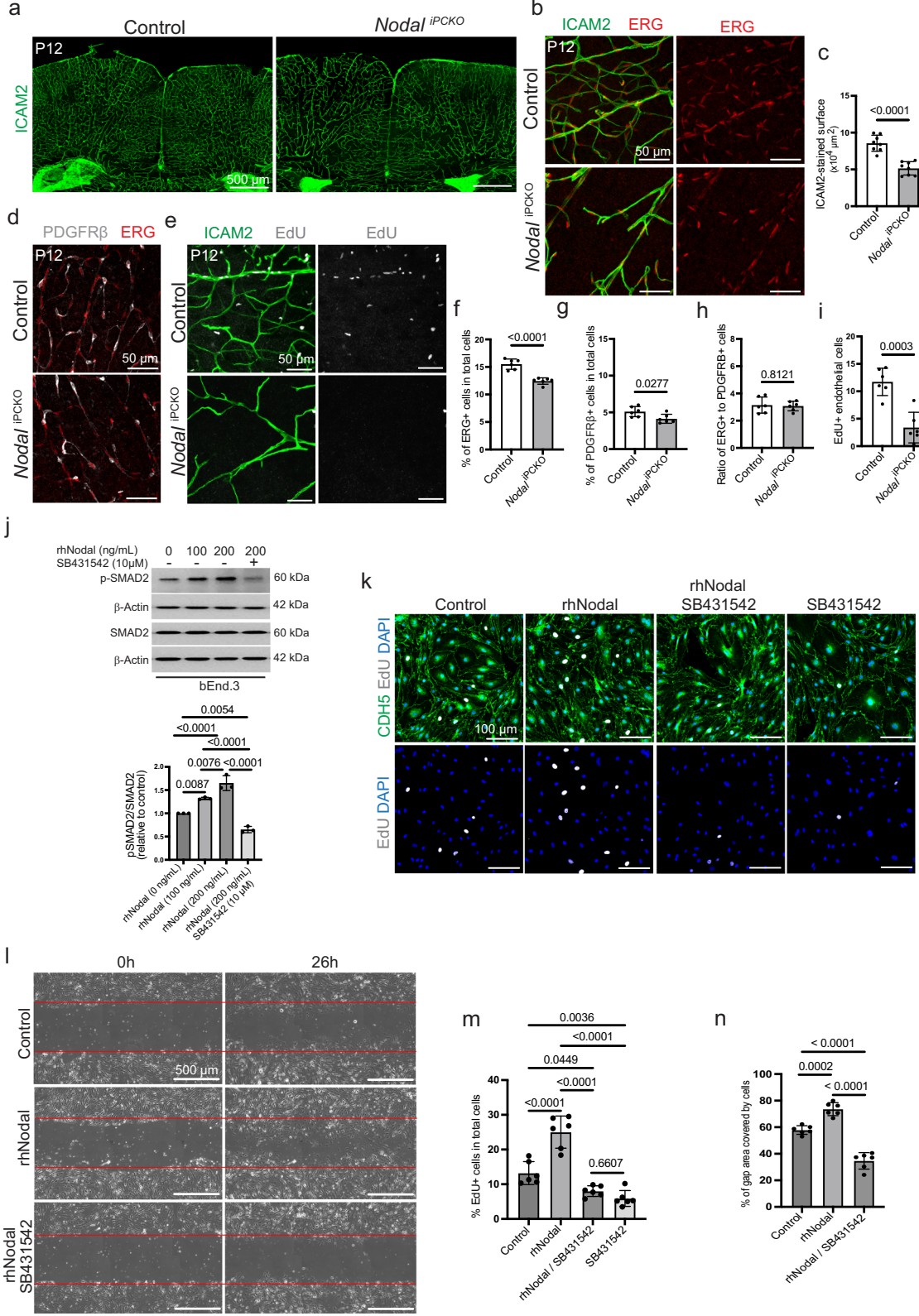

Taken together, our findings support that microglia can be directly activated by loss of Nodal, which is again consistent with an angiocrine signaling role of the pericyte-derived TGFβ family ligand. Subsequent alterations in the brain microenvironment, such as elevated vascular permeability or changes in ECs or astrocytes later in postnatal development, might further enhance the activation of microglia in *Nodal* [iPCKO] mutants.

## Discussion

Research on angiocrine signaling has uncovered very important roles of endothelial cells or specific EC subpopulations in organ growth and regeneration[1,2], whereas comparably little attention has been given to the potential role of pericytes as a source of secreted factors acting on cell populations in the surrounding tissue. In contrast, it is very well established that pericytes are critical for blood vessel integrity and the

**Fig. 4 | Pericyte-derived nodal is required for brain vascular development. a** Tile scan of coronal sections through P12 *Nodal*[iPCKO] and littermate control brain cortex immunostained for ICAM2 (green). **b** Confocal images of ICAM2+ (green) and ERG+ (red) endothelial cells in *Nodal*[iPCKO] and littermate control brain. **c** Graph showing the ICAM2-stained vascular density. Data represent mean ± s.e.m. (*n* = 8 mice per group); *P*-values, unpaired two-tailed Student *t*-test. Maximum intensity projections of P12 *Nodal*[iPCKO] and littermate control brain sections. Images show PDGFRβ+ pericytes (gray) and ERG+ (red) ECs (**d**), and EdU+ (gray) proliferating cells together with ICAM2+ (green) ECs (**e**). Graphs showing the ratio of ERG+ EC nuclei to total cells (**f**), ratio of PDGFRβ+ cells to total cells (**g**), ratio of ERG+ cells to PDGFRβ+ cells (**h**), and the number of EdU+ ECs per area (283 × 283 × 22 μm) in P12 *Nodal*[iPCKO] and littermate control brain cortex (**i**). Data represent mean ± s.e.m. (*n* = 6 in **f**–**i**); *P*-values, unpaired two-tailed Student *t*-test. **j** Western blot analysis of total and phosphorylated SMAD2 (p-SMAD2) in lysates of mouse brain endothelial cells (bEnd.3) treated with recombinant human Nodal (rhNodal) and SB431542 inhibitor, as indicated. Shown is a representative blot of three independent experiments. Molecular weight marker (kDa) is indicated. β-Actin is shown as a loading control. Quantitation of p-SMAD2/SMAD2 ratio is shown in the graph on the right. Data represent mean ± s.e.m. (*n* = 3 independent experiments); *P*-values, one-way ANOVA with Tukey's test. **k** Confocal images of VE-cadherin+ (CDH5, green) and EdU+ (white) bEnd.3 cells treated with rhNodal in the presence/absence of SB431542 inhibitor. **l** Bright field images showing that rhNODAL increases bEnd.3 cell migration in vitro in a scratch wound assay, which is blocked by SB431542. Left column shows the start of the assay (0 h) and red lines indicate the edges of scratch wounds, whereas images on the right are taken after 26 h. Graphs showing the impact of rhNodal on bEnd.3 cell proliferation (**m**) and migration (**n**). Data represent mean ± s.e.m. (*n* = 6); *P*-values, one-way ANOVA with Tukey's test. Source data are provided as a Source Data file.

prevention of excessive vascular leakage. The latter is most evident in the blood–brain barrier, where pericytes together with specialized ECs and astrocyte endfeet protect the brain against the entry of potentially harmful substances and cells from the blood stream[85,86]. In the lung, pericytes have been proposed to be a source of myofibroblasts driving tissue fibrosis[87,88] but have also been implicated in the regulation of leukocyte trafficking and cytokine signaling during inflammatory responses[89]. Historically, the reliable identification of pericytes has long been hampered by the lack of unique markers that are not shared by vascular smooth muscle cells, fibroblasts or other cell populations[90,91]. Accordingly, morphological criteria, in particular the close association with the endothelial monolayer, have been indispensable for the characterization of pericytes. These limitations are, at least to a great part, overcome by single-cell or single-nucleus RNA-seq approaches[19,34,35], which provide detailed molecular signatures and enable important insights into organ-specific and intra-organ heterogeneity of cells.

Stimulated by the availability of scRNA-seq data hinting at organ-specific angiocrine roles of pericytes, we have explored the role of three differentially expressed pericyte-derived candidate regulators, HGF, BDNF and Nodal in postnatal lung and brain, which were selected as model organs (Fig. 7a, b). In the lung, our data show that pericyte-derived HGF impairs alveologenesis, which we attribute to defective proliferation of c-Met-expressing AT2 cells. This finding is consistent with our own previous work showing that the YAP1 and TAZ, transcriptional regulators in the Hippo pathway, regulate the expression of HGF and angiopoietin-1 in pericytes[22]. The critical role of c-Met in AT2 cells is firmly established by previous studies, which have used diverse approaches such as in vitro stimulation approaches or pharmacological inhibition of HGF in vivo[92,93]. Most importantly, it has been established that the AT2 cell-specific inactivation of the *Met* gene impairs airspace morphogenesis but also pulmonary vascular development[53]. As AT2 cell-derived alveolar type 1 epithelial cells are an important source of vascular endothelial growth factor A (VEGF-A), a master regulator of angiogenic blood vessel growth, during lung development[56], impaired epithelial morphogenesis might directly lead to vascular defects. Pericyte-derived HGF may also have clinical relevance, as reduced HGF levels and mutations in the human *HGF* gene increase the risk of severe bronchopulmonary dysplasia in premature infants[94,95]. Conversely, treatment with recombinant HGF was shown to protect neonatal mice from hyperoxia-induced lung injury[96]. In adults, HGF modulates oxidative stress in alveolar epithelial cells, and lower HGF levels correlate with more severe chronic obstructive pulmonary disease (COPD)[53,97,98].

We also identify pericyte-derived BDNF as an important regulator of postnatal lung morphogenesis. The neurotrophin and its receptor, the Trk family receptor tyrosine kinase TrkB, are best known role in the formation and function of the nervous system[99]. However, more recent work has identified a specialized population of AT2 cells during lung regeneration in the adult mouse as a critical source of the neurotrophin. BDNF signals to TrkB+ mesenchymal alveolar niche cells, which have been shown to promote epithelial self-renewal and myofibrogenesis in response to lung injury[68,100]. BDNF signaling has also been linked to various aspects of blood vessel growth, EC migration and survival[41,43,44]. Our cell type-specific genetic experiments now demonstrate that interactions via pericyte-derived BDNF and endothelial TrkB directly contribute to postnatal lung development.

Remarkably, *Pdgfrb-CreERT2*-controlled inactivation of the *Hgf* or *Bdnf* genes causes no notable alterations in the postnatal brain, whereas the third factor investigated in our study, the TGFβ family ligand Nodal, is not expressed by pulmonary pericytes, explaining the absence of defects in *Nodal*[iPCKO] lungs. In contrast, we find that the loss of pericyte-derived Nodal impairs EC proliferation, decreases the vascularization of the postnatal brain, and causes microhemorrhaging. Previous studies have reported that Nodal promotes vascularization in breast cancer[101], and its inhibition suppresses angiogenesis and the progression of human gliomas[57]. *Nodal*[iPCKO] brains show signs of reactive astrogliosis and microglia activation. Although astrocytes are known to respond to defects in endothelial cells or compromised BBB integrity[102–104], our data argue that Nodal can directly influence astrocyte reactivity. As reactive astrocytes are typically induced by neuroinflammation or brain injury and ischemia[105–107], future work might address the expression and function of Nodal in pathophysiological settings. Given that TGFβ treatment has been shown to reduce the expression of reactive astrocyte-associated genes in culture[107], potentially redundant roles of other TGFβ superfamily members need to be considered. Moreover, reactive astrocytes can be induced by cytokines secreted by reactive microglia, including Il-1α, TNFα, and C1q[107], which generates a scenario of molecular crosstalk between multiple cell populations in and around the brain vasculature. The crosstalk between microglia and other components of the BBB, including pericytes, is essential for CNS homeostasis and maintaining a healthy brain environment[107–110]. Disruption of this communication can lead to neuroinflammation and contribute to various neuropathological conditions[64,107,111]. It is therefore important that Nodal negatively regulates the enrichment of factors associated with microglial activation[51,82,107,112,113]. Pericyte-derived Nodal may have context-dependent clinical relevance in the brain. Nodal upregulation promotes tumor progression in glioblastoma and other cancers, whereas its inhibition suppresses glioma angiogenesis and growth[57,114]. Overexpression of Nodal was also shown to be neuroprotective by mitigating inflammation and oxidative stress in hypoxic–ischemic brain injury[115], but the natural source of the growth factor in this and other pathological settings remains to be investigated.

Another important open question concerns the cause of the increased permeability and focal hemorrhaging seen in *Nodal*[iPCKO] brains, which we have not resolved in this study. As TGFβ family receptors are expressed by multiple cell types, it is possible that the

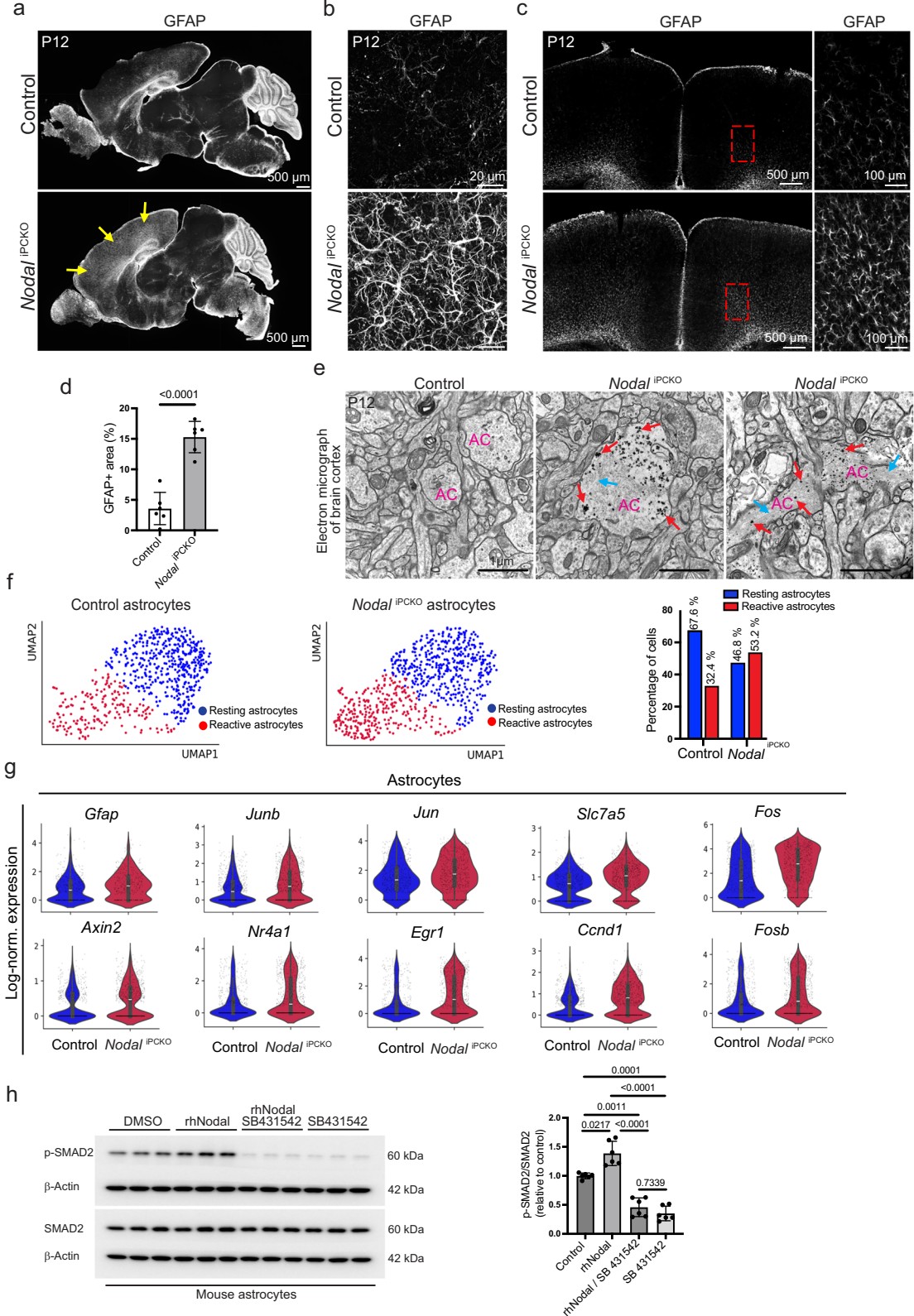

loss of Nodal induces changes in the endothelium. In fact, previous work has established that TGFβ signaling regulates vascular growth and barrier function especially in the central nervous system[116–118]. Notably, EC-specific Smad4 mutants show pronounced intracranial hemorrhaging and BBB breakdown, accompanied by reduced mural cell coverage[118]. Likewise, EC-specific loss of the TGFβ receptors TGFBR1 and/or TGFBR2 was shown to impair retinal vascularization,

increase vascular permeability and trigger the infiltration of immune cells[119]. Pericyte-derived TGFβ1 was also shown to increase barrier properties of immortalized mouse brain capillary endothelial cells (MBEC4) in co-culture with rat brain pericytes in vitro[120].

Taken together, it is clear that TGFβ family ligands and their receptors control fundamental aspects of vascular growth and integrity, which is likely to involve complex interactions between multiple

**Fig. 5 | Nodal regulates astrocyte reactivity. a** Tile scan of sagittal sections through brains stained for GFAP (white). Yellow arrows indicate an increase in GFAP+ reactive astrocytes in *Nodal*[iPCKO] cortex relative to littermate control. **b** Higher magnification confocal images of GFAP+ astrocytes in *Nodal*[iPCKO] brain cortex relative to control. **c** Tile scan confocal view of P12 *Nodal*[iPCKO] and control coronal brain sections immunostained for GFAP (white). Panels on the right show higher magnification of areas in red boxes. **d** Quantitation of GFAP+ area in the *Nodal*[iPCKO] and littermate control brain cortex. Data represent mean ± s.e.m. (*n* = 6); *P*-value, unpaired two-tailed Student *t*-test. **e** Electron micrographs showing astrocytes of mouse brain cortex, reactive astrocytes in *Nodal*[iPCKO]. Red arrows indicate glycogen granules and blue arrows mark intermediate filaments in mutant astrocytes (AC). **f** UMAP projections showing reactive astrocytes (red dots) versus resting astrocytes (blue dots). Ratio of the two populations is displayed in a bar graph on the right. **g** Violin plots showing the upregulation of markers of reactive astrocytes in scRNA-seq samples from *Nodal*[iPCKO] relative to control (*n* = 635 control and 730 mutant astrocytes; two animals per group); Boxplots in the center of the violins show median (white bar), quartiles (box), and the farthest data point within 1.5 * inter-quartile range (whiskers). **h** Western blot analysis of SMAD2 and p-SMAD2 in lysates from cultured mouse primary astrocytes treated with rhNodal and SB431542 inhibitor, as indicated. Quantitation of p-SMAD2/SMAD2 ratio is shown in the graph on the right. Data represent mean ± s.e.m. (*n* = 6); *P*-values, Brown–Forsythe and Welch ANOVA with Dunnett post-hoc test. Source data are provided as a Source Data file.

cell types. As the morphogenesis of complex tissue structures is likely to be mediated by communication between several different cell types and through multiple signaling pathways, a limitation of our study is that we have focused on a few selected factors. Reciprocal interactions between different cell types might also explain why defects in our mutant models often involve a reduction of PDGFRβ+ pericytes, an observation that remains to be investigated in future studies. Like all genetic tools, the *Pdgfrb-CreERT2* line used in our study has some limitations, one of which is the targeting of vascular smooth muscle cells, which might also lead to phenotypic alterations. As vSMCs are mostly associated with larger caliber arteries and veins, it is not obvious how they would directly communicate with non-vascular cell types located in the vicinity of capillaries and thereby contribute to the phenotypic alterations reported in our study. In addition, *Hgf* expression is strongly enriched in lung pericytes but absent from vSMCs. Likewise, we find that Nodal expression is much higher in brain pericytes than in brain-derived vSMCs, arguing that the latter are less likely to act directly on astrocytes and microglial cells.

Despite these limitations, our study identifies pericytes as central regulators of tissue morphogenesis and microenvironmental homeostasis during postnatal development. We demonstrate that pericytes are vital sources of growth factors and reveal that their signaling is organ-specific, enabling tailored functional roles aligned with distinct morphogenetic processes and tissue-specific environments.

## Methods
### Mouse models
Mice used in this study were housed in the animal facility of the Max Planck Institute for Molecular Biomedicine, Münster. Animals were maintained in individually ventilated cages under controlled conditions (temperature: 22 ± 1.5 °C; humidity: 55 ± 5%) with a 14-h light/10-h dark cycle, and had ad libitum access to food and water. All animal experiments were performed according to the institutional guidelines and laws, approved by the local animal ethical committee and were conducted at the Max Planck Institute for Molecular Biomedicine with necessary permissions (Az 81-02.04.2020.A471) granted by the Landesamt für Natur, Umwelt und Verbraucherschutz (LANUV) of North Rhine-Westphalia, Germany. Animals were combined in groups for experiments irrespective of their sex.

C57BL/6J mice were used for the analysis of wild-type lung and brain. In vivo labeling of pericytes was performed by mating *Pdgfrb(BAC)-CreERT2* transgenic animals[12] (available as strain #029684 from The Jackson Laboratory) and *Rosa26*[mT/mG] reporter mice[24]. Cre activity was induced in pups resulting from this mating by intraperitoneal injection of pups at postnatal day 1 (P1) with a single dose of 50 µg 4-hydroxy tamoxifen (4-OHT) (H7904, Sigma) in ethanol-peanut oil (P52144, Sigma). For inducible genetic experiments employing a mural cell-specific loss-of-function approach, *Pdgfrb(BAC)-CreERT2* transgenics were interbred with mice carrying loxP-flanked alleles for *Bdnf* (*Bdnf*[lox/lox])[121], *Hgf* (*Hgf*[lox/lox])[122], or *Nodal* (*Nodal*[lox/lox])[123] in separate crosses. To inactivate *Ntrk2* in the postnatal endothelium, *Ntrk2*[lox/lox] mice[38] and *Cdh5(PAC)-CreERT2*[+/T] transgenic mice[124] were interbred. Cre activity was induced by three consecutive intraperitoneal injections of 50 µg tamoxifen (T5648, Sigma) in ethanol-peanut oil (P52144, Sigma) from P1 to P3.

### Lung sample preparation and immunohistochemistry
For immunohistochemical analysis of mouse lungs, pups at P21 were anaesthetized by intraperitoneal injection of xylazine (Bayer, Rompun 2%; 10 mg/kg) and ketamine (Zoetis, Ketavet 100 mg/ml; 100 mg/kg) dissolved in PBS. The chest cavity of each terminally anesthetized pup was opened to access the heart and lungs. A warm (37 °C) solution of 6% gelatin (G1890, Sigma) in PBS was gently perfused through the right ventricle using manual pressure. To allow the gelatin to solidify, an ice-cold tissue paper was placed over the exposed heart and lungs for 15 min. Subsequently, the ventral trachea was cannulated using an intravenous catheter (BD Insyte, 381212), which was secured with a suture. The lungs were then inflated to full capacity by gently injecting warm (37 °C) 1% low-gelling agarose (A4018, Sigma) in PBS. The agarose-inflated lungs were further chilled by placing an ice-cold tissue paper on them for 20 min. Afterward, the lungs were excised and placed in a 2% paraformaldehyde (PFA; Sigma, P6148) solution in PBS at 4 °C for 30 min. Following this initial fixation, the lungs were incubated in cold PBS for 30 min. After washing with cold PBS, the lung lobes were sliced into 150 µm sections using a vibrating blade microtome (VT1200, Leica) and then fixed in 4% PFA at 4 °C for 1 h. After the second fixation, the lung samples were washed thoroughly by incubating them twice in PBS for 30 min at room temperature (RT). Lung slices were subsequently blocked in a blocking solution composed of 5% donkey serum and 0.5% Triton X-100 in PBS for a minimum of 2 h at RT or overnight (O/N) at 4 °C. Following blocking, the vibratome sections were treated with primary antibodies diluted in the blocking solution overnight at 4 °C. The sections were washed once in 0.5% Triton X-100 in PBS (PBST) for 20 min and then three times in PBS for 10 min each at RT. After washing, the sections were incubated with secondary antibodies diluted in blocking solution for 2 h at RT or O/N at 4 °C. Nuclei were counterstained with DAPI (D9542, Sigma, 2 µg/ml). After four wash steps with PBS, the sections were mounted using FluoroMount-G (Southern Biotech, 0100-01) and covered with cover slips. The mounted samples were stored at 4 °C.

The following primary antibodies were used for lung staining: rat anti-RAGE (1:200, R&D Systems, MAB1179), rabbit anti-Aquaporin 5 (1:200, Millipore, 178615), goat anti CD31/PECAM1 (1:200, R&D Systems, AF3628), rat anti-PDGFRβ (1:100, eBioscience, 14-1402), goat anti-PDGFRβ (1:100, R&D Systems, AF1042), rabbit anti-Prosurfactant Protein C (1:200, Millipore, AB3786), chicken anti-GFP (1:300, 2BScientific Ltd., GFP-1010), mouse anti-αSMA-Cy3 (1:300, Sigma C6198), rat anti-DC-LAMP/CD208 (1:200, Novus Biologicals/Dendritics, DDX0192P-100), rabbit anti-NKX2.1/TTF1 (1:200, abcam, ab76013), rat anti-ICAM2/CD102 (1:100, BD Pharmingen, 553326), goat anti-HGFR/c-Met (1:100, R&D Systems, AF527), rabbit anti ERG (1:100, Abcam, ab110639), rabbit anti-Cleaved Caspase-3 (1:100, Cell Signaling, #9664), and Goat anti-TrkB (1:50, Biotechne; AF1494).

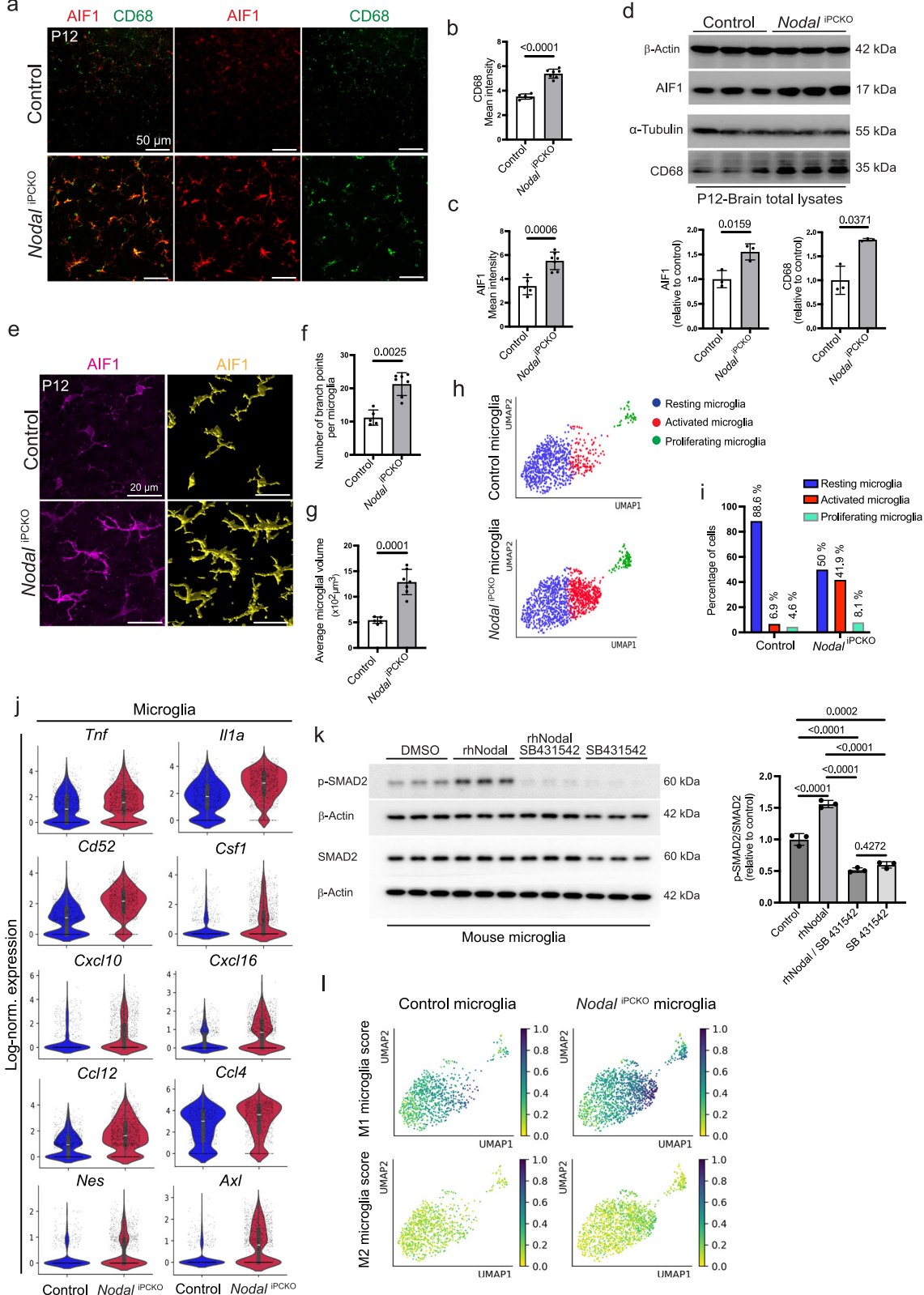

## Brain sample preparation and immunohistochemistry

For immunohistochemical analysis of mouse brains, pups at P12 were anaesthetized by intraperitoneal injection of xylazine (Bayer, Rompun 2%; 10 mg/kg) and ketamine (Zoetis, Ketavet 100 mg/ml; 100 mg/kg) dissolved in PBS. Following terminal anesthesia, the chest cavity was surgically opened to expose the heart. To clear the circulatory system of blood, a puncture was made in the right atrium, and 10 ml of ice-cold

PBS was perfused through the left ventricle using a peristaltic pump (Pump P-1, GE Healthcare). Tissue fixation was initiated immediately afterwards by perfusing 10 ml of ice-cold 1% paraformaldehyde (PFA; Sigma, P6148) through the same route. Once perfusion was complete, brains were carefully dissected from the skull and post-fixed by immersion in 4% PFA at 4 °C O/N. The fixed brains were subsequently washed four times (15 min each) in PBS. For sectioning, the brains were

**Fig. 6 | Microglia activation is controlled by pericyte-derived Nodal. a** P12 Brain cortex of *Nodal*[iPCKO] and control littermates stained for AIF1 (red) and CD68 (green). Graphs showing the increased expression of CD68 (**b**) and AIF1 (**c**) in *Nodal*[iPCKO] sections of the brain cortex compared to control littermate. Data represent mean ± s.e.m. (*n* = 7); *P*-value, unpaired two-tailed Student *t*-test. **d** Western blot analysis of AIF1 and CD68 proteins in brain cortex lysates from P12 *Nodal*[iPCKO] and control littermates. Molecular weight marker (kDa) is indicated. β-Actin and α-Tubulin are shown as loading controls. Quantitation of AIF1 and CD68 is presented in the graphs below. Data represent mean ± s.e.m. (*n* = 3); *P*-values, unpaired two-tailed Student *t*-test (AIF1) and unpaired two-tailed Welch's test (CD68). **e** Confocal images showing AIF1 (magenta) immunostaining of *Nodal*[iPCKO] and control brain cortex sections (left panels). Imaris surface-rendered image of AIF1+ microglia from the left panels shows the morphology of microglia and their cellular projections (yellow, right panels). Graphs showing the number of branch points per microglia (**f**) and the average microglial volume (**g**) in cortical sections from *Nodal*[iPCKO] and control brains. Data represent mean ± s.e.m. (*n* = 5 controls and 7 mutants);

*P*-values, unpaired two-tailed Mann–Whitney test in **f** and unpaired two-tailed Welch's test in (**g**). **h** UMAP projections showing activated microglia (red dots) versus resting microglia (blue dots) in *Nodal*[iPCKO] and control scRNA-seq data. **i** Graph showing the ratio of resting, activated and proliferating microglia in *Nodal*[iPCKO] and control brain. **j** Violin plots showing increased expression of markers of activated microglia in P12 *Nodal*[iPCKO] brain cortex relative to control (*n* = 830 control and 1346 mutant microglia; two animals per group); Boxplots in center of the violins show median (white bar), quartiles (box), and the farthest data point within 1.5 * inter-quartile range (whiskers). **k** Western blot analysis of total SMAD2 and p-SMAD2 in lysates from cultured mouse primary microglia treated with rhNodal and SB431542 inhibitor, as indicated. Molecular weight marker (kDa) is indicated. β-Actin is shown as a loading control. Graph on the right shows quantification of the p-SMAD2/SMAD2 ratio. Data represent mean ± s.e.m. (*n* = 3); *P*-values, one-way ANOVA with Tukey's test. **l** UMAP plots showing the gene set score (scanpy score_genes) for type 1 (M1) and 2 (M2) microglia in *Nodal*[iPCKO] versus control brain cortex. Source data are provided as a Source Data file.

cut either along the sagittal midline or into 2 mm-thick coronal sections using an acrylic brain matrix designed for mice (RBMA-200C, World Precision Instruments). The brain hemispheres or coronal sections were embedded in 4% low-gelling-temperature agarose (Sigma, A9414) dissolved in PBS at 40 °C. After embedding, the samples were rapidly cooled on ice to solidify the agarose. Once the agarose had solidified, the blocks were trimmed and mounted onto a specimen holder using cyanoacrylate adhesive (UHU GmbH & Co. KG). Sections with a thickness of 100 μm were then cut using a vibratome (VT 1200S, Leica). Vibratome sections were blocked and permeabilized O/N at 4 °C in a solution containing 1% bovine serum albumin (BSA; Sigma, P6148), 2% normal donkey serum (Abcam, ab7475), and 0.5% Triton-X-100 (Sigma, T8787) in PBS. Primary antibodies were diluted in freshly prepared blocking solution and incubated O/N at 4 °C. Following primary antibody incubation, the sections were washed once with 0.5% Triton-X-100 in PBS, followed by three washes with PBS (20 min each at 4 °C). The sections were then incubated overnight with species-specific Alexa Fluor-conjugated secondary antibodies (Invitrogen), diluted 1:500 in the blocking buffer. Nuclei were counterstained with 4′,6-diamidino-2-phenylindole (DAPI) (Sigma, D9542; 2 μg/ml). Following secondary antibody incubation for 2 h at RT or O/N at 4 °C, the sections were washed with PBS as described and mounted using Fluoromount G (Southern Biotech, 0100-01).

The following primary antibodies were used for brain immunostaining: rabbit anti-AQP4 (1:100, Sigma HPA014784), mouse anti-αSMA-Cy3 (1:200, Sigma C6198), goat anti CD31/PECAM1 (1:200, R&D Systems, AF3628), rabbit anti ERG (1:50, Abcam, ab110639), rabbit anti-GFAP (1:200, DAKO, Z0334), goat anti-GFAP (1:200, Novus Biologicals, NB100-53809), rat anti-CD68 (1:100, Abcam, ab53444), chicken anti-GFP (1:200, 2BScientific Ltd., GFP-1010), rabbit anti-GLUT1 (1:100, Millipore, 07-1401), goat anti-AIF1 (1:100, Novus Biologicals, NB100-1028), rat anti-ICAM2/CD102 (1:100, BD Pharmingen, 553326), rat anti-Nestin (1:200, Santa Cruz, sc101541), rat anti-PDGFRβ (1:100, eBioscience, 14-1402), goat anti-PDGFRβ (1:100, R&D Systems, AF1042), rat anti-TER-119 (1:200, R&D Systems, MAB1125), goat anti-Collagen IV (1:100, Millipore; AB769), Rabbit anti-SOX2 (1:100, Abcam, ab97959), goat anti-SOX9 (1:100, R&D Systems, AF3075), rat anti-Nestin (1:100, Cosmo Bio, BAM-73-100-EX), rabbit anti-FOSB (1:100, Cell Signaling, #2251), rabbit anti-FOS (1:100, Abcam, ab190289), goat anti-CXCL10 (1:100, Biotechne, AF466-SP), mouse anti-Nestin (1:100, Santa Cruz, sc-23927) and Isolectin B4 Alexa Fluor-488 (1:50, Invitrogen; I21411).

The following donkey-raised secondary antibodies (all in 1:500 dilution) were used for immunostaining of brain and lung samples: anti-rabbit IgG conjugated to Alexa Fluor (AF) 488 (Thermo Fisher Scientific, A21206), anti-chicken IgY AF488 (Jackson ImmunoResearch, 703-545-155), anti-rat IgG AF488 (Thermo Fisher Scientific, A21208),

anti-goat IgG AF488 (Invitrogen, A-11055), anti-mouse IgG AF546 (Thermo Fisher Scientific, A10036), anti-rat IgG AF594 (Thermo Fisher, A21209), anti-rabbit IgG AF594 (Thermo Fisher Scientific, A21207), anti-goat IgG AF594 (Thermo Fisher Scientific, A-11058), anti-rabbit IgG AF647 (Thermo Fisher Scientific, A-31573), anti-rat IgG AF647 (Jackson ImmunoResearch, 712-605-153), anti-goat IgG AF647 (Thermo Fisher Scientific, A-21447), and anti-mouse IgG AF647 (Thermo Fisher Scientific, A-31571). Nuclei were counterstained with DAPI (1 μg/ml) together with secondary antibodies.

## Cell culture

Mouse brain endothelial cells (b.End3, ATCC, cat. #CRL-2299) were cultured in DMEM (Sigma, D6546) supplemented with penicillin/streptomycin (PAA, P11-010) and 10% FCS, and kept in a humidified incubator at 37 °C, 10% $CO_2$. Cells were seeded into six-well plates coated with 0.1% gelatin for protein extraction. or into μ-Slide 24-well (Ibidi, 82426) for immunostaining.

Mouse C57 mixed astrocytes (Lonza, M-AsM-330) were cultured in Astrocyte Growth Medium BulletKit™ (AGM™ BulletKit™, CC-3186) and kept in a humidified incubator at 37 °C, 5% $CO_2$. Cells were seeded into six-well plates coated with poly-L-lysin (2 μg/cm², Sciencell, 0403) for protein extraction, or into μ-Slide 24-well (Ibidi, 82426) for immunostaining.

Mouse microglia (Sciencell, M1900) were cultured in Microglia Medium (Sciencell, #1901), which consists of 500 ml of basal medium, supplemented with 25 ml of fetal bovine serum (FBS, Cat. No. 0025), 5 ml of microglia growth supplement (MGS, Cat. No. 1952) and 5 ml of antibiotic solution (P/S, Cat. No. 0503). Microglia were seeded into six-well plates coated with poly-L-lysin (2 μg/cm², Sciencell, 0403) for protein extraction, or into μ-Slide 24-well (Ibidi, 82426) for immunostaining, kept in a humidified incubator at 37 °C, 5% $CO_2$.

## Stimulation and inhibitor treatment of cultured cells

bEnd.3 cells in a six-well plate were starved in basal medium for 1 h at 37 °C, then treated with basal medium containing Nodal (100 or 200 ng/ml, R&D Systems, 3218-ND-025) for 30 min, with or without the SB431542 inhibitor (10 μM, Selleckchem, S1067). Following stimulation, cells were processed for protein isolation. For immunostaining, cells were seeded in μ-Slide 24-well plates (Ibidi, 82426) and treated under the same conditions for 16 h.

Mouse astrocytes and microglia, cultured separately in six-well plates at 37 °C, were treated with Nodal (200 ng/ml) or SB431542 inhibitor (10 μM) for 30 min. Inhibitor treatment was also performed simultaneously with Nodal stimulation. After treatment, the cells were harvested for protein isolation. DMSO was used as a control treatment.

For immunostaining, cells were seeded in μ-Slide 24-well plates (Ibidi, 82426) and treated under the same conditions for 16 h.

a

Pulmonary pericyte-derived HGF and BDNF

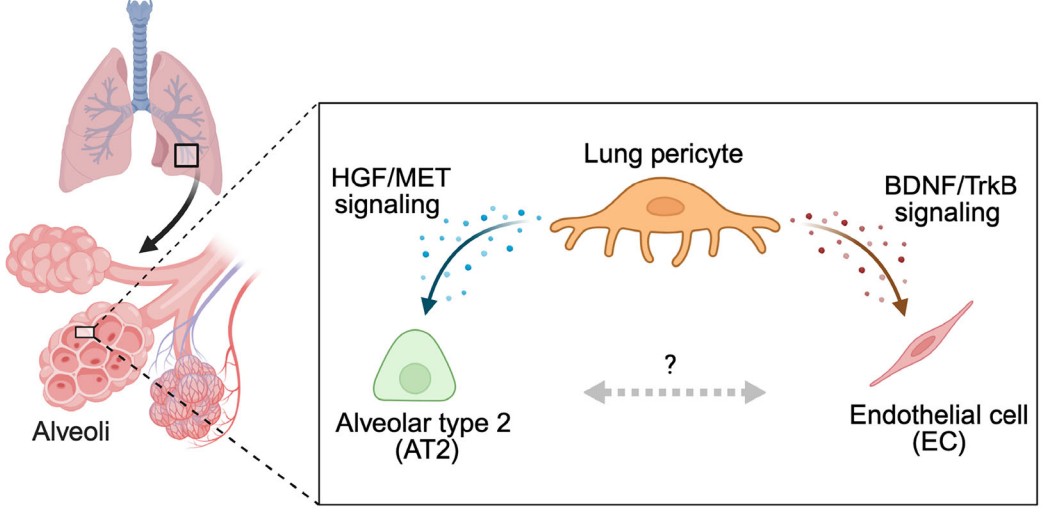

b

Brain pericyte-derived Nodal

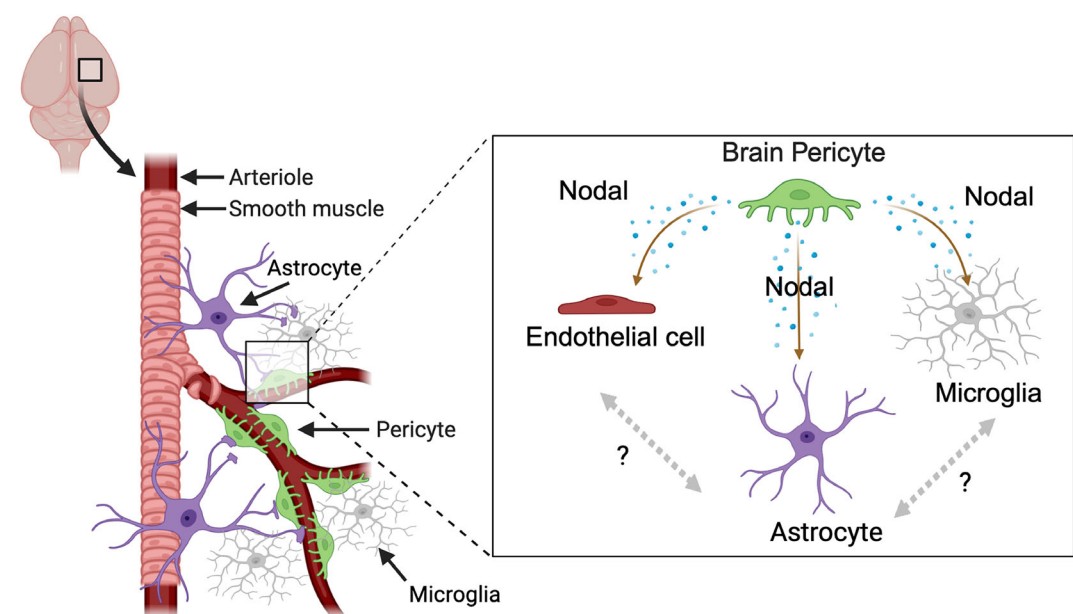

**Fig. 7 | Summary of findings.** Schematic representation of the modulation of tissue microenvironments in the lung (**a**) and brain (**b**) by pericyte-derived factors. Created in BioRender. Rasouli, S. J. (2026) https://BioRender.com/pf1pwy6, licensed under CC BY 4.0.

## Protein isolation and Western blotting

For immunoblotting, cells were washed twice with ice-cold PBS containing 1 mM PMSF, then lysed on ice in a lysis buffer (20 mM Tris–HCl, pH 8.0, 150 mM NaCl, 0.5% Triton X-100, 0.1% SDS, 0.1% sodium deoxycholate, 2 mM EDTA) supplemented with Halt Protease Inhibitor Cocktail (Thermo Scientific, 78429) and Phosphatase Inhibitor Cocktail Set V (EMD Millipore, 524629). The lysates were incubated for 20 min at 4 °C. Following vortexing, cell lysates were centrifuged for 10 min at 4 °C, and protein concentrations in the supernatants were measured using the BCA Protein Assay Kit (Pierce, 23225). Lysates were combined with 2× sample loading buffer in a 1:1 ratio and heated at 95 °C for 5 min.

2 µg of total proteins were then separated by SDS–polyacrylamide gel electrophoresis (SDS–PAGE) and transferred to Immobilon-P polyvinylidene fluoride (PVDF) membranes, which were briefly treated with 100% methanol for 20 s. Membranes were blocked in either 1–4% BSA/TBST or 0.3% skim milk/TBST for 1 h before being incubated with primary antibodies in blocking buffer O/N at 4 °C with gentle agitation. After three washes with TBST, membranes were exposed to peroxidase-conjugated secondary antibodies diluted in either 1% BSA/TBST or 0.3% skim milk/TBST buffer for 1 h at room temperature with gentle agitation. The membranes were then washed and developed using the ECL Prime detection kit (GE Healthcare, RPN2232).

For tissue protein isolation, mouse brain cortices and lungs were perfused with cold PBS containing PhosSTOP (Roche) through the left and right ventricles, respectively. The perfused tissues were dissected before snap freezing in liquid nitrogen. Tissues were then homogenized in lysis buffer using a Pestle (Argos) and centrifuged for 20 min at 4 °C to clarify the lysates. Total protein concentrations were determined with the BCA Protein Assay Kit (Pierce, 23225). 20 μg of total protein from the lysates was separated by SDS–PAGE and transferred to PVDF membranes. After blocking, the membranes were incubated with primary antibodies, followed by washing and detection using horseradish peroxidase-conjugated secondary antibodies and the ECL Prime detection kit.

The following antibodies were used for immunoblotting: mouse anti-β-ACTN (1:6000, Invitrogen, AM4302), mouse anti-α-Tubulin (1:6000, Sigma, T5168), rabbit monoclonal anti-SMAD2 (1:5000), Cell Signaling, 5339), rabbit monoclonal anti-Phospho-SMAD2 (1:500, Cell Signaling, 3108), goat anti-AIF1 (1:5000, Novus Biologicals, NB100-1028), rat anti-CD68 (1:3000, Abcam, ab53444), goat anti-Rabbit IgG, HRP-linked whole Ab (1:5000, Cell Signaling, 7074), sheep anti-Mouse IgG, HRP-linked whole Ab (1:40,000, HG-Healthcare, NA931), Peroxidase AffiniPure Bovine anti-Goat IgG (H + L) (1:20,000, Jackson ImmunoResearch, 805-035-180) and goat anti-rat HRP-linked whole Ab (1:20,000, Amersham, NA935).

## Cell immunostaining

Cells were cultured in their respective media within μ-Slide 24-well plates (Ibidi, 82426). The cells were fixed using 4% PFA for 10 min, followed by permeabilization with ice-cold 0.1% Triton X-100 in PBS for 5 min at 4 °C. After washing with PBS, cells were blocked with a solution containing 4% donkey serum and 2% BSA in PBS for 1 h at RT. Following blocking, the cells were incubated with primary antibodies diluted in the blocking buffer for 1 h at RT. After another washing step, secondary antibodies, also diluted in the blocking buffer, were applied for an additional hour. Finally, after washing steps with PBS, 250 μl of Fluoromount-G was added to each well of the μ-Slide for mounting.

## EdU incorporation assay in vivo and in vitro

EC proliferation in the brain and lungs was evaluated by administering a single intraperitoneal injection of EdU (150 μg/50 μl, Invitrogen, A10044 to P12 pups or 250 μg/100 μl to P21 pups). After 2 h, the brains or lungs were dissected and processed as described above. EdU-positive cells were visualized using the Click-iT EdU Imaging Kit (Thermo Fisher Scientific, C10340).

To assess bEnd.3 proliferation in culture, cells were treated with Nodal and SB431542 inhibitor as indicated, and were incubated with EdU (10 μM) in an old medium for 60 min at 37 °C. EdU-positive cells were visualized using the Click-iT EdU Imaging Kit (Thermo Fisher Scientific, C10340).

## Scratch wound assay

A scratch wound assay was performed on cultured bEnd.3 cells. Scratches were created using a 10 μl pipette tip positioned perpendicular to the bottom of the well near the well wall. The pipette tip was gently dragged across the well under light pressure to generate uniform scratches. Following scratch formation, the culture medium was replaced with low-serum medium (2% FBS) supplemented with either Nodal (200 ng/ml) or a combination of Nodal (200 ng/ml) and SB431542 inhibitor (10 μM). In case of the latter, DMSO was added as a vehicle control. Tile-scan bright-field images of the scratch areas were captured using a Zeiss AxioObserver Z1. Image analysis was conducted using ZEN Blue software and Fiji.

## Dextran injection

A 5 mg/ml solution of 70 kDa Dextran (Texas Red conjugate, lysin fixable, Thermofisher, D1864) was prepared in sterile PBS. Pups were anesthetized before administering 50 μl of the dextran solution into the bloodstream via retro-orbital injection.

## Image acquisition and analysis

Confocal image acquisition was performed using a Zeiss confocal microscope LSM780 and LSM880 equipped with the following objective lenses: ×10 plan apochromat (APO), numerical aperture (NA) 0.45, ×20 Plan APO NA 0.8, water immersion ×40 LD C-APO NA 1.20, and oil immersion ×63 Plan APO NA 1.40. The confocal data were then processed with the ZEN 2.3 SP1 FP3 software (black edition), Fiji and Imaris.

Astrocyte endfeet coverage in the brain was calculated by dividing the surface area positive for Aqp4 immunostaining by the total vascular area. Vessel density was quantified with Imaris from 3D surface-rendered reconstructions of ICAM2+ surfaces, using an average of six fields (213 × 213 × 20 μm) per lung sample, or an average of four fields (213 × 213 × 30 μm) per brain sample. Airspace volume was measured using Imaris from 3D surface-rendered reconstructions of RAGE+ areas, using an average of six fields (213 × 213 × 30 μm) per lung sample. Arterial oxygen saturation (SpO$_2$) and respiratory rate were monitored using the MouseOx Plus (Starr Life Sciences, USA).

Nuclear enrichment of the protein of interest was measured by identifying DAPI-stained nuclei and segmenting them as regions of interest (ROIs). The mean fluorescence intensity of the antibody of interest was subsequently quantified within these ROIs. Detailed information on the number of biological replicates (animals or independent experiments) is provided in the Source Data File. All images shown are representative of the respective staining obtained from multiple experiments, with identical laser excitation and confocal detection parameters maintained within each experiment.

## Transmission electron microscopy

For electron microscopy, pups at P12 were anaesthetized by intraperitoneal injection of xylazine (Bayer, Rompun 2%; 10 mg/kg) and ketamine (Zoetis, Ketavet 100 mg/ml; 100 mg/kg) dissolved in PBS, and transcardially perfused with 10 ml of PBS and 40 ml of 2% PFA and 2% glutaraldehyde in 0.1 M cacodylate buffer (pH 7.2). The brain was removed and further fixed by immersion in the above-mentioned solution for 3 h at RT. An acrylic matrix for mouse brains (World Precision Instruments Cat. No. RBMA-200C) was used to section the brain in coronal slices from which smaller pieces belonging to the cortex were collected and further fixed in reduced 1% osmium tetroxide containing 1.5% potassium hexacyanoferrate. Next, the tissue was dehydrated and embedded in epon. Ultrathin 60 nm sections were cut on an ultramicrotome (Leica UC6) and counterstained with uranyl and lead. Images were taken using an electron microscope (Tecnai 12 Biotwin TEM, FEI) and representative pictures were documented on a 2K CCD camera (Veleta, EMSIS Münster).

## Single-cell RNA-sequencing

For the single cell sequencing of postnatal lung, two Hgf$^{lox/lox}$ control mice and two Hgf$^{iPCKO}$ P21 old pups or two Bdnf$^{lox/lox}$ control mice and two Bdnf$^{iPCKO}$ P21 old pups were analyzed, respectively. Lung lobes were dissected, finely minced using sharp scissors and transferred into enzymatic digestion solution containing 40 U/ml collagenase I (Gibco, 170100-017), 40 U/ml Collagenase IV (Gibco, 170100-019), 4U/ml Dispase II (Thermo Fisher, 17105041) and 2 U/μl DNase I (Worthington, LK003170) prepared in DMEM (Gibco, 31053028) medium. Digestion was performed in a water bath at 37 °C for 30 min. During this time, the tissue was also mechanically dissociated by passing it through a needle-syringe several times. Enzymatic digestion was stopped by addition of FACS buffer (2% FBS in DMEM) and the lysate was passed through 70 μm cell strainer (Falcon, 352350). The filtrate was centrifuged for 5 min at 4 °C, and the pellet was resuspended in RBC lysis

solution (eBioscience, 00-4333-57) and further incubated for 10 min on ice. Post resuspension in FACS buffer, the cell lysis was passed through a 50 μm Cell Trics filter (Sysmex, 04-0042-2317). Cells were centrifuged for 5 min at 4 °C, and the pellet was resuspended in cold MACS depletion buffer (autoMACS Running Buffer-MACS Separation Buffer, Miltenyi Biotec, 130-091-221). To proceed with magnetic separation and depletion of CD45+ cells and erythrocytes from the cell suspension, CD45 microbeads (Miltenyi Biotec, 130-052-301) and Ter119 microbeads (Miltenyi Biotec, 130-049-901) were added to the cells resuspended in MACS depletion buffer and incubated for 20 min at 4 °C. After washing, the cell solutions were passed through the MACS MS column (Miltenyi Biotec, 130-042-201) attached to a MACS separator placed on a magnetic Multistand. Filtrate solution consisting of CD45-/Ter119- cells was collected, and the cell number was counted using Luna Automated Cell Counter (L10001). For each mouse genotype, 40,000 cells were prepared for further sequencing from the two mouse littermates, using an equal number of cells per sample. Cells were loaded into a BD Rhapsody Cartridge (BD Bioscience, 633733) and captured on the BD Rhapsody Express Single-Cell Analysis System (BD Bioscience). Single-cell mRNA whole transcriptome (WTA) libraries were created using the BD Rhapsody Whole Transcriptome Analysis (WTA) Amplification Kit (BD Bioscience, 633801) and DNA sequencing was performed on a NextSeq500 (Illumina) using 2 ×75 bp paired end reads with an 8 bp single index.

For the single cell sequencing of postnatal cerebral cortex, two P12 *Nodal* [lox/lox] control and two *Nodal* [iPCKO] mice were anesthetized by intraperitoneal injection of Xylazine (16 mg/kg) and Ketamine (100 mg/kg) and transcardially perfused with 10 ml of ice-cold PBS supplemented with Heparin (25 U/ml). Brains were collected and the meningeal layers removed. Next, the cerebral cortex of each hemisphere was dissected and transferred to ice-cold DMEM supplemented with Penicillin/Streptomycin, Glutamine (GlutaMAX, Gibco) and 25 mM HEPES (PAA, Cat. No. S11-001), hereafter dissection media. The cortices belonging to mice from the same genotype were pooled together and minced using scalpels. The resulting paste was resuspended in 1 ml of enzyme blend containing Papain (25 U/ml, Worthington Cat. No. LK003176), DNAseI (113 U/ml, Worthington Cat. No. LK003170), and Liberase DH (100 μg/ml, Roche Cat. No. 5401054) dissolved in pre-warmed dissection media and incubated 30 min at 37 °C. During this time, the tissue was further homogenized by repeatedly pipetting the solution up and down every 10 min using filtered 1 ml tips. After the first 15 min of incubation, 0.5 ml of the described enzyme blend was further added. Next, the tissue homogenate was filtered through a 70 μm nylon mesh into a 50 ml Falcon tube, the filter was washed with 1 ml of pre-warmed dissection media and the final volume obtained was measured. In order to remove debris and myelin, the cell suspension was mixed with 1.7× volume 22% BSA (Carl Roth Cat. No. 8076.2) dissolved in PBS and centrifuged at 1000×*g* for 12 min and RT. The supernatant was aspirated and the remaining cell pellet resuspended in 1 ml of the described enzyme blend for a final 20 min incubation step at 37 °C with pipetting every 5 min. The resulting single-cell suspension was filtered through a 40 μm cell strainer and diluted by the addition of 8 ml of pre-warmed dissection media. After centrifugation at 300×*g* for 5 min, the supernatant was discarded and the cell pellet was resuspended in 1 ml of Red Blood Cell Lysis Buffer (Sigma Cat. No. R7757). After a 1 min incubation at RT, 20 ml of ice-cold PBS supplemented with 2% fetal calf serum was added and the whole suspension was centrifuged at 300×*g* for 5 min at 4 °C. Next, the supernatant was discarded and the cell pellet resuspended in 1 ml of filter-sterilized endothelial cell buffer (15 mM HEPES, 153 mM NaCl, 5.6 mM KCl, 1.7 mM CaCl$_2$, 1.2 mM MgCl$_2$ and 10% BSA, pH 7.4)[125]. Cell concentration was assessed using an automated cell counter and $4 \times 10^4$ cells from each genotype were loaded into a BD Rhapsody Cartridge (BD Biosciences Cat. No. 633733) for cell capture.

## scRNA-seq data analysis

**Preprocessing.** We used STAR version 2.7.10a[126] to generate a reference genome index for GRCm39, with Gencode annotations vM33. FASTQ reads were mapped against the reference genome index using STAR with the settings " --soloType CB_UMI_Complex --soloUMIlen 8 --soloCellFilter None --outSAMtype BAM SortedByCoordinate --soloFeatures Gene --runRNGseed 1 --soloMultiMappers EM --readFilesCommand zcat --outSAMattributes NH HI AS nM NM MD jM jI MC ch CB UB GX GN sS CR CY UR UY". Libraries using standard BD Rhapsody beads were mapped using the adapter parameters "--soloAdapterSequence NNNNNNNNNACTGGCCTGC-GANNNNNNNNNGGTAGCGGTGACA --soloCBposition 2_0_2_8 2_21_2_29 3_1_3_9 --soloUMIposition 3_10_3_17 --soloCBwhitelist BD_CLS1.txt BD_CLS2.txt BD_CLS3.txt", libraries with BD Rhapsody enhanced beads with --soloAdapterSequence NNNNNNNNNGTGANNNNNNNNNNGACA --soloCBposition 2_0_2_8 2_13_2_21 3_1_3_9 --soloUMIposition 3_10_3_17 --soloCBwhitelist BD_CLS1_v2_draft.txt BD_CLS2_v2_draft.txt BD_CLS3_v2_draft.txt".

Raw counts were imported as AnnData[127] objects. We removed low complexity barcodes with the knee plot method, and further filtered out cells with a high mitochondrial mRNA content, as well as unusually high total and gene counts, using manually determined cutoffs for each sample. Doublets were scored with Scrublet 0.2.3[128]. Finally, each sample's gene expression matrix was normalized using scran (1.22.1)[129] with Leiden clustering[130] input at resolution 0.5.

G2M and S phase scores were assigned to each cell using gene lists[131] and the scanpy (1.9.6[132]) sc.tl.score_genes_cell_cycle function.

**Embedding, clustering, and annotation.** The normalized expression matrix was subset to the 3000 most highly variable genes (HVG, sc.pp.highly_variable_genes, flavor "seurat"). For some analyses, expression values were cell-cycle regressed using scanpy.pp.regress_out on G2M and S-phase scores. The top 100 principal components (PCs) were calculated and batch-corrected using Harmony (0.0.5)[133]. The PCs served as a basis for *k*-nearest neighbor calculation (sc.pp.neighbors, n_neighbors=30), which were used as input for UMAP[134] layout (sc.tl.umap, min_dist = 0.3). Cells were clustered using scanpy.tl.leiden, and a suitable resolution was chosen in each sample for the main cell type annotation. Cluster marker genes were calculated using a pseudobulk approach, comparing aggregate counts with 2 pseudoreplicates for each cluster to all remaining cells (pyDeSEQ2 0.4.8). Finally, expression of select marker genes was plotted using Matplotlib (3.8.4)[135] "imshow", and clusters were annotated accordingly.

Sample and cell-type-specific subsets were subclustered using the top 2000 HVGs and 30PCs. Clusters were annotated at suitable Leiden resolutions using known and calculated cell-type markers.

**Differential expression analysis.** Differentially expressed genes were calculated using a pseudobulk approach, comparing aggregate counts with 2 pseudoreplicates for each condition (pyDeSEQ2 0.4.8).

**Comparison to lung P14 pericytes.** We downloaded raw sequencing data and cell annotations by Hurskainen et al. [36] from GEO (GSE151974) and processed them analogously to the datasets generated in this study, except for using STAR settings appropriate for 10X Chromium v3 ("–soloType CB_UMI_Simple –soloCBlen 16 –soloUMIstart 17 –soloUMIlen 12 –soloCBwhitelist 10xv3_whitelist.txt"). Data were integrated with data from this study using the Harmony batch-correction method. Differential expression followed the pseudobulk approach outlined above.

**Ligand–receptor signaling analysis.** We used CellphoneDB (v5.0.0)[136] with default parameters to analyze ligand–receptor interactions. Mouse gene names were converted to human gene names prior to the

analysis. For analysis of the *Nodal* [iPCKO] dataset, we manually included interactions of Nodal with Acvr1b, Acvr2a, and Acvr2b, based on interactions reported in STRING.

## Statistical analysis and reproducibility

Sample size was not determined using statistical calculations; instead, it was based on prior experience and the reproducibility of results across multiple independent experiments. In all reported quantifications, "*n*" represents biological replicates. For in vitro experiments, each biological replicate corresponds to independently treated and processed cell culture samples. For in vivo experiments, each biological replicate corresponds to an individual animal (i.e., mouse pups). For every experiment, the number of animals per group (*n*) included individuals from at least two different litters. For scRNA-seq experiments, lung or brain cells were pooled from two different animals per group. No animals or data were excluded from the analysis. In vivo samples used in the study were randomized in the sense that they were collected from different litters on different days, and the downstream experiments were performed for different batches at different time points. Biological replicates of in vitro experiments were randomized as they were performed independently in different batches at different time points. The investigators were not blinded to allocation during experiments and outcome assessment. Unless otherwise specified, the results represent data from three or more independent biological experiments to ensure reproducibility. Data were processed with the GraphPad Prism 10 software (version 10.6.1). Distribution analysis was performed using the D'Agostino–Pearson, Anderson–Darling, Shapiro–Wilk, or Kolmogorov–Smirnov normality tests, selected according to the sample size. Based on these results, two-tailed non-parametric tests were applied for comparisons between two groups. For normally distributed data with equal variances, an unpaired two-tailed Student's *t*-test was performed, whereas Welch's *t*-test was used when variances were unequal. The Mann–Whitney test was applied when at least one group did not meet the assumption of normality. Comparisons among multiple groups were performed using either one-way ANOVA or the Brown–Forsythe and Welch ANOVA, followed by an appropriate post hoc test for multiple comparisons. Effect sizes were calculated as eta squared for unpaired and Welch's *t*-tests, *R*-squared for ANOVA (both one-way or Browne–Forsythe and Welch), and rank-biserial correlation (*r*) for Mann–Whitney tests. The source data used for quantitative analyses, along with detailed statistical results including effect sizes and degrees of freedom, are provided in the Source Data file. Values are presented as mean ± standard error of the mean (s.e.m.). Exact *P* values are provided on all Figures and Supplementary Figs., with the exception of *P* values lower than 0.0001 (shown as <0.0001). In all cases, statistical significance was assessed with a 95% confidence interval; therefore, a *p*-value < 0.05 was considered significant. The source data used for quantitative analysis, as well as detailed statistical results, are provided in the Source Data file.

## Reporting summary

Further information on research design is available in the Nature Portfolio Reporting Summary linked to this article.

## Data availability

The scRNA-seq data generated in this study have been deposited in the Gene Expression Omnibus under accession number GSE285933. The mouse reference genome GRCm39 with GENCODE M26 annotation (https://www.gencodegenes.org/mouse/release_M26.html) was used for mapping the reads in this study. All other relevant data supporting the key findings of this study are available within the article and its Supplementary Information files. All measurements used for quantification are provided in the Source Data file along with the results of the different statistical analyses performed. Source data are provided with this paper. All individual mouse lines used in this study are commercially available at The Jackson Laboratory or through the lead author. All other biological materials described in this article are available through commercial suppliers as indicated. Source data are provided with this paper.

## Code availability

Custom code for scRNA-seq analysis, based on existing packages and own contributions, is publicly available at https://keeper.mpdl.mpg.de/d/301c44f11ccb455ab199/. Dependencies "scrna-tools" and "anndataview" can be found at https://github.com/Bioinformatics-Service-MPI-Munster.

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

## Acknowledgements
The authors thank Dr. Stefan Volkery and Malte Stasch for expert advice with confocal microscopy; Dr. Dagmar Zeuschner and Karina Mildner for electron microscopy; Dr. Hyun-Woo Jeong and Katja Mueller for help with RNA-seq experiments; Dr. Rüdiger Klein for generously providing *Ntrk2* conditional mutants; Dr. Dietmar Vestweber for kindly supplying bEnd.3 cells; Dr. Emma Watson for the help with the Dextran injection. This work was supported by the Max Planck Society (R.H.A.) and the German Research Foundation (CRC 1366, project no. 394046768, S.J. R. and R.H.A.).

## Author contributions
S.J.R. and R.H.A. designed experiments and interpreted results. S.J.R. performed the vast majority of the experiments. S.J.R. and R.D.-H performed isolation of brain cells for the single-cell sequencing and prepared samples for transmission electron microscopy. S.J.R., A.A., and M.E.P. performed the lung cell isolation for the single-cell sequencing. S.J.R. and K.K. performed the bioinformatic transcriptomic analysis. S.J.R and P.G. conducted the migration assay and EdU incorporation assay using bEnd.3 cells. S.J.R. and R.H.A. wrote the manuscript.

## Funding

## Competing interests
The authors declare no competing interests.
