## [Transparent Peer Review file · Nature Communications]

Pericytes are organ-specific regulators of tissue morphogenesis

Corresponding Author: Professor Ralf Adams

Version 0:

Reviewer comments:

Reviewer #1

(Remarks to the Author)

Research on angiocrine signaling has primarily focused on endothelial cells (ECs) and their subpopulations in organ growth and regeneration, while the role of pericytes as a source of secreted factors influencing neighboring cells has been relatively understudied. In this study, Rasouli et al. investigate the role of three pericyte-derived paracrine factors—HGF, BDNF, and Nodal—in postnatal lung and brain development. Using many versatile methods including multiple tissue-specific and inducible mouse genetics, high-resolution imaging, single-cell RNA sequencing, and in vitro assays, the authors explore the function of these pericyte-derived growth factors. They demonstrate that pericyte-derived hepatocyte growth factor (HGF) is essential for alveologenesis, while brain-derived neurotrophic factor (BDNF) regulates lung development by interacting with the TrkB receptor in pulmonary endothelial cells. Neither factor appears necessary for postnatal brain development.

Conversely, pericyte-derived Nodal, a TGF β family growth factor, plays a critical role in regulating blood vessel growth and barrier function in the postnatal brain but is not required for lung morphogenesis. These findings establish pericytes as key regulators of angiocrine signaling in an organ-specific manner. The manuscript is well-written and presents significant and critical insights into pericyte function during development. However, the findings are largely descriptive, lacking a deep mechanistic investigation. Additionally, the rationale for studying HGF, BDNF, and Nodal is unclear, as their effects appear unrelated and independent. Consequently, the overall significance of this work is diminished.

Major review

1. A major limitation of this study is the overlap between pericytes and SMCs due to the use of Pdgfrb-CreERT2, which does not inclusively label pericytes. How do the authors account for potential contamination from SMCs? In line 108, the manuscript states that “Pdgfrb-CreERT2-mediated recombination at P1–P3 does not label PDGFR α + fibroblasts, α SMA+ bronchial SMCs, or myofibroblasts”. But much data was generated after P12. What about SMC components after P12? In addition, the scRNA-seq data indicate that pericytes and SMCs cluster together as mural cells (Fig S1a,b). Additional markers should be incorporated to clearly distinguish pericytes from SMCs and characterize pericyte specific pathways.
2. The study focuses on pericyte-derived factors in murine models, but what is known about their role in human lung and brain development? Discussing the potential clinical implications of pericyte-derived HGF, BDNF, and Nodal in disease contexts would strengthen the impact of the findings.
3. The rationale for investigating HGF, BDNF, and Nodal is weak if the justification for selecting these three factors is only based on single-cell RNA sequencing data. What are their baseline expression levels in wild-type animal organs? In Fig. 1g, are there additional factors uniquely expressed in the lung or brain? If so, a complete list should be provided. The Discussion should address the contributions of other factors.
4. Is the baseline level of these three factors Hgf, Bdnf, and Nodal in a wild-type animal? In Fig 1g, are there other factors inclusively expressed in the lung or brain? Please provide the entire list. The contribution from other factors should be included in the Discussion. In each of three KO animals, can phenotypes be rescued by supplementing these factors back? What the correlation among these three factors? Do they regulate each other? What if double KO Hgf and BDNF?
5. Figures 2n, 4a, 4j, 5h, 6b require quantification. WB validation of TrkB and phosphorylated TrkB (p-TrkB) in Fig. 2n should be performed using isolated endothelial cells rather than whole lung lysates.
6. This study only studies pericytes from the brain and lung; thus, “organ-specific” is an overstatement; tone down the statement to be more brain and lung-specific, for the title and introduction.

7. The Discussion lacks a thorough examination of the study's limitations and directions for future research.
8. It would be beneficial to explore whether Hgf, Nodal, and Bdnf play similar roles in the adult lung. For example, when These animals are over P21 then admin tamoxifen.
9. Fig. 2c, 2g, 2h: The Pdgfr β staining in Fig. 2c suggests that Pdgfrb⁺ cells are reduced and exhibit morphological enlargement in knockout mice. In Fig. 2g–h, the total Erg⁺ endothelial cells are lower in Bdnf KO mice (40%) compared to controls (50%), but the Erg⁺ to Pdgfrb⁺ ratio remains the same. Does this suggest a reduction in Pdgfrb⁺ cells in KO mice? Clarification on whether Pdgfrb⁺ cell numbers are affected in each knockout model and whether their changes are proportional to ECs is necessary.
10. The authors conclude that pericyte-derived BDNF is critical for lung development but not for the brain. However, lung tissues were evaluated at P21, while brain tissues were examined at P12. Could this time difference affect the results? It would be valuable to assess brain vasculature at P21 for consistency.
11. Provide a stronger rationale for the inclusion of AIF1 and CD68 in the Fig 6 results section.

Minor

1. Please provide an explanation or full name of what iPCKO in Bdnf iPCKO in line 143 to help better understand.
2. The methodology for quantifying ICAM2-stained surface area should be described in Fig2q.
3. The first paragraph describes microglial morphology changes, while the second paragraph discusses activation based on chemokine expression from scRNA-seq data. Since the study does not directly analyze microglial morphology, the introduction should clarify the definition of activated microglia.
4. Are there references supporting the role of phosphorylated SMAD2 in brain microglial inflammation? If so, these should be cited.
5. Ensure proper formatting of mouse (italicized, lowercase) vs. human (uppercase) gene and protein names.

Reviewer #2

(Remarks to the Author)

Rasouli and co-workers performed an innovative study to investigate the organ-specific regulatory roles of pericytes in tissue morphogenesis. The authors investigated the functions of the pericyte-derived factors BDNF, HGF, and Nodal in the lungs and in the brain during postnatal development with a focus on elucidating signaling interactions between pericytes and other cell types. Overall, the notion that pericyte-expressed molecules regulate tissue morphogenesis is interesting and a highly timely topic. However, there are several issues that need to be addressed before this preliminary report will make a solid study. The authors should consider the following points to further develop their work:

1. The authors used Pdgfrb-CreERT2 mice to achieve pericyte-specific gene deletion. However, this Cre line presents a critical limitation. The Jackson Laboratory has pointed out that “hemizygous mice are smaller than their non-carrier siblings” (Strain #029684) indicating that the presence of the Pdgfrb-CreERT2 allele itself may induce developmental differences even prior to experimental manipulation. This inherent phenotype raises concerns that the observed phenotypes may be confounded by baseline abnormalities rather than genetic modification. To ensure the validity of their findings, the authors should confirm their results using, for example, PDGFR β -P2A-CreERT2 (The Jackson Laboratory, Strain #030201), which does not exhibit baseline developmental discrepancies.
2. Several conclusions were drawn with a lack of experimental validation resulting in unjustified overinterpretation of the results. The study proposes that pericyte-derived BDNF hampers lung development. However, the authors did not provide convincing data that directly evaluates changes in lung development. The authors should include specific analyses examining lung structure and function similar to what they previously performed (PMID: 29934496). Additionally, the authors state that “inactivation of the HGF gene impairs alveologenesis due to defective interaction with AT2 epithelial cells” (Abstract, line 39). However, the manuscript does not provide evidence that supports the instructive role of AT2 cells in this process. The authors should consider manipulating AT2 epithelial cells and evaluate whether this disruption specifically contributes to impaired alveologenesis. In line 328, the authors suggest that loss of pericyte-derived Nodal might induce reactive astrogliosis directly which could be enhanced by vascular leakage. However, this statement is descriptive and lacks experimental evidence investigating vascular leakage. Similarly, in line 367, the claim that “microglia can be directly activated by Nodal, with alterations in the brain microenvironment such as increased vascular permeability”, lacks examination of vascular permeability. The authors could perform Miles assays to evaluate vascular permeability. Furthermore, in line 235, the authors state that “the reduced vascularization could be secondary to the defects...”. Yet, no experimental evidence was presented in support of this hypothesis. The authors need to perform appropriate experiments to validate these assertions.
3. The manuscript contains several inconsistencies in both methodology and data analysis, which affect the interpretation of the results. The authors assess pericyte coverage using different analyses across different figures. Sup. Fig. 2d and Sup. Fig. 3a report pericyte coverage, whereas Fig. 2c, Fig. 3q, Sup. Fig. 5b, and Fig. 4d quantify the ratio of ECs (ERG⁺ cells) to pericytes (PDGFR β ⁺ cells). Additionally, cell proliferation is evaluated using different markers. Ki67 staining is used in Fig. 2e, Fig. 2p, and Sup. Fig. 5e, whereas EdU staining is employed in Fig. 4e and Sup. Fig. 2a. Furthermore, in Fig. 4e and Sup. Fig. 2a, EdU is co-stained with the endothelial cell marker ICAM2, whereas in Fig. 2e, Fig. 2p, and Sup. Fig. 5e, single staining of Ki67 is performed without specifying the cell type. The authors need to apply a consistent approach to evaluate proliferation and specify the proliferative cell type analyzed in all images. When quantifying vascular area, ICAM2 is predominantly used throughout the manuscript, but Sup. Fig. 3a and Sup. Fig. 7c use Glut1 staining. To enhance the consistency of the study, the authors should either standardize their methodologies and analyses across figures or provide a clear explanation for using different approaches.

4. In Fig.3b, the authors examine the role of pericyte-derived HGF in alveologenesi. However, the data presented are insufficient to fully support their claim. Given the authors' previous work in Nature Communications (PMID: 29934496), where alveolar development was assessed through parameters such as capillary formation and airspace volume, similar analyses should be conducted.
5. In Fig.5h, line 319, the authors claim that "pericyte-derived Nodal might directly regulate astrocyte behavior...". However, the experiment only measures the phosphorylation of astrocyte downstream signal transducer upon rhNodal treatment. It is recommended to perform additional experiments such as co-culture of pericyte and astrocyte, or conditioned medium treatment, to further examine the impact of pericyte-derived Nodal on astrocyte behavior.
6. Several images lack quantification. For example, Fig. 4a, Fig. 6a, Sup. Fig. 9b and Sup. Fig. 8b present images without corresponding statistical analysis. The authors need to include appropriate data analyses for these images to strengthen their conclusions.
7. The manuscript would benefit from further refinement of its language to arrive at more solid conclusions. In Sup. Fig. 7c, lines 296–300, the authors state that reactive GFAP+ astrocytes are present in both regions with and without vascular leakage. However, this conclusion does not deliver a clear message regarding the relationship between astrogliosis and vascular leakage. Similarly, when discussing markers associated with microglial activation in Fig. 6f and Sup. Fig. 10a, the rationale for separately mentioning Axl and Nes as "other markers of activated microglia" in line 345 is unclear. The authors should clarify why these two markers are specifically highlighted.
8. Despite performing several scRNA-seq experiments of postnatal lung and cerebral cortex, the manuscript does not make sufficient use of the scRNA-seq data. The authors could further explore the interaction networks between pericytes and other cell types by performing CellPhoneDB analyses on their scRNA-seq data, especially for specifying cell-cell communication mediated by receptor-ligand complexes
9. The study is performed solely with mouse models and in vitro assays. It is recommended to include the analyses of human data or mention the limitation of the study in the discussion.
10. In Sup. Fig. 1a and Sup. Fig. 4a, there are no enriched Met and Nodal transcripts in the heatmaps displaying top marker genes, as stated by the authors in line 213 and line 255. The authors should provide clarification or further analysis to support these claims.
11. In Sup. Fig. 3a, although Glut1 staining is performed, the analysis in Sup. Fig. 3b is "% of ICAM2-stained surface area".
12. The manuscript contains quite a few typographical errors. For example, "Nodel" should be "Nodal" on page 16, line 392. On page 9, line 226, it should be "Sftpb" instead of "Sftbp".
13. On page 14, line 334, the extra "that" should be removed.

Reviewer #3

(Remarks to the Author)

In the entitled manuscript Pericytes are organ-specific regulators of tissue morphogenesis by Rasouli et al., the authors characterize the role of three pericyte-derived factors, Hgf, BDNF, and Nodal, that are exclusively enriched in lung pericytes (Hgf & BDNF) or brain pericytes (Nodal). They provide a thorough analysis into how the Hgf and BDNF are crucial for lung development, most notably alveolar type 2 cells and the endothelium. Whereas pericyte expression of Nodal is important for blood vessel growth in the brain and loss of such results in reactive astrocytes and microglia independent of blood-brain barrier leakage. They demonstrate that pericyte-derived Hgf and BDNF did not alter brain vascular development where nodal expression from pericytes in the lung where not required for lung development. These were extremely thorough studies for each organ and pericyte-derived factor and therefore I have mostly minor comments that could help strengthen these studies.

- The discussion is limited on the knock-out of Nodal, Hgf, and BDNF from Pdgfrb-expressing meninges and perivascular fibroblasts. Is it possible that loss of these factors could affect meningeal development/homeostasis and/or large diameter vessels where perivascular fibroblasts reside?
- The same timeframe for tamoxifen exposure and therefore knocking out these genes was applied, however how does this relate to the stage in which the lung vs. the brain are in development or homeostasis? I think discussion of this needs to be expanded to identify the necessity of these pericyte-factors for either development or homeostasis. The authors sometimes state that these factors are required for homeostasis (perhaps to dampen inflammatory responses), however the use of proliferation and differentiation markers would suggest they are necessary for developmental programs.
- For BDNF they went into the downstream receptor, Ntrk2, and knocking that out in the endothelium to investigate a similar phenotype. While this is thorough work and maybe not necessary for all factors manipulated in this study, the discussion on the expression patterns/cell type specificity for their respective receptors felt thin.
- RAGE was used in figure 1 but was not explained until Figure 3k.

- Somewhat related, Figure 3b the authors state that loss of lung pericyte-Hgf impairs alveologenesis, however this is showing staining of RAGE, and not sure how this staining demonstrates alveologenesis besides the increase in fluorescent intensity in the HGF-iPCKO example. Especially since the authors then discuss that RAGE is an indication of inflammation for Figure 3k. But then overall RAGE is reduced in mutants?
- The organization of the manuscript threw me off a little bit, starting with lung pericyte-derived BDNF and then jumping to brain pericyte-derived Hgf and BDNF for the supplement in the narrative. Maybe there is another way to organize this?
- For the electron microscopy studies in Figure 4i, the endothelial protrusions are interesting, however their function/importance is not known, perhaps related to sensing or controlling of cerebral blood flow. Additionally, it is possible that there is a zonation specific feature for having more or less endothelial protrusions ie. Arterioles, arteriole-capillary transition, capillary, post-capillary venule, and venules. It would be more convincing if this data is quantified and the authors demonstrate that they were exclusively studying capillary vessels. Same goes for the perivascular space, some types of vessels have more or less perivascular spaces. So classification of the vessel type along with quantification of these features would greatly strengthen these data. Additionally, it is unclear what the red arrows are suppose to indicate, they are not pointing at tight junctions like the figure legend indicates. The white arrow in the right bottom image looks to be pointing at an astrocytic endfoot not a pericyte cell body or process. What are the green arrows pointing to?
- The authors looked at blood-brain barrier leakage but indicated this likely not a junctional problem, based only on EM data? So is it a transcytosis problem? Is there some way the authors can look at this in their electron micrographs or via IHC to explain how the BBB leakage is occurring?
- Authors state that microglia are highly ramified in the cortex of Nodal-iPCKO mice, however there is no quantification of this.

Reviewer #4

(Remarks to the Author)

R1: Review of "Pericytes are organ-specific regulators of tissue morphogenesis" by Rasouli et al.

The authors are investigating the vascular network, particularly molecular signals by pericytes controlling the behavior of other cell types during organ growth and patterning. Namely, the authors were choosing lung and brain as representative organs and explored the function of several pericyte-derived, potentially angiocrine-acting factors, namely brain-derived neurotrophic factor (BDNF), hepatocyte growth factor (HGF) and Nodal, a TGF β family protein. While the authors present some intriguing first results, these seem to be rather preliminary.

When the authors follow up on an interesting initial finding, such as the finding that pericyte-specific nodal deletion reduces brain vascularization, the subsequent analysis of astrocyte and microglia cell behavior stays rather superficial. An in depth analysis of pericyte-derived nodal effects on vascular network, developmental effects vis a activation of cells is missing. While some of the overall findings are interesting, this study rather describes initial findings that require an in depth analysis to be able to draw conclusions.

Major points:

1. The author used several conditional knockout mouse lines to investigate the function of pericyte-derived BDNF, HGF, and Nodal on postnatal brain and lung development. However, some characterizations of these mouse lines are missing. To confirm the specificity and efficiency of tamoxifen-induced depletion, the authors should investigate the reduction of interested factors (namely, BDNF, HGF and Nodal) in the pericytes of these mouse lines on the protein level (e.g., through IHC stainings or WB on FACS isolated cells). Also, the authors should utilize their single cell RNA sequencing data to reveal the alteration in cellular compositions (by UMAP or proportional bar graph) after BDNF, HGF and Nodal depletion.
2. In Figure 1, the author showed some differential regulated genes (e.g., Nodal, Hgf, Bdnf, Amgpt1,...etc.) in the lung and brain pericytes. However, the source of these single cell RNA sequencing data is not clearly identified. Are these data from their previous experiments or from a public source? In case of comparing their own data to the public sources, the author should note if there is age, isolation, and sequencing differences, as well as provide proper data integration control and analysis.
3. Figure 4: Please confirm effects of Nodal depletion on Smad phosphorylation on mouse tissue.
4. Figure 5 is revealing increased GFAP level in the cortex of Nodal depleted mice in pericytes. At this early timepoint of cortical development, astrogenesis could be influenced rather than astrocyte activation. Further marker of astrocytogenesis and/or astrocyte activation should be used to clarify the biological process. Furthermore, it should be clarified if the increased GFAP level are the consequence of pericyte-derived nodal depletion or of increased BBB opening. The cell culture experiments do not show any novelty, as it is known that TGFbeta family members activate the TGFbeta signaling pathway in astrocytes.
5. Figure 6 is revealing microglia activation in Nodal depleted mice in pericytes. As the Nodal depletion is leading to blood leakage into the brain, it is not surprising that microglia are activated. Again, it should be clarified if the microglia activation is due to pericyte-derived nodal depletion or of increased BBB opening.

6. Supplementary Fig. 8d. Why are no control cells GFAP positive? Is there a problem with the cell culture?

Version 1:

Reviewer comments:

Reviewer #1

(Remarks to the Author)

The authors have addressed my previous comments very thoroughly. I have one minor suggestion: a recent study (PMID: 39806101) indicates that HIGD1B may serve as a pericyte marker. Could the authors conduct an analysis of this marker to determine whether it would distinguish pericytes from smooth muscle cells in their dataset?

Reviewer #2

(Remarks to the Author)

The revised manuscript investigates the organ-specific roles of pericyte-derived factors in postnatal tissue morphogenesis, focusing on BDNF, HGF, and Nodal signaling in the lung and brain. The quality of the revised manuscript is very much improved. Still, several critical questions remain unanswered or insufficiently addressed. As a result, the authors' conclusions do not appear to be solidly supported by the presented data. In order to further develop their work, the authors need to address the following points:

1. The previous comments of this reviewer about Pdgfrb-CreERT2 lines have not been sufficiently addressed. The authors state that Pdgfrb-CreERTs does not induce similar lung or brain defects across different mutant backgrounds. However, this does not resolve the key concern raised in the original review that the Pdgfrb-CreERT allele itself causes baseline developmental abnormalities as stated specifically for this line by the Jackson Lab ("Important Note: As of August 2017, it has been the experience at The Jackson Laboratory Repository that when breeding hemizygous mice to wildtype siblings or C57BL/6J inbred mice, the resulting of their findings, the authors should validate key findings using an alternative Cre line that does not exhibit baseline developmental abnormalities such as PDGFR β litters have less hemizygous offspring than expected, which is 20% instead of the expected 50%. Hemizygous mice are smaller than non-carrier siblings."). Therefore, to ensure the validity -P2A-CreERT2 (The Jackson Laboratory, Strain #030201) and explicitly acknowledge the known developmental phenotype of Pdgfrb-CreERT2 mice and its potential impact on data interpretation in the manuscript.
2. The authors' response did not adequately address the issue of unjustified overinterpretation of the results. First, the statement in the abstract that "inactivation of the HGF gene impairs alveologenesis due to defective interaction with AT2 epithelial cells" has not been validated. The revised manuscript does not provide experimental evidence supporting an instructive role of AT2 cells in this process. Immunostaining of c-Met expression in AT2 cells only indicates receptor presence but does not demonstrate functional interaction. Although the authors cite published studies showing that AT2 cells specific c-Met deletion impairs alveologenesis (PMID: 17583691), this raises questions about the novelty of the current findings and these studies should also be included in the manuscript. To substantiate the proposed mechanism, direct manipulation of AT2 cells in the present study is still required. Therefore, the claim that HGF loss impairs alveologenesis through defective AT2 interaction should be revised or removed from the abstract unless experimentally validated. Second, the authors state that "the reduced vascularization could be secondary to the defects in alveolar epithelium observed in postnatal Hgf iPCKO lung" (revised line 273). However, simply demonstrating the absence of C-Met in endothelial cells does not address whether the vascular defect is secondary to epithelial abnormalities. To support this claim, experiments elucidating the temporal sequence of epithelial versus vascular defects are still required. Without such data, the conclusion that the vascularization defect is secondary remains unsupported.
3. The authors argue that co-culture of pericytes and astrocytes is not feasible and therefore rely on rhNodal treatment to study the effect of pericyte-derived Nodal on astrocytes. However, the treatment with recombinant protein does not recapitulate the effects of pericyte-derived Nodal and therefore cannot support the claim that "pericyte-derived Nodal might directly regulate astrocyte behavior" (revised line 368). To directly address this point, treating astrocytes with concentrated pericyte-conditioned medium would solve the culture-compatibility issues and provide more physiologically relevant evidence.
4. There is still no consensus as to the relationship between vascular leakage and astrogliosis. While the authors propose that "vascular leakage might contribute to astrogliosis," their newly added data of brains at P8 in supplementary Fig. 13 show that no vascular leakage is detected whereas GFAP⁺ astrocytes are already increased. This contradicts the proposed conclusion and indicates that astrogliosis can occur independently of barrier defects. Therefore, the authors' interpretation is not sufficiently supported by the data and requires additional experimental validation.

Reviewer #3

(Remarks to the Author)

The authors have made substantial improvements to the overall paper. They provided stronger evidence that manipulation of Hgf, BDNF, and Nodal in Pdgfrb-expressing pericytes is likely limited to pericytes in the lung or brain and not other Pdgfrb-expressing cell types. But they acknowledge this cannot be ruled out completely. The added data of ligand-receptor interactions is helpful and clarification of RAGE and the addition of other lung parameters strengthens their conclusions

there. Finally the addition of quantifying microglia morphology in the Nodal-PC KOs helps make their case. I still have a few remaining comments that need to be addressed.

1) I appreciate that the authors acknowledge the developmental differences between the brain and lung and the addition of the P8 assessment as well. I still do not feel like they set the stage for the developmental/homeostatic timeframe each organ is in when they administer tamoxifen and knock-out the various genes at P1. A simple description of the developmental phase the lung and brain are in during deletion of these various genes would be helpful. Additionally, it would be extremely helpful to add in the age labels to all the figures, this is done in the supplement nicely but not in the main figures.

2) Still not sure of the EM images since there is no quantification and lack of similar looking/types of blood vessels here. The authors describe the endothelial protrusions as abnormal, when they have been observed before in the healthy brain (PMID: 26604921). Also, still not sure where they are referencing to the perivascular space in Fig. 4j. I wonder if the authors are mixing up enlargement of the perivascular space with potential thickening of the vascular basement membranes which are two different things and difficult to make out here. And it is unclear how many areas and animals per genotype were looked at for these studies to make these conclusions.

3) The authors still do not provide or discuss a clear mechanism for BBB leakage and vessel rupture. The lack of vessel integrity in the Nodal-PC KOs seem to still ride on the electron micrographs, but junctions appear to be fine. The authors state that the basement membrane is enlarged (and perivascular spaces?) and there are endothelial protrusions, however how these features relate to vessel integrity is not discussed. The reviewer panel showing transcytosis helps and could explain BBB leakage but not vessel rupture. Further, the activation of microglia and astrocytes in the areas lacking active leakage is interesting but this does not rule out historical bleeds or leakage points that have been cleaned up and thus no presence of Ter119+ RBCs/IgG.

Reviewer #4

(Remarks to the Author)

The manuscript by Rasouli et al. entitled "Pericytes are organ-specific regulators of tissue morphogenesis" addressed now all our major concerns, particularly the analysis of vascular leakage and astrocyte proliferation/activation in Nodal mutants. The manuscript is now suitable for publication.

Version 2:

Reviewer comments:

Reviewer #1

(Remarks to the Author)

The authors have addressed my 2nd round comments well and I have no more questions. The manuscript is now suitable for publication.

Reviewer #2

(Remarks to the Author)

The authors have addressed most concerns and have adequately rephrased several of their conclusions and interpretations. The major concern regarding the *Pdgfrb*-CreERT line was only partially clarified. The authors state that their *Pdgfrb*-CreERT line does not exhibit developmental defects as reported by The Jackson Laboratory. However, no experimental evidence is provided in support of this claim. It is therefore suggested that the authors include comparative data of body weights and fertility-related parameters (e.g., litter size at birth). If no differences are observed, then this would settle the case. This is not a semantic issue. Given that the Jackson lab deposited line is tainted, solid clarification is necessary, first, to corroborate the robustness of the authors' findings and second, to alert the readers that there may be reproducibility issues when referring to publicly available reagents, such as the Jackson lab strain in this case.

Reviewer #3

(Remarks to the Author)

The authors have fully addressed all of my comments and concerns. I believe the manuscript is in good form for publication.

We would like to thank the reviewers for their constructive comments and the many insightful suggestions, which have helped a great deal to improve the quality of our manuscript. While a detailed point-by-point response to each comment is provided further below, we would like to begin with a summary of the most important changes and additions made in the revised manuscript:

1. Volcano plots showing the differential expression of genes (**Fig. 1h**) and ligand transcripts (**Fig. 1i**) in pericytes from P14 lung² versus P12 brain.
2. Incorporation of additional markers to clearly distinguish pericytes from SMCs. These analyses are now included in **Supplementary Fig. 3b, c**, **Supplementary Fig. 9b, c**, and **Supplementary Fig. 11a, b**.
3. CellPhoneDB analyses on our scRNA-seq datasets to explore and characterize the potential interaction networks between pericytes and other cell types (**Supplementary Fig. 1** and **Supplementary Fig. 2**).
4. Discussion of the potential clinical implications of pericyte-derived HGF, BDNF, and Nodal in the disease context.
5. New scRNA-seq data uncovered other potential paracrine pericyte-derived regulators, which we are now presenting as supplementary information in Excel format;
Supplementary Table 1: A full list of additional secreted factors that are significantly and uniquely expressed in either the lung or the brain pericytes.
Supplementary Table 2: A full list of additional secreted factors significantly expressed in lung, brain, or both organs
Supplementary Table 3: A full list of genes differentially expressed between lung and brain pericytes that encode secreted factors
Supplementary Table 4: A full list of differentially expressed genes (DEGs) between lung and brain pericytes
6. Inclusion of additional quantitative analyses for all the requested figure panels.
7. Added quantitation of lung airspace volume, lung volume measurement, and lung function parameters (arterial oxygen saturation and respiratory rate) for P21 *Bdnf*^{iPCKO}, *Ntrk2*^{ΔEC}, *Hgf*^{iPCKO}, *Nodal*^{iPCKO} mutants and their respective controls.
8. New analysis for vascular density in lung and brain cortex (based on ICAM2+ surface area) for *Bdnf*^{iPCKO}, *Ntrk2*^{ΔEC}, *Hgf*^{iPCKO}, *Nodal*^{iPCKO} mutants and their respective controls.
9. Quantitation of the percentage of PDGFRβ⁺ cells relative to total cells in each mutant model.
10. Quantitation of the p-TrkB/TrkB ratio in *Bdnf*^{iPCKO} and littermate control total lung lysates. We also show a reduced phosphorylation of the TrkB receptor in endothelial cells of *Bdnf*^{iPCKO} lungs using immunostaining (**Supplementary Fig. 3 g and h**).
11. Analysis of the effect of *Hgf*, *Bdnf* and *Nodal* deletion in adult lungs (new data shown in rebuttal letter).
12. Characterization of the effect of *Hgf* and *Bdnf* deletion in P21 brain (**Supplementary Fig. 5** and **Supplementary Fig. 7**).
13. Analyses of vascular leakage and astrocyte activation in P8 *Nodal* mutant and littermate control brains (**Supplementary Fig. 13**).

14. New analyses of vascular leakage and microglia activation in *Nodal* mutant and littermate control brains at P8 (**Supplementary Fig. 18**) and P12 (**Supplementary Fig. 17**).
15. Validation that c-Met is present in LAMP3⁺ AT2 cells (**Figure 3 a, d** and **Supplementary Fig. 8b, c**) but absent from ERG⁺ endothelial cells (**Supplementary Fig. 8d**).
16. Additional EdU labeling showing cell proliferation in both brain and lung across all mutant models.
17. Addition of a statement summarizing the limitations of our study to the Discussion.
18. Additional data validating *Met* expression in AT2 cells compared to other lung cells and *Nodal* expression in pericytes compared to other brain cell types (new data shown in rebuttal letter).
19. Analysis of scRNA-seq data to validate effective depletion of *Nodal* in brain pericytes (**Supplementary Fig. 11c, d**) and of *Bdnf* (**Supplementary Fig. 3a**) and *Hgf* (**Supplementary Fig. 9a**) in lung pericytes by *Pdgfrb-CreERT2*.
20. Reanalysis of our scRNA sequencing data to evaluate changes in cellular composition after BDNF, HGF, and *Nodal* depletion (new data shown in rebuttal letter).
21. New data confirming the effects of *Nodal* depletion on Smad phosphorylation on mouse tissue (shown in rebuttal letter).
22. New EdU staining to evaluate astrocyte proliferation in the cortex of *Nodal* mutant mice (shown in rebuttal letter).
23. Quantification of morphological changes in cortical microglia by measuring microglia average volume and AIF1 mean intensity in all mutant and control brains.

Reviewer #1 (Remarks to the Author)

Research on angiocrine signaling has primarily focused on endothelial cells (ECs) and their subpopulations in organ growth and regeneration, while the role of pericytes as a source of secreted factors influencing neighboring cells has been relatively understudied. In this study, Rasouli et al. investigate the role of three pericyte-derived paracrine factors—HGF, BDNF, and *Nodal*—in postnatal lung and brain development. Using many versatile methods including multiple tissue-specific and inducible mouse genetics, high-resolution imaging, single-cell RNA sequencing, and in vitro assays, the authors explore the function of these pericyte-derived growth factors. They demonstrate that pericyte-derived hepatocyte growth factor (HGF) is essential for alveologenesis, while brain-derived neurotrophic factor (BDNF) regulates lung development by interacting with the TrkB receptor in pulmonary endothelial cells. Neither factor appears necessary for postnatal brain development. Conversely, pericyte-derived *Nodal*, a TGF β family growth factor, plays a critical role in regulating blood vessel growth and barrier function in the postnatal brain but is not required for lung morphogenesis. These findings establish pericytes as key regulators of angiocrine signaling in an organ-specific manner. The manuscript is well-written and presents significant and critical insights into pericyte function during development. However, the findings are largely descriptive, lacking a deep mechanistic investigation. Additionally, the rationale for studying HGF,

BDNF, and Nodal is unclear, as their effects appear unrelated and independent. Consequently, the overall significance of this work is diminished.

Major review

1. A major limitation of this study is the overlap between pericytes and SMCs due to the use of *Pdgfrb*-CreERT2, which does not inclusively label pericytes. How do the authors account for potential contamination from SMCs?

In line 108, the manuscript states that “*Pdgfrb*-CreERT2-mediated recombination at P1–P3 does not label PDGFR α + fibroblasts, α SMA+ bronchial SMCs, or myofibroblasts”. But much data was generated after P12. What about SMC components after P12?

Reply: Thank you for this comment. We totally agree that *Pdgfrb* is not an exclusively pericyte-specific marker and we have therefore now clearly distinguished pericytes and vascular SMCs in the revised manuscript. In lung, our scRNA-seq data shows that *Pdgfrb* expression is higher in pericytes relative to SMCs. In brain, a similar but smaller difference is notable (see below).

Reviewer Fig. 1: *Pdgfrb* expression in pericytes and SMCs based on scRNA-seq analysis of control lungs.

Further analyses of our scRNA-seq data revealed that *Bdnf* is expressed by both control lung pericytes and SMCs, whereas *Pdgfrb*-CreERT2-mediated inactivation of the gene primarily affects pericytes and causes a more moderate reduction of transcript levels in the SMC cluster. Furthermore, *Hgf* expression is restricted to lung pericytes and not observed in SMCs.

Reviewer Fig. 2: *Bdnf* and *Hgf* expression in pericytes and SMCs based on scRNA-seq analysis of lungs.

Our previous work also established that tamoxifen-induced activation of *Pdgfrb*-CreERT2 at P1–P3 does not lead to labeling of endothelial cells (at P7, P21, and P50), PDGFR α + fibroblasts, α SMA+ bronchial smooth muscle cells, or myofibroblasts (at P7 and P21), presumably reflecting the lower expression of *Pdgfrb* in these cells¹.

[FIGURE REDACTED]

Reviewer Fig. 3: Adapted from Kato et al., 2018. **a.** Scheme showing the time points of tamoxifen administration (P1–3) and analysis for the *Pdgfrb-CreERT2* R26-mT/mG mice. High magnification images of *Pdgfrb-CreERT2* R26-mT/mG lung sections at indicated stages showing pulmonary GFP+ PCs (green), PECAM1+ ECs, PDGFR β + PCs, Notch3+ PCs, NG2+ PCs, PDGFR α + fibroblasts or α SMA+ bronchial smooth muscle cells/myofibroblasts (red, as indicated). Arrows in b-d indicate PC cell bodies, in **b–f.** GFP-positive PDGFR β + cells (**c**) or Notch3+/NG2+ cells (**d**) GFP-negative PDGFR α + cells (**d**) or α SMA+ cells (**e, g**) are marked. Scale bar, 20 μ m (**b, c**), 10 μ m (**d**), 15 μ m (**e–f**) and 30 μ m (**g**).

In addition, we have characterized the *Pdgfrb-CreERT2* line in the brain at P12 and P21 (see below). Although *Pdgfrb-CreERT2*-mediated GFP expression can be detected in both pericytes and SMCs, our study focuses on the brain cortex, where the majority of endothelial cells are capillary-associated and therefore predominantly covered by pericytes. Importantly, our scRNA-seq data show that *Nodal* expression is markedly enriched in cortical pericytes compared to SMCs (see below), underscoring that pericytes represent the principal source of the factor in the brain cortex.

Reviewer Fig. 4: Characterization of *Pdgfrb-CreERT2 R26-mT/mG* mice. Confocal images of sections from brain cortex sections at P12 (**a** and **b**) and P21(**c** and **d**) showing brain GFP+ PCs (green), IB4+ or ICAM2+ ECs, PDGFR β + PCs, α SMA-covered arteries. Scale bar, 100 μ m (**a** and **b**), 50 μ m (**c** and **d**).

In addition, the scRNA-seq data indicate that pericytes and SMCs cluster together as mural cells (Fig S1a,b). Additional markers should be incorporated to clearly distinguish pericytes from SMCs and characterize pericyte specific pathways.

We thank the reviewer for this helpful suggestion. As recommended, our revised bioinformatic analyses distinguish pericytes from SMCs (see subclusters in Supplementary Fig. 3b, c, Supplementary Fig. 9b, c, and Supplementary Fig. 11a, b).

Furthermore, we have characterized potential interactions of pericyte-enriched secreted

factors, which is presented in Supplementary Fig. 1a and Supplementary Fig. 2a of the revised manuscript.

2. The study focuses on pericyte-derived factors in murine models, but what is known about their role in human lung and brain development? Discussing the potential clinical implications of pericyte-derived HGF, BDNF, and Nodal in disease contexts would strengthen the impact of the findings.

Thank you for this suggestion. While our study focuses on developmental processes and the role of pericyte-derived factors, we now mention potential clinical implications of pericyte-derived factors in human lung and brain development in the revised Discussion.

HGF, for example, plays a protective role in the developing and injured lung. In premature infants, reduced HGF levels in tracheal aspirates are associated with an increased risk of severe bronchopulmonary dysplasia (BPD)³. Nonsynonymous mutations in the *HGF* gene may further predispose infants to BPD⁴. Experimental studies show that recombinant HGF treatment ameliorates hyperoxia-induced lung injury in neonatal mice⁵. In adult lung disease, the HGF/c-Met pathway modulates oxidative stress in alveolar epithelial cells, and reduced HGF secretion in peripheral lung tissue correlates with COPD severity⁶⁻⁸. These observations underscore the potential importance of pericyte-derived HGF in both developmental and chronic lung pathologies.

BDNF–hypoxia interactions may contribute to pulmonary hypertension⁹. In the lung, TrkB and STAT3 activation promotes non-small-cell lung cancer proliferation¹⁰, whereas STAT3–BDNF–TrkB signaling supports alveolar epithelial regeneration after injury¹¹, highlighting context-dependent roles.

Nodal has been extensively studied in cancer biology, where its upregulation has been reported in metastatic melanoma, pancreatic and breast cancers, as well as glioblastoma¹²⁻¹⁷. Inhibition of Nodal suppresses angiogenesis and tumor growth in human gliomas, suggesting that Nodal may represent a potential therapeutic target in these malignancies^{15,16}. Beyond oncology, Nodal also plays a neuroprotective role in the central nervous system, mitigating inflammation and oxidative stress in models of hypoxic–ischemic brain injury¹⁸.

Taken together, these studies indicate that HGF, BDNF, and Nodal may influence human lung and brain development and function in health and disease.

3. The rationale for investigating HGF, BDNF, and Nodal is weak if the justification for selecting these three factors is only based on single-cell RNA sequencing data. What are their baseline expression levels in wild-type animal organs? In Fig. 1g, are there additional factors uniquely expressed in the lung or brain? If so, a complete list should be provided. The Discussion should address the contributions of other factors.

Thank you for this comment. We agree that we need to explain better why HGF, BDNF, and Nodal were chosen as potential pericyte-derived regulators of tissue morphogenesis for

further analysis. This decision was made early in the project, mostly based on bulk sequencing data that we are not presenting here because of issues relating to the contamination with non-mural cells. Despite of this initial limitation, publicly available scRNA-seq data published by the Betsholtz laboratory from adult mice (<http://betsholtzlab.org/VascularSingleCells/database.html>) confirmed differential expression of transcripts for the three growth factors between mural cells from brain and lung, respectively. In the course of the project, emerging results and, in particular, new scRNA-seq data uncovered other potential paracrine pericyte-derived regulators, which we are now presenting as supplementary information in Excel format, as requested by the reviewer.

To address the other questions raised in this comment, we have now included baseline expression levels of *Hgf*, *Bdnf* and *Nodal* in wild-type brain and lung, which are summarized in Excel format in Supplementary Table 1. This Table also includes a full list of additional secreted factors that are significantly and uniquely expressed in either the lung or the brain. In addition, volcano plots of differentially expressed genes (DEGs) between lung and brain pericytes are shown in Fig. 1g, h.

	baseMean	log2FoldChange	lfcSE	stat	pvalue	padj	percent_expressed_P12_brain_pericytes	mean_expression_P12_brain_pericytes	percent_expressed_P14_lung_pericytes	mean_expression_P14_lung_pericytes	name
1											
2	Pdgfrd	47.65933	-6.699126335	1.306020579	-4.868031	1.13E-06	2.00E-05	0.490196078	0.003937958	22.92993631	0.314279886 Pdgfrd
3	Bdnf	38.631503	-6.357538403	1.315890727	-4.6238305	3.77E-06	6.10E-05	0.490196078	0.006452	23.77919321	0.309065299 Bdnf
4	Scgb3a2	25.071771	-7.731548434	2.704390624	-3.1978073	0.00138477	0.01254716	0	0	21.86836518	0.247424439 Scgb3a2
5	Grp	45.110928	-8.76505683	2.793403844	-3.15763825	0.00034838	0.00377149	0	0	14.01273885	0.231333722 Grp
6	Ntf3	25.060896	-7.730737342	2.704392434	-3.1974376	0.00138654	0.01255836	0	0	16.56050955	0.209114926 Ntf3
7	Wnt4	21.709218	-4.60136343	1.086395351	-4.3945122	1.11E-05	0.00016544	0.580392157	0.008344174	16.77282378	0.204619568 Wnt4
8	Hgf	30.887347	-7.38950192	2.675715374	-3.0747801	0.00210658	0.01789398	0	0	16.1358811	0.202017346 Hgf
9	Vegfd	20.48923	-5.319612978	1.355027892	-3.8994563	9.64E-05	0.00120873	0.490196078	0.004561406	14.22505308	0.163772616 Vegfd
10	Ccl19	19.395396	-7.242286737	2.668803499	-3.0199562	0.00252811	0.02097848	0	0	13.5881104	0.151736224 Ccl19
11	Edn3	13.674879	-3.763236919	1.153336622	-3.0150791	0.00030026	0.00330438	0.580392157	0.008015216	12.31422505	0.134009809 Edn3
12	Eda	15.980754	-4.866968027	1.395530554	-3.5802399	0.00034328	0.00372492	0.490196078	0.003761332	10.1910828	0.131727902 Eda
13	Sema3c	23.475416	-7.592584773	2.706285393	-3.1390536	0.00169494	0.01486483	0	0	10.40339703	0.128103053 Sema3c
14	Ccl6	10.221723	-4.024372495	1.462340445	-3.0561143	0.00224226	0.0188808	0.490196078	0.003491476	10.1910828	0.110784996 Ccl6
15	Il15	11.328206	-3.390845433	1.195935166	-3.2856334	0.00101753	0.00964084	0.580392157	0.007862416	9.554140127	0.108283509 Il15
16	Lgals9	34.940714	1.742117864	0.480758375	3.9534378	7.70E-05	0.00098552	30.39215686	0.341844751	7.855626327	0.079616614 Lgals9
17	Tnfrsf9	16.945198	1.827408176	0.702176562	3.07834535	0.00208154	0.01773312	13.7254902	0.15631268	2.972399151	0.035079062 Tnfrsf9
18	Adm	27.812361	3.531337198	0.652680828	5.67117377	1.42E-08	3.40E-07	24.01960784	0.291808527	1.910828025	0.019517285 Adm
19	Il2	12.915138	2.428286087	0.856350697	3.34920018	0.00081045	0.00790446	19.11764706	0.184313753	2.972399151	0.019127053 Il2
20	Ccl3	15.447602	3.825559003	0.93461857	4.3972689	1.10E-05	0.00016352	8.823529412	0.118043569	0.8492569	0.008609686 Ccl3
21	Pin	1591.7312	15.27217056	3.174958918	6.06798774	1.30E-09	3.52E-08	100	0	0	0 Pin
22	Nodal	56.643856	9.927923266	2.682382861	4.00848803	6.11E-05	0.00079757	46.56862745	0.654804355	0	0 Nodal
23	Nts	44.344979	9.502113177	2.648025584	3.83495977	0.00012558	0.00153557	20.0903922	0.362184897	0	0 Nts

Supplementary Table 1. List of additional secreted factors that are significantly and uniquely expressed in either the lung or brain pericytes.

4. Is the baseline level of these three factors Hgf, Bdnf, and Nodal in a wild-type animal? In Fig 1g, are there other factors inclusively expressed in the lung or brain? Please provide the entire list. The contribution from other factors should be included in the Discussion.

We agree, but feel that we have already addressed this point above in our reply to point 3.

We would also like to emphasize that the main goal of our study is to establish that pericyte-derived factors control morphogenetic processes. To this end, we have investigated three candidate regulators and two model organs. In contrast, we are not claiming that we have comprehensively investigated all potential pericyte-derived signals and, in the absence of direct experimental evidence, meaningful discussion of the putative contribution of other factors remains speculative. Nevertheless, we mention potential roles of other factors and additional cellular sources in the context of limitations of our study in the Discussion.

In each of three KO animals, can phenotypes be rescued by supplementing these factors back?

Thank you for this comment. We have not attempted to supplement these factors in order to rescue the KO phenotypes for several reasons. First, it would be technically challenging to administer the three factors during postnatal development. Second, the injected factors would act in a systemic fashion on different organs and cell types, likely leading to complex phenotypic outcomes. Furthermore, it is fairly evident that such experiments would go substantially beyond the scope of the current study.

What the correlation among these three factors? Do they regulate each other? What if double KO Hgf and BDNF?

Thank you for this comment. As stated earlier, we have investigated the three factors as candidate regulators to establish that pericyte-derived signals regulate tissue morphogenesis. As the logistics of genetic experiments in mice is quite complex and demanding, we could not investigate a very long list of putative pericyte-derived molecular signals (see below) or many different organs. By the same token, the characterization of the potential interplay between HGF, BDNF and Nodal or the effect of double or triple compound mutants would be clearly outside the scope of the current study.

However, in our previous work we found that expression of both *Hgf* and *Bdnf* was downregulated in YAP1/TAZ-deficient pulmonary pericytes. Together with our current data showing a role for HGF in alveolarization and BDNF in pulmonary vascular development, this suggests that a combined loss of HGF and BDNF could potentially result in stronger and more complex defects, affecting both alveolarization and pulmonary vascularization. This might be an interesting direction for future research.

**Reviewer Fig. 5:** Loss of YAP1 and TAZ in PDGFRβ+ pericytes reduces the expression of several growth factors in these cells, including *Bdnf* and *Hgf*. Actively translating polyribosome-bound RNA in pericytes was isolated and sequenced from control and $Yap1,Wwtr1^{iPCKO}$ mutant lungs in triplicate[1].

5. Figures 2n, 4a, 4j, 5h, 6b require quantification.

Thank you for this comment. The quantification for Fig. 5h was already included in the original manuscript. We would like to clarify that Fig. 4a is a tile scan of coronal sections through the P12 *Nodal*^{iPCKO} and littermate control brain cortex immunostained for ICAM2 (green). However, the quantitation of the ICAM2-stained endothelial surface in Fig. 4b has been quantified and is presented in Fig. 4c. In addition, we have now added the missing quantifications for the following panels in the revised version of the manuscript: Fig. 2n (now Fig. 2q), Fig. 4j (now Fig. 4k), and Fig. 6b (now Fig. 6d).

WB validation of TrkB and phosphorylated TrkB (p-TrkB) in Fig. 2n should be performed using isolated endothelial cells rather than whole lung lysates.

We have attempted WB validation of TrkB and p-TrkB in isolated endothelial cells several times. Despite optimization of our protocol to minimize processing time and prevent loss of protein phosphorylation, we consistently observed a strongly diminished phospho-TrkB signal after the cell-sorting procedure.

We have therefore used phospho-TrkB (p-TrkB) immunostaining to show the reduction of the signal in the *Bdnf*^{iPCKO} pulmonary ICAM2⁺ endothelium. The new data are presented in Supplementary Fig. 3g, h of the revised manuscript.

Reviewer Fig. 6: Immunostaining showing reduced p-TrkB signal in endothelial cells of *Bdnf*^{iPCKO} lungs.

6. This study only studies pericytes from the brain and lung; thus, “organ-specific” is an overstatement; tone down the statement to be more brain and lung-specific, for the title and introduction.

Thank you for your comment, which we have carefully considered. However, it is correct that we are showing organ-specific differences even though we had to limit ourselves to two model organs. The extent of the genetic experiments, scRNA-seq approaches and other demanding methodologies in this study did not permit the characterization of additional organs or factors.

Importantly, we make it unambiguously clear in the Abstract and Introduction that we are focusing on lung and brain. At the conceptual level, we can nevertheless conclude that pericytes acquire organ-specific features and secrete factors that control morphogenetic processes in the surrounding tissue so that the title of our manuscript is justified.

7. The Discussion lacks a thorough examination of the study's limitations and directions for future research.

We agree and have revised the Discussion so that it includes a section highlighting the limitations of our study.

8. It would be beneficial to explore whether Hgf, Nodal, and Bdnf play similar roles in the adult lung. For example, when These animals are over P21 then admin tamoxifen.

By tamoxifen injection to adult animals (10 week), consisted of intraperitoneal (i.p.) injections of 1 mg tamoxifen (1 mg ml⁻¹) over 5 consecutive days, we have evaluated the effect of *Hgf*, *Bdnf* and *Nodal* inactivation in adult lungs, which did not reveal any obvious defects in *Nodal* or *Bdnf* mutant lungs, which, instead, showed normal vascularization and alveolarization (see below). However, we found some hotspots with high levels of RAGE staining in *Hgf*^{iPCKO} mutant lungs.

Reviewer Fig. 7: No obvious defects and normal vascularization and alveolarization in adult *Bdnf*^{iPCKO} lungs.

(a) Experimental timeline for Tamoxifen (TMX) administration and imaging of adult mice.

(b) Confocal images of ICAM2⁺ endothelium (grey) and RAGE⁺ epithelium (red) show that lung development is not affected in adult *Bdnf*^{iPCKO} mutants compared to littermate controls.

(c-e) Graphs showing the ICAM2-stained vascular density **(c)** airspace volume **(d)**, and lung volume measurement **(e)**.

(f, g) Maximum intensity projections of *Bdnf*^{iPCKO} and control lung sections showing EC nuclei (ERG⁺, red), pericytes (PDGFRβ⁺, white) **(e)**, and LAMP3⁺ AT2 cells **(f)** in total cells (DAPI⁺, blue).

(h-k) Graphs showing the percentage of ERG⁺ cells in total cells **(h)**, percentage of PDGFRβ⁺ cells in total cells **(i)**, ratio of ERG⁺ to PDGFRβ⁺ cells **(j)**, and the percentage of LAMP3⁺ AT2 cells in total cells **(k)** in adult *Bdnf*^{iPCKO} and littermate control lungs. Data represent mean ± s.e.m. (n=3); P-values, unpaired two tailed student t-test.

Reviewer Fig. 8: Hotspots with high levels of RAGE staining in adult *Hgf*^{iPCKO} mutant lungs.

(a) Experimental timeline for Tamoxifen (TMX) administration and imaging of adult mice. **(b)** Confocal images of ICAM2⁺ endothelium (grey) and RAGE⁺ epithelium (red) show that lung development is not affected in adult *Hgf*^{iPCKO} mutants compared to littermate controls. However, we have observed hotspots with high levels of staining for RAGE in adult *Hgf*^{iPCKO} mutant lungs.

(c-e) Graphs showing the ICAM2-stained vascular density (c) airspace volume (d), and lung volume measurement (e).
 (f, g) Maximum intensity projections of *Hgf*^{iPCKO} and control lung sections showing EC nuclei (ERG+, red), pericytes (PDGFRβ+, white) (f), and LAMP3+ AT2 cells (g) in total cells (DAPI+, blue).
 (h-k) Graphs showing the percentage of ERG+ cells in total cells (h), percentage of PDGFRβ+ cells in total cells (i), ratio of ERG+ to PDGFRβ+ cells (j), and the percentage of LAMP3+ AT2 cells in total cells (k) in adult *Hgf*^{iPCKO} and littermate control lungs. Data represent mean ± s.e.m. (n=4); P-values, unpaired two tailed student t-test.

Reviewer Fig. 9: No obvious defects and normal vascularization and alveolarization in adult *Nodal*^{iPCKO} lungs. (a) Experimental timeline for Tamoxifen (TMX) administration and imaging of adult mice. (b) Confocal images of ICAM2+ endothelium (grey) and RAGE+ epithelium (red) show that lung development is not affected in adult *Nodal*^{iPCKO} mutants compared to littermate controls. (c-e) Graphs showing the ICAM2-stained vascular density (c) airspace volume (d), and lung volume measurement (e). (f-g) Confocal images of ERG, PDGFRβ, and LAMP3 in control and *Nodal*^{iPCKO} lungs. (h-k) Bar graphs of ERG+, PDGFRβ+, ERG/PDGFRβ ratio, and LAMP3+ cells.

(f, g) Maximum intensity projections of *Nodal*^{iPCKO} and control lung sections showing EC nuclei (ERG+, red), pericytes (PDGFRβ+, white) **(f)**, and LAMP3+ AT2 cells **(g)** in total cells (DAPI+, blue).

(h-k) Graphs showing the percentage of ERG+ cells in total cells **(h)**, percentage of PDGFRβ+ cells in total cells **(i)**, ratio of ERG+ to PDGFRβ+ cells **(j)**, and the percentage of LAMP3+ AT2 cells in total cells **(k)** in adult *Nodal*^{iPCKO} and littermate control lungs. Data represent mean ± s.e.m. (n=4); P-values, unpaired two tailed student t-test.

9. Fig. 2c, 2g, 2h: The Pdgfrβ staining in Fig. 2c suggests that Pdgfrb+ cells are reduced and exhibit morphological enlargement in knockout mice. In Fig. 2g–h, the total Erg+ endothelial cells are lower in *Bdnf* KO mice (40%) compared to controls (50%), but the Erg+ to Pdgfrb+ ratio remains the same. Does this suggest a reduction in Pdgfrb+ cells in KO mice?

Clarification on whether Pdgfrb+ cell numbers are affected in each knockout model and whether their changes are proportional to ECs is necessary.

We thank the reviewer for raising this point. We have quantified the percentage of PDGFRβ+ cells relative to total cells in each knockout model, and these data have been incorporated into the revised manuscript (Fig. 2h, i; Fig. 3v, w; Fig. 4g, h). In essence, *Bdnf* or *Hgf* mutant lungs and *Nodal* mutant brains show small but statistically significant reductions in pericyte numbers. These relatively modest reductions probably reflect mutual signaling interactions with other cell types in the surrounding tissue (e.g. ECs expressing PDGF-B). Given that hypomorphic mutants lacking certain signaling sites in PDGFRβ show preserved pericyte numbers and limited defects^{19,20}, it is unlikely that the observed reductions in pericyte numbers play a major role in our study. Nevertheless, we mention this point in the Discussion of the revised manuscript.

The same analysis of pericyte numbers has also confirmed that there are no changes in *Bdnf* (Suppl. Fig. 4f, g) or *Hgf* mutant brains (Suppl. Fig. 6f, g) and *Nodal* mutant lungs (Suppl. Fig. 10i, j), which confirms that phenotypic alterations are indeed organ-specific and therefore unlikely to be caused by systemic or unspecific/indirect alterations.

10. The authors conclude that pericyte-derived BDNF is critical for lung development but not for the brain. However, lung tissues were evaluated at P21, while brain tissues were examined at P12. Could this time difference affect the results? It would be valuable to assess brain vasculature at P21 for consistency.

As kindly recommended by the reviewer, we have assessed the mutant phenotypes at P21 and found that *Pdgfrb-CreERT2*-controlled inactivation of *Bdnf* or *Hgf* does not affect vascular development or glial cell responses in brain. These new results are presented in Supplementary Fig. 5 and Supplementary Fig. 7 are fully consistent with our findings at P12.

11. Provide a stronger rationale for the inclusion of AIF1 and CD68 in the Fig 6 results section.

AIF1 and cluster of differentiation 68 (CD68) are well-established markers of microglial cells in the CNS, with their upregulation commonly observed in activated microglia. AIF1 contributes to rearrangement of the plasmamembrane and cytoskeleton, whereas CD68 functions as a scavenger receptor in the monocyte lineage²¹⁻²⁶.

Minor

1. Please provide an explanation or full name of what iPCKO in *Bdnf* iPCKO in line 143 to help better understand.

We thank the reviewer for pointing out this omission. We state in the manuscript that iPCKO refers to inducible *Pdgfrb(BAC)-CreERT2*-mediated knockout and we now introduce this term when the inducible knockout strategy is first mentioned for the first time.

2. The methodology for quantifying ICAM2-stained surface area should be described in Fig2q.

Thank you for this suggestion. The methodology for quantifying the ICAM2-stained surface area is now described in the revised manuscript.

3. The first paragraph describes microglial morphology changes, while the second paragraph discusses activation based on chemokine expression from scRNA-seq data. Since the study does not directly analyze microglial morphology, the introduction should clarify the definition of activated microglia.

We thank the reviewer for this important comment. In the revised manuscript, we have clarified the definition of “activated microglia” in the manuscript. We now state that activation can be reflected by both morphological and transcriptional changes. In our study, activation was assessed by quantifying microglial morphology as well as by analyzing transcriptional changes, including chemokine expression identified by scRNA-seq.

4. Are there references supporting the role of phosphorylated SMAD2 in brain microglial inflammation? If so, these should be cited.

Relevant references supporting the role of TGF β -mediated activation of SMAD2 in regulating brain microglial quiescence have been added to the revised manuscript^{27,28}.

5. Ensure proper formatting of mouse (italicized, lowercase) vs. human (uppercase) gene and protein names.

Thank for this suggestion. We made sure that we followed the nomenclature for genes and transcripts in the revised manuscript. There is no equally clear standard for proteins other than having no italicization, but we have made sure that we adhere to commonly used symbols and names.

Reviewer #2 (Remarks to the Author)

Rasouli and co-workers performed an innovative study to investigate the organ-specific regulatory roles of pericytes in tissue morphogenesis. The authors investigated the functions

of the pericyte-derived factors BDNF, HGF, and Nodal in the lungs and in the brain during postnatal development with a focus on elucidating signaling interactions between pericytes and other cell types. Overall, the notion that pericyte-expressed molecules regulate tissue morphogenesis is interesting and a highly timely topic. However, there are several issues that need to be addressed before this preliminary report will make a solid study. The authors should consider the following points to further develop their work:

1. The authors used *Pdgfrb*-CreERT2 mice to achieve pericyte-specific gene deletion. However, this Cre line presents a critical limitation. The Jackson Laboratory has pointed out that “hemizygous mice are smaller than their non-carrier siblings” (Strain #029684) indicating that the presence of the *Pdgfrb*-CreERT2 allele itself may induce developmental differences even prior to experimental manipulation. This inherent phenotype raises concerns that the observed phenotypes may be confounded by baseline abnormalities rather than genetic modification. To ensure the validity of their findings, the authors should confirm their results using, for example, PDGFR β -P2A-CreERT2 (The Jackson Laboratory, Strain #030201), which does not exhibit baseline developmental discrepancies.

We thank the reviewer for the kind assessment of our work and the useful comments. We agree that there might be a potential impact of the *Pdgfrb*-CreERT2 allele itself, which obviously applies to every Cre/creERT2 line. Here, a comparison of the different mutants gives a clear answer. We show that the expression/activity of *Pdgfrb*-CreERT2 does not induce lung defects in *Nodal* mutant lungs (Supplementary Fig. 10). In the brain, *Pdgfrb*-CreERT2 does not affect glial cell responses or vascular development in *Bdnf* or *Hgf* mutants, respectively (Supplementary Fig. 4, 5, 6 and 7). These results demonstrate that the reported phenotypic alterations are not caused by the presence of the *Pdgfrb*-CreERT2 allele. While MGI (<https://www.informatics.jax.org/>) currently lists 5 mouse lines expressing tamoxifen-inducible Cre recombinase under the control of *Pdgfrb*, we are not aware of studies that have directly compared recombination efficiency and other specific features of the different lines.

2. Several conclusions were drawn with a lack of experimental validation resulting in unjustified overinterpretation of the results. The study proposes that pericyte-derived BDNF hampers lung development. However, the authors did not provide convincing data that directly evaluates changes in lung development. The authors should include specific analyses examining lung structure and function similar to what they previously performed¹.

We thank the reviewer for this valuable comment. In response, we have now incorporated quantitative analyses of airspace volume, lung volume, and lung vascular area for the *Bdnf* and *Ntrk2* mutants in Figure 2, but also for the other mutants, into the revised manuscript.

Additionally, the authors state that “inactivation of the HGF gene impairs alveologenesis due to defective interaction with AT2 epithelial cells” (Abstract, line 39). However, the manuscript does not provide evidence that supports the instructive role of AT2 cells in this process. The authors should consider manipulating AT2 epithelial cells and evaluate whether this disruption specifically contributes to impaired alveologenesis.

Thank you for this comment. As shown in Figure 3 of the manuscript, loss of *Hgf* results in impaired alveolarization, which is further supported by our new quantification of airspace and lung volume in *Hgf* mutants compared to controls. Moreover, our immunostaining demonstrates that c-Met is expressed in AT2 cells - identified by LAMP3 or pro-surfactant protein C (proSP-C) staining - supporting that AT2 cells are direct targets of HGF (Figure 3a, d and Supplementary Fig. 8b, c). Furthermore, we would like to point out that there is a robust body of published literature on HGF-mediated regulation of AT2 cells and bronchioalveolar stem cells (BASCs)^{1,29-32}. Other studies have shown that conditional inactivation of the Hgf receptor (c-Met) in AT2 cells impairs alveologenesis (refs. 42 and 43; cited previously in the initial submission)^{6,33}, further supporting an instructive role of AT2 cells in mediating HGF-dependent alveolar development.

[FIGURE REDACTED]

Reviewer Fig. 10: Adapted from Calvi et al., 2013⁶. a. Representative histology of mice deficient in epithelial *Met* expression and controls at two weeks of age. Note patchy airspace enlargement in the targeted mice. Scale bar: 100 μ m. b. Quantitation of SPC+ cells in the airspace by SPC immunohistochemistry in *Met*-deficient mutants relative to control. * $p < 0.05$.

[FIGURE REDACTED]

Reviewer Fig. 11: Adapted from Yamamoto et al., 2007³³. a-b. Histological sections of lungs from newborn *c-met*^{+/+} and *c-met*^{SP-C- Δ/Δ} mice treated with Dox from E14.5 showing dilated distal airspaces with reduced primary septation in *c-met*^{SP-C- Δ/Δ} lungs. Scale bars, 200 μ m.

In line 328, the authors suggest that loss of pericyte-derived Nodal might induce reactive astrogliosis directly which could be enhanced by vascular leakage. However, this statement is descriptive and lacks experimental evidence investigating vascular leakage.

Thank you for the comment. As shown in Supplementary Fig. 12 of the revised manuscript, we have directly assessed vascular permeability in the P12 brain cortex by dextran injection, in combination with TER-119 labeling of red blood cells and IgG staining to evaluate serum extravasation. While confocal microscopy revealed that reactive GFAP+ astrocytes are present in regions of vascular leakage (Supplementary Fig. 12d, f), we also observed GFAP+ astrocytes in regions without TER-119, IgG or Texas Red-dextran extravasation (Supplementary Fig. 12e, g and Reviewer figure below).

Reviewer Fig. 12. Reactive astrogliosis in the *Nodal*^{iPCKO} cortex.

(a, b) Confocal images of P12 brain cortex stained for GFAP (grey) from *Nodal*^{iPCKO} and control littermates injected with Texas Red-dextran (70 kDa) (red), with local leakage (white arrows) **(a)**. Reactive astrocytes (yellow arrows) can also be observed in areas of *Nodal*^{iPCKO} cortex with no leakage (asterisks) **(b)**.

Furthermore, we have now included new analyses of P8 brains (Supplementary Fig. 13), when no vascular leakage was detected using the same approaches as above. Even in these early mutants, we still observed increased GFAP⁺ astrocytes in the cortex, indicating that reactive astrogliosis is initiated prior to vascular leakage.

Reviewer Fig. 13/Supplementary Figure 13. Reactive astrogliosis emerges early in *Nodal*^{iPCKO} cortex.

(a) Confocal images of brain cortex stained for ICAM2 (green) and mouse immunoglobulin G (IgG, red) showing lack of extravasated IgG in P8 *Nodal*^{iPCKO} and control littermates (3/3 brains), but GFAP⁺ astrocytes (white) can also be seen in *Nodal*^{iPCKO} mutants.

(b) Confocal images of GLUT1⁺ (red) capillaries in brain cortex. Red blood cells (Ter119, green) are confined to vessels in P8 *Nodal*^{iPCKO} and control littermates (3/3 brains), but GFAP⁺ astrocytes (white) can also be observed in *Nodal*^{iPCKO} mutants. Quantitation of GFAP⁺ area in the P8 *Nodal*^{iPCKO} and littermate control brain cortex is provided in Supplementary Figure 13c.

Similarly, in line 367, the claim that “microglia can be directly activated by Nodal, with alterations in the brain microenvironment such as increased vascular permeability”, lacks examination of vascular permeability. The authors could perform Miles assays to evaluate vascular permeability.

We thank the reviewer for this important suggestion. To directly address vascular permeability, we assessed barrier integrity in the P12 brain cortex by confocal imaging after dextran injection, together with TER-119 labeling of red blood cells and endogenous mouse IgG staining to detect serum extravasation (Supplementary Fig. 12 and Supplementary Fig. 17). In the P12 cortex, AIF1+ microglia is observed in regions with dextran extravasation (Supplementary Fig. 17a), which might reflect that microglial activation occurs secondary to vascular defects. Importantly, AIF1+ microglial cells are also present in regions lacking detectable dextran extravasation (Supplementary Fig. 17b), indicating that microglial activation is not restricted to sites of vascular leakage.

As for the activation of astrocytes mentioned earlier, we performed additional analyses in the P8 brain (new data included Supplementary Fig. 18). In the P8 cortex, where we detected no vascular leakage, microglial activation was still evident in Nodal mutant animals. These data establish that microglia activation is initiated prior to vascular leakage.

Reviewer Fig. 14/Supplementary Figure 18. Microglia activation emerges early in the *Nodal*^{iPCKO} cortex. (a, b) Confocal images of brain cortex stained for ICAM2 (green) and mouse immunoglobulin G (IgG, white) showing lack of extravasated IgG in P8 *Nodal*^{iPCKO} and control littermates (3/3 brains) (a, b). Imaris surface-rendered image of AIF1+ microglia (yellow) showing that microglia volume is increased in P8 *Nodal*^{iPCKO} mutants (b).

(c) Confocal images of GLUT1+ (red) capillaries in brain cortex. Red blood cells (Ter119, green) are confined to vessels in P8 *Nodal*^{iPCKO} and control littermates (3/3 brains), but increased microglia volume (white) can also be observed in P8 *Nodal*^{iPCKO} mutants.

Quantitation of AIF1 expression and average microglial volume in the P8 *Nodal*^{iPCKO} and littermate control brain cortex are provided in Supplementary Figure 18d, e.

Furthermore, in line 235, the authors state that “the reduced vascularization could be secondary to the defects...”. Yet, no experimental evidence was presented in support of this hypothesis. The authors need to perform appropriate experiments to validate these assertions.

Thank you for your comment. Our scRNA-seq data indicate that *Met* is expressed by AT2 cells but not by lung endothelial cells (Supplementary Fig. 8a, Supplementary Fig. 9d-g). We further validated this by immunostaining, which confirmed that c-Met is present in LAMP3⁺ AT2 cells (Figure 3 a, d and Supplementary Fig. 8b, c) and absent from ERG⁺ endothelial cells (Supplementary Fig. 8d).

These findings indicate that the reduced vascularization in postnatal *Hgf*^{iPCKO} lungs is unlikely to reflect a direct functional role in the postnatal pulmonary endothelium.

3. The manuscript contains several inconsistencies in both methodology and data analysis, which affect the interpretation of the results. The authors assess pericyte coverage using different analyses across different figures. Sup. Fig. 2d and Sup. Fig. 3a report pericyte coverage, whereas Fig. 2c, Fig. 3q, Sup. Fig. 5b, and Fig. 4d quantify the ratio of ECs (ERG⁺ cells) to pericytes (PDGFRβ⁺ cells).

We thank the reviewer for pointing out this concern and apologize for the lack of clarity in our original submission. To address this, we have now performed additional analyses quantifying both the percentage of endothelial cells (ERG⁺) and pericytes (PDGFRβ⁺) relative to the total cell population, as well as the ratio of endothelial cells to pericytes in both lung and brain samples. These results are now clearly presented in the revised manuscript, ensuring consistency across figures and improving the interpretability of our data.

Additionally, cell proliferation is evaluated using different markers. Ki67 staining is used in Fig. 2e, Fig. 2p, and Sup. Fig. 5e, whereas EdU staining is employed in Fig. 4e and Sup. Fig. 2a. Furthermore, in Fig. 4e and Sup. Fig. 2a, EdU is co-stained with the endothelial cell marker ICAM2, whereas in Fig. 2e, Fig. 2p, and Sup. Fig. 5e, single staining of Ki67 is performed without specifying the cell type. The authors need to apply a consistent approach to evaluate proliferation and specify the proliferative cell type analyzed in all images.

We thank the reviewer for this comment and apologize for the previous inconsistency. As requested, we have now included EdU labeling to evaluate cell proliferation in both brain and lung across all mutant models.

When quantifying vascular area, ICAM2 is predominantly used throughout the manuscript, but Sup. Fig. 3a and Sup. Fig. 7c use Glut1 staining. To enhance the consistency of the study, the authors should either standardize their methodologies and analyses across figures or provide a clear explanation for using different approaches.

We apologize for any confusion. GLUT1 is a widely used vessel marker in the central nervous system because the protein is highly expressed in brain endothelial cells. Importantly, both GLUT1 and ICAM2 immunostaining produced consistent results.

Throughout the original manuscript, ICAM2 staining was primarily used to quantify vascular area, but we have now included ICAM2 staining in Supplementary Figure 3a (**Supplementary Fig. 6a in the revised manuscript**) for better consistency.

However, for the assessment of red blood cell leakage by TER-119 immunostaining, we had to GLUT1 instead of ICAM2 because primary antibodies were raised in the same host species and could not be combined.

4. In Fig.3b, the authors examine the role of pericyte-derived HGF in alveologenesis. However, the data presented are insufficient to fully support their claim. Given the authors' previous work in Nature Communications¹, where alveolar development was assessed through parameters such as capillary formation and airspace volume similar analyses should be conducted.

Thank you for this comment. As shown in Figure 3 of the manuscript, loss of *Hgf* results in impaired alveolarization, which is further supported by our new quantification of airspace volume, lung volume, and ICAM2-stained surface (capillary formation) in *Hgf* mutants compared to controls. The same analyses have also been included for the other mutants in the revised manuscript.

5. In Fig.5h, line 319, the authors claim that “pericyte-derived Nodal might directly regulate astrocyte behavior...”. However, the experiment only measures the phosphorylation of astrocyte downstream signal transducer upon rhNodal treatment. It is recommended to perform additional experiments such as co-culture of pericyte and astrocyte, or conditioned medium treatment, to further examine the impact of pericyte-derived Nodal on astrocyte behavior.

We appreciate suggestion and agree that it would be informative to study the role of pericytes in co-culture experiments. We have carefully considered this option, but, based on bulk RNA-seq analysis, cultured pericytes rapidly lose the expression of several key markers, including *Nodal*. It is probably not overly surprising that pericytes lose organ-specific feature when they are placed in culture and no longer faithfully retain their *in vivo* properties, making such experiments less physiologically relevant.

Additionally, astrocytes are highly sensitive to the composition of the culture medium and the substrate coating, which differ substantially from those required for pericyte culture. Combining these specialized culture conditions, e.g. by mixing media, will alter the baseline behavior of these cells and complicate the interpretation of results.

For these reasons, we opted to use recombinant human Nodal (rhNodal) together with the TGF- β receptor inhibitor SB431542 in astrocyte cultures because this is a more controlled and direct way to assess the role of signaling interactions.

6. Several images lack quantification. For example, Fig. 4a, Fig. 6a, Sup. Fig. 9b and Sup.

Fig. 8b present images without corresponding statistical analysis. The authors need to include appropriate data analyses for these images to strengthen their conclusions.

Thank you for this helpful comment. We would like to clarify that **Fig. 4a** is a tile scan of coronal sections through the P12 *Nodal*^{iPCKO} and littermate control brain cortex immunostained for ICAM2 (green). The ICAM2-stained surface shown in **Fig. 4b** has been quantified and is presented in **Fig. 4c**.

In addition, as requested, we have now performed and included the missing quantifications for Fig. 6a (now presented in **Fig. 6b, c**), **Supplementary Fig. 9b** (now presented in **Fig. Supplementary Fig. 15e**), and Supplementary Fig. 8b (now presented in **Fig. Supplementary Fig. 14d, e**) in the revised manuscript.

7. The manuscript would benefit from further refinement of its language to arrive at more solid conclusions. In Sup. Fig. 7c, lines 296–300, the authors state that reactive GFAP⁺ astrocytes are present in both regions with and without vascular leakage. However, this conclusion does not deliver a clear message regarding the relationship between astrogliosis and vascular leakage.

We thank the reviewer for this important point. Based on the literature, vascular leakage can indeed induce astrocyte activation. Consistent with this notion, we show GFAP⁺ astrocytes in regions with vascular leakage (**Supplementary Fig. 12d, f**). However, we also observed GFAP⁺ astrocytes in regions without TER-119 or IgG extravasation or overt vascular leakage (**Supplementary Fig. 12e, g**).

To further clarify the relationship, we have now included new data of brains at P8 in **Supplementary Fig. 13**. At this early stage, no vascular leakage was detected in mutants, whereas the number of GFAP⁺ astrocytes is already increased. These findings strengthen the conclusion that vascular leakage might contribute to astrogliosis, GFAP⁺ astrocytes emerge prior to barrier defects.

Similarly, when discussing markers associated with microglial activation in Fig. 6f and Sup. Fig. 10a, the rationale for separately mentioning *Axl* and *Nes* as “other markers of activated microglia” in line 345 is unclear. The authors should clarify why these two markers are specifically highlighted.

Thank you for this comments. We have now provided references for the roles of *Nestin* and *Axl* in microglia in the relevant section of the main text.

8. Despite performing several scRNA-seq experiments of postnatal lung and cerebral cortex, the manuscript does not make sufficient use of the scRNA-seq data. The authors could further explore the interaction networks between pericytes and other cell types by performing CellPhoneDB analyses on their scRNA-seq data, especially for specifying cell-cell communication mediated by receptor-ligand complexes.

We agree. As suggested by the reviewer, we have now performed CellPhoneDB analyses on

our scRNA-seq datasets to explore potential interaction networks between pericytes and other cell types. These analyses specifically highlight cell–cell communication. The results of these new analyses have been included in the revised manuscript (Supplementary Fig. 1 and 2) to provide a more comprehensive characterization of pericyte interactions within lung and the cerebral cortex.

9. The study is performed solely with mouse models and in vitro assays. It is recommended to include the analyses of human data or mention the limitation of the study in the discussion.

We agree and have added a section to the Discussion that summarizes the limitations of our study.

10. In Sup. Fig. 1a and Sup. Fig. 4a, there are no enriched *Met* and *Nodal* transcripts in the heatmaps displaying top marker genes, as stated by the authors in line 213 and line 255. The authors should provide clarification or further analysis to support these claims.

In Supplementary Figures 1a (Supplementary Figure 3a in the new version) and 4a (Supplementary Figures 9a in the new version), only the top five marker genes for each cluster are displayed, which may not include *Met*. However, as shown in Supplementary Figure 3c (Supplementary Figures 9d in the new version), *Met* transcripts are enriched in epithelial cells compared to other lung cell types.

In addition, we are providing here a more detailed analysis of *Met* transcripts in different cell types of the lung, which also confirms that expression is highest in AT2 cells: we separately examined *Met* expression and found that *Met* is enriched in AT2 cells compared to other lung cell types (see new data below).

Reviewer Fig. 15. Additional data regarding the expression of *Met* and lung in lung and brain cortex, respectively. Analysis of scRNA-seq data showing prominent *Met* expression in AT2 cells of the lung (left) and of *Nodal* in brain pericytes (right).

As noted in Figure 1g, *Nodal* is not expressed by lung pericytes, explaining its absence in Suppl. Fig. 1a and 4a, which are based on our lung scRNA-seq data. However, analysis of P12 brain scRNA-seq data shows that *Nodal* expression is enriched in pericytes compared to

other brain cell types, as indicated by the violin plot provided above.

11. In Sup. , although Glut1 staining is performed, the analysis in Sup. Fig. 3b is “% of ICAM2-stained surface area”.

We thank the reviewer for this comment and apologize for any confusion. GLUT1 is a well-validated marker for brain endothelial cells. In our study, the analysis of both GLUT1 and ICAM2 staining produced consistent results. Throughout the manuscript, ICAM2 staining was primarily used to quantify vascular area and we have therefore now included ICAM2 immunostaining in Suppl. Fig. 6a of the revised manuscript (was Suppl. Figure 3a in the original submission).

12. The manuscript contains quite a few typographical errors. For example, “Nodel” should be “Nodal” on page 16, line 392. On page 9, line 226, it should be “Sftpb” instead of “Sftbp”.

We thank the reviewer for pointing out these typographical errors. All instances have been carefully corrected in the revised manuscript. We have also thoroughly checked the revised manuscript for any additional typographical errors.

13 On page 14, line 334, the extra “that” should be removed.

Thank you. This has been corrected.

Reviewer #3 (Remarks to the Author)

In the entitled manuscript Pericytes are organ-specific regulators of tissue morphogenesis by Rasouli et al., the authors characterize the role of three pericyte-derived factors, Hgf, BDNF, and Nodal, that are exclusively enriched in lung pericytes (Hgf & BDNF) or brain pericytes (Nodal). They provide a thorough analysis into how the Hgf and BDNF are crucial for lung development, most notably alveolar type 2 cells and the endothelium. Whereas pericyte expression of Nodal is important for blood vessel growth in the brain and loss of such results in reactive astrocytes and microglia independent of blood-brain barrier leakage. They demonstrate that pericyte-derived Hgf and BDNF did not alter brain vascular development where nodal expression from pericytes in the lung where not required for lung development. These were extremely thorough studies for each organ and pericyte-derived factor and therefore I have mostly minor comments that could help strengthen these studies.

Thank you very much for the feedback and your assessment of our study.

- The discussion is limited on the knock-out of Nodal, Hgf, and BDNF from Pdgfrb-expressing meninges and perivascular fibroblasts. Is it possible that loss of these factors could affect meningeal development/homeostasis and/or large diameter vessels where perivascular fibroblasts reside?

Thank you for these questions. Based on brain scRNA-seq data, *Nodal* expression is restricted to pericytes and not vessel-associated fibroblasts. Likewise, *Bdnf* and *Hgf* are not expressed by brain *Pdgfrb*⁺ fibroblasts or pericytes. We have not investigated potential changes in the meninges, which would presumably affect maturation of this layer rather than development, which largely occurs in the embryo and therefore before gene inactivation in

Reviewer Fig. 16. Additional data regarding marker expression in *Nodal* mutant and control pericytes and fibroblasts. Heatmaps showing the expression that the expression of *Nodal* is confined to brain pericytes and absent from brain fibroblasts. Note absence of *Hgf* and *Bdnf* expression in both populations.

- The same timeframe for tamoxifen exposure and therefore knocking out these genes was applied, however how does this relate to the stage in which the lung vs. the brain are in development or homeostasis? I think discussion of this needs to be expanded to identify the necessity of these pericyte-factors for either development or homeostasis. The authors sometimes state that these factors are required for homeostasis (perhaps to dampen inflammatory responses), however the use of proliferation and differentiation markers would suggest they are necessary for developmental programs.

Agree. The main aim of our study is indeed to demonstrate that pericytes acquire organ-specific features and regulate morphogenetic processes during postnatal development, whereas homeostasis in healthy adults or alterations in disease models would clearly a separate study involving appropriate mutant models. The activation of astrocytes and microglia in *Nodal* mutants indeed argues that a follow-up study might be promising. In this setting, it would be desirable to induce gene inactivation in adults and thereby after the completion of developmental processes.

As correctly pointed out by the reviewer, we had initially focused on two different time periods for brain and lung because preliminary data and published studies had indicated that these are critical developmental windows in the two organs. However, the inclusion of new data from additional stages for brain (P8 and P21) has, in our view, largely eliminated the differences in the characterization of brain and lung phenotypes.

- For BDNF they went into the downstream receptor, *Ntrk2*, and knocking that out in the endothelium to investigate a similar phenotype. While this is thorough work and maybe not necessary for all factors manipulated in this study, the discussion on the expression patterns/cell type specificity for their respective receptors felt thin.

We thank the reviewer for this valuable comment. We have now strengthened the data regarding receptor expression, which, for example, clearly demonstrates that c-Met, the HGF receptor, is predominately expressed in AT2 cells, identified by LAMP3 or pro-surfactant

protein C (proSP-C) staining (Fig. 3a,d and Supplementary Fig. 8b, c). This observation is further validated by our lung scRNA-seq data (Supplementary Fig. 2a, b, Supplementary Fig. 8a, Supplementary Fig. 9e–g).

Given that previous studies have already established that conditional inactivation of c-Met in AT2 cells impairs alveologenesis (refs. 42 and 43; cited previously in the initial submission)^{6,33}, we have not reinvestigated this function of the HGF receptor. In contrast, published evidence for the function of the Ntrk2 in endothelial cells was limited, so we decided to conduct loss-of-function experiments.

Regarding the potential receptors for pericyte-derived Nodal, we previously examined the expression of candidate receptors for the TGF β family ligand Nodal, as shown in the initial submission (Supplementary Fig. 6d–e; now updated as Supplementary Fig. 11e–f). Furthermore, leveraging our new analysis of brain scRNA-seq data (Supplementary Fig. 1a, b) and CellPhoneDB (CPDB) interaction analysis, we now provide evidence for significant ligand–receptor interactions between pericyte-derived Nodal and the corresponding receptors, which are expressed by multiple cell types including astrocytes and microglia (Supplementary Fig. 1b).

- RAGE was used in figure 1 but was not explained until Figure 3k.

We apologize for this oversight and have modified the text in the revised manuscript accordingly.

- Somewhat related, Figure 3b the authors state that loss of lung pericyte-Hgf impairs alveologenesis, however this is showing staining of RAGE, and not sure how this staining demonstrates alveologenesis besides the increase in fluorescent intensity in the HGF-iPCKO example. Especially since the authors then discuss that RAGE is an indication of inflammation for Figure 3k. But then overall RAGE is reduced in mutants?

We thank the reviewer for this thoughtful comment and the opportunity to clarify this point. In Figure 3, alveolar epithelial cell reduction was assessed using multiple markers, including LAMP3, NKX2.1, c-MET, AQP5, and RAGE, with additional validation coming from our scRNA-seq data. To further strengthen our conclusions, we have now performed quantitative analyses of airspace volume and lung volume in *Hgf* mutants compared to controls. These new results confirm that alveologenesis is significantly impaired in the absence of pericyte-derived HGF, which further strengthens our conclusion. Thus, our analysis of lung defects involves several approaches and does not rely on a single marker.

Regarding RAGE, we agree that it has dual roles: it is widely used as a marker of alveolar type I epithelial cells, but its upregulation in focal regions can also indicate localized inflammatory responses³⁴. In our study, we observed an overall reduction of RAGE+ AT1 epithelial area in *Hgf* mutants, consistent with impaired alveolar development. However, a subset of “hotspot” regions exhibits elevated RAGE staining, which likely reflects localized inflammation rather than normal alveolar differentiation.

- The organization of the manuscript threw me off a little bit, starting with lung pericyte-derived BDNF and then jumping to brain pericyte-derived Hgf and BDNF for the supplement in the narrative. Maybe there is another way to organize this?

Thank you for your comment. We appreciate that there might be other ways to present the data in the manuscript, but we have decided to stick to the original presentation to minimize the transitions between the different organs and factors.

- For the electron microscopy studies in Figure 4i, the endothelial protrusions are interesting, however their function/importance is not known, perhaps related to sensing or controlling of cerebral blood flow. Additionally, it is possible that there is a zonation specific feature for having more or less endothelial protrusions ie. Arterioles, arteriole-capillary transition, capillary, post-capillary venule, and venules. It would be more convincing if this data is quantified and the authors demonstrate that they were exclusively studying capillary vessels. Same goes for the perivascular space, some types of vessels have more or less perivascular spaces. So classification of the vessel type along with quantification of these features would greatly strengthen these data.

We thank the reviewer for this insightful comment. Unfortunately, due to technical limitations of our electron microscopy set-up, which is not an ion-milling system, full three-dimensional quantification of vessel features is not feasible. Nevertheless, we have carefully analyzed capillaries with diameters of 5–10 μm , including a total of 18 wild-type and 28 mutant capillaries. Within these capillaries, we observed that endothelial protrusions are more pronounced in *Nodal* mutants, whereas the perivascular space is more variable and not consistently expanded in all mutants.

Additionally, it is unclear what the red arrows are suppose to indicate, they are not pointing at tight junctions like the figure legend indicates. The white arrow in the right bottom image looks to be pointing at an astrocytic endfoot not a pericyte cell body or process. What are the green arrows pointing to?

We apologize for any confusion that this may have caused. We have now carefully corrected the annotations in Fig. 4J and rearranged the arrows to ensure they accurately indicate the structures described in the figure legend. The revised figure and legend have been updated accordingly: Endothelial cell (EC), pericyte (PC), astrocyte (AC), basement membrane (red arrowheads), extracellular space (*), endothelial protrusions (black arrows).

Reviewer Fig. 17/Fig. 4J. Electron micrographs of *Nodal* mutant and control brain cortex.

- The authors looked at blood-brain barrier leakage but indicated this likely not a junctional problem, based only on EM data? So is it a transcytosis problem? Is there some way the authors can look at this in their electron micrographs or via IHC to explain how the BBB leakage is occurring?

Thank you for this comment. Based on our electron microscopy analyses, we observed an accumulation of vesicles within endothelial cells (indicated by arrows in the micrographs) in the P12 mutant brain, suggesting that the increased BBB leakage is associated with enhanced transcytosis. Nevertheless, we appreciate that this evidence is insufficient for definitive conclusions and we therefore remain cautious with our statements in the manuscript.

Reviewer Fig. 18. Electron micrographs of *Nodal* mutant and control brain cortex. Accumulation of vesicles in endothelial cells (mainly non-clathrin coated, black arrows), indicating an increase in caveolar transcytosis. Clathrin-coated vesicle is marked by red arrow. Endothelial cell (ec), pericyte (pc), astrocyte (AC), basement membrane (white arrowhead).

- Authors state that microglia are highly ramified in the cortex of *Nodal*-iPCKO mice, however there is no quantification of this.

We now provide quantitation for microglial morphology (branch points and microglia average volume) in **Figure 6 e, f**. These results show strongly increased morphological complexity in *Nodal*^{iPCKO} microglia compared to controls. However, we also note that *Nodal*^{iPCKO} microglia show increased AIF1 immunostaining and, to minimize the bias caused by different AIF1 levels in our analysis, we have used surface-rendering (with Imaris software) for the quantitation of branch points and cell volume.

Reviewer Fig. 19/ Fig. 6e, f. Analysis of AIF1 expression and microglial morphology.

Reviewer #4 (Remarks to the Author):

R1: Review of “Pericytes are organ-specific regulators of tissue morphogenesis” by Rasouli et al.

The authors are investigating the vascular network, particularly molecular signals by pericytes controlling the behavior of other cell types during organ growth and patterning. Namely, the authors were choosing lung and brain as representative organs and explored the function of several pericyte-derived, potentially angiocrine-acting factors, namely brain-derived neurotrophic factor (BDNF), hepatocyte growth factor (HGF) and Nodal, a TGFβ family protein. While the authors present some intriguing first results, these seem to be rather preliminary.

When the authors follow up on an interesting initial finding, such as the finding that pericyte-specific nodal deletion reduces brain vascularization, the subsequent analysis of astrocyte and microglia cell behavior stays rather superficial. An in depth analysis of pericyte-derived nodal effects on vascular network, developmental effects vis a activation of cells is missing. While some of the overall findings are interesting, this study rather describes initial findings that require an in depth analysis to be able to draw conclusions.

We are very grateful for this assessment and feedback. It is appreciated that the analysis of the different phenotypes could go deeper, but the main task of our study is to establish that pericytes acquire organ-specific features and control morphogenetic processes through the release of growth factors.

Major points:

1. The author used several conditional knockout mouse lines to investigate the function of pericyte-derived BDNF, HGF, and Nodal on postnatal brain and lung development. However, some characterizations of these mouse lines are missing. To confirm the specificity and efficiency of tamoxifen-induced depletion, the authors should investigate the reduction of interested factors (namely, BDNF, HGF and Nodal) in the pericytes of these mouse lines on the protein level (e.g., through IHC stainings or WB on FACS isolated cells).

We thank the reviewer for this suggestion. Unfortunately, as is often the case for secreted growth factors, we found that available antibodies do not produce specific and reliable immunostaining. However, our single-cell RNA-seq data demonstrate effective reduction of *Nodal* transcript in brain pericytes (**Supplementary Fig. 11c, d**) but also of *Bdnf* (**Supplementary Fig. 3a**) and *Hgf* (**Supplementary Fig. 9a**) in lung pericytes. These results confirm that the *Pdgfrb-CreERT2*-mediated gene inactivation is successful and specific (see below).

Reviewer Fig. 20. Validation of successful gene inactivation in our experiments. (a-c) Violin plots depicting the expression of *Nodal*, *Bdnf*, and *Hgf* in pericyte populations. *Nodal* expression is effectively depleted in brain pericytes of *Nodal*^{iPCKO} mutants (a), whereas *Hgf* (b) and *Bdnf* (c) transcripts are strongly reduced in *Hgf*^{iPCKO} (Supplementary Fig. 9c) and *Bdnf*^{iPCKO} (Supplementary Fig. 3c) lung pericytes, respectively.

Also, the authors should utilize their single cell RNA sequencing data to reveal the alteration in cellular compositions (by UMAP or proportional bar graph) after BDNF, HGF and Nodal depletion.

Thank you for this suggestion. In response, we have reanalyzed our scRNA seq data to evaluate changes in cell composition after loss of pericyte-derived BDNF, HGF, or Nodal, respectively. We are providing proportional bar graphs here (see below), which lack error bars and p-values due to an insufficient number of independent replicates, and these analyses will be also available to readers through the transparent peer review (TPR) file.

Nodal^{iPCKO} vs. control brain

Hgf^{iPCKO} vs. control lung

Bdnf^{iPCKO} vs. control lung

Reviewer Fig. 21. Analysis of cell composition in the indicated mutant models and organs relative to control based on scRNA-seq data.

2. In Figure 1, the author showed some differential regulated genes (e.g., *Nodal*, *Hgf*, *Bdnf*, *Amgpt1*,...etc.) in the lung and brain pericytes. However, the source of these single cell RNA sequencing data is not clearly identified. Are these data from their previous experiments or from a public source? In case of comparing their own data to the public sources, the author should note if there is age, isolation, and sequencing differences, as well as provide proper data integration control and analysis.

We agree that this is a very important point and provide the following information in the Methods and main text:

“Comparing a public scRNA-seq resource for mouse lung provided by the Thébaud laboratory (ref. 29; cited previously in the initial submission)² with our own scRNA-seq data.”

“Comparison to lung P14 pericytes

We downloaded raw sequencing data and cell annotations by Hurskainen et al. (ref. 29; cited previously in the initial submission)². (<<https://doi.org/10.1038/s41467-021-21865-2>>) from GEO (GSE151974) and processed them analogously to the datasets generated in this study, except for using STAR settings appropriate for 10X Chromium v3 (“—soloType CB_UMI_Simple –soloCBlen 16 –soloUMIstart 17 –soloUMIlen 12 –soloCBwhitelist 10xv3_whitelist.txt”).”

Moreover, the Methods section provides a fairly detailed description of the sample numbers and tissue processing, the scRNA-seq procedure, and the bioinformatic analysis of the resulting data.

3. Figure 4: Please confirm effects of Nodal depletion on Smad phosphorylation on mouse tissue.

Thank you for this suggestion. We are providing this result below:

Reviewer Fig. 22. Reduced p-SMAD2 immunostaining in P12 *Nodal*^{iPCKO} brain. Maximum intensity projections of P12 *Nodal*^{iPCKO} and littermate control brain sections stained for GFAP (magenta), pSMAD2 (grey) and DAPI (blue). Graph shows decreased expression of pSMAD2 in *Nodal*^{iPCKO} brains. Data represents mean ± s.e.m. (n=4); P-value, unpaired two tailed student t-test.

4. Figure 5 is revealing increased GFAP level in the cortex of Nodal depleted mice in pericytes. At this early timepoint of cortical development, astrogenesis could be influenced rather than astrocyte activation. Further marker of astrocytogenesis and/or astrocyte activation should be used to clarify the biological process.

To address this concern, we performed EdU staining to evaluate astrocyte proliferation. The resulting data show that the increased GFAP levels in the *Nodal* mutant cortex are not due to elevated astrocyte proliferation, indicating that the observed changes likely reflect astrocyte activation rather than enhanced astrogenesis.

Reviewer Fig. 23. Maximum intensity projections of P12 *Nodal*^{iPCKO} and littermate control brain sections stained for GFAP (magenta) and EdU (grey).

Furthermore, it should be clarified if the increased GFAP level are the consequence of pericyte-derived nodal depletion or of increased BBB opening. The cell culture experiments do not show any novelty, as it is known that TGFbeta family members activate the TGFbeta signaling pathway in astrocytes.

Thank you for this comment. We have now included new analyses of the P8 brain (**Supplementary Fig. 13**), in which no vascular leakage was detected using the same approaches. Despite the absence of barrier disruption, we still observed increased GFAP+ astrocytes in the cortex, indicating that reactive astrogliosis is initiated prior to the emergence of vascular leakage.

Moreover, *in vitro* experiments confirm that cultured mouse primary astrocytes are responsive to recombinant rhNodal protein. Western blot analysis of cell lysates from rhNodal-treated astrocytes reveals a significantly increased phosphorylation of the downstream signal transducer SMAD2 (p-SMAD2), which is abolished by SB431542 (**Fig. 5h**). These data support that pericyte-derived Nodal might directly regulate astrocyte behavior in the postnatal brain. Cell culture experiments also show that SB431542 administration leads to an increase in GFAP+ astrocytes (**Supplementary Fig. 14f, g**), which is consistent with the known role of TGFβ as a negative or, depending on context, positive regulator of astrocyte reactivity (ref 48 in the initial submitted version)³⁵. Supporting our findings in the scRNA-seq analysis, rhNodal stimulation reduces the nuclear immunostaining of FOS and FOSB (Supplementary Fig. 15c-f). Conversely, SB431542 treatment of

astrocytes increases nuclear localization of SOX9, FOS and FOSB (**Supplementary Fig. 14f, h** and **Supplementary Fig. 15c-f**).

Taken together, these findings support that the loss of pericyte-derived Nodal can induce reactive astrogliosis directly.

5. Figure 6 is revealing microglia activation in Nodal depleted mice in pericytes. As the Nodal depletion is leading to blood leakage into the brain, it is not surprising that microglia are activated. Again, it should be clarified if the microglia activation is due to pericyte-derived nodal depletion or of increased BBB opening.

We thank the reviewer for this important suggestion. To directly address vascular permeability, we assessed barrier integrity in the P12 brain cortex by confocal imaging after dextran injection, together with TER-119 labeling of red blood cells and endogenous mouse IgG staining to detect serum extravasation (**Supplementary Fig. 12** and **Supplementary Fig. 17**). In the P12 cortex, AIF1+microglia are observed in regions with dextran extravasation (**Supplementary Fig. 17a**), which is consistent with the possibility that microglial activation may occur secondary to vascular defects. Importantly, AIF1+ microglia are also present in regions that lack detectable dextran/IgG extravasation (**Supplementary Fig. 17b**), indicating that microglial activation is not restricted to sites of vascular leakage.

To further test whether barrier disruption is required for microglial activation, we performed additional analyses in the P8 brain (**Supplementary Fig. 18** and **Reviewer Fig. 13**). In P8 cortex we detected no vascular leakage using the same approaches, yet microglial activation was still evident in Nodal mutant animals. Together, these data indicate that while vascular leakage can be associated with microglial activation in P12 animals, microglial activation is initiated prior to detectable barrier disruption.

Reviewer Fig. 24/Supplementary Figure 18. Microglia activation emerges early in the *Nodal*^{iPCKO} cortex. (a, b) Confocal images of brain cortex stained for ICAM2 (green) and mouse immunoglobulin G (IgG, white) showing lack of extravasated IgG in P8 *Nodal*^{iPCKO} and control littermates (3/3 brains) (a, b). Imaris surface-rendered image of AIF1+ microglia (yellow) showing that microglia volume is increased in P8 *Nodal*^{iPCKO} mutants (b).

(c) Confocal images of GLUT1+ (red) capillaries in brain cortex. Red blood cells (Ter119, green) are confined to vessels in P8 *Nodal*^{IPCKO} and control littermates (3/3 brains), but increased microglia volume (white) can also be observed in P8 *Nodal*^{IPCKO} mutants.

Quantitation of AIF1 expression and average microglial volume in the P8 *Nodal*^{IPCKO} and littermate control brain cortex are provided in Supplementary Figure 18d, e.

6. Supplementary Fig. 8d. Why are no control cells GFAP positive? Is there a problem with the cell culture?

Thank you for altering us to this issue. After reviewing our data, we have replaced this image with a more representative example that accurately reflects GFAP expression in the control condition (now shown in **Suppl. Fig. 14f**). However, as is also evident in the quantitation of this data (**Suppl. Fig. 14g**), GFAP expression is much lower under control condition compared to SB431542 treatment.

References

1. Kato, K. *et al.* Pulmonary pericytes regulate lung morphogenesis. *Nat Commun* **9**, 2448 (2018).
2. Hurskainen, M. *et al.* Single cell transcriptomic analysis of murine lung development on hyperoxia-induced damage. *Nat Commun* **12**, 1565 (2021).
3. Lassus, P., Heikkila, P., Andersson, L.C., von Boguslawski, K. & Andersson, S. Lower concentration of pulmonary hepatocyte growth factor is associated with more severe lung disease in preterm infants. *J Pediatr* **143**, 199-202 (2003).
4. Li, J. *et al.* Exome Sequencing of Neonatal Blood Spots and the Identification of Genes Implicated in Bronchopulmonary Dysplasia. *Am J Respir Crit Care Med* **192**, 589-596 (2015).
5. Ohki, Y. *et al.* Hepatocyte growth factor treatment improves alveolarization in a newborn murine model of bronchopulmonary dysplasia. *Neonatology* **95**, 332-338 (2009).
6. Calvi, C. *et al.* Hepatocyte growth factor, a determinant of airspace homeostasis in the murine lung. *PLoS Genet* **9**, e1003228 (2013).
7. Kirkham, P.A. & Barnes, P.J. Oxidative stress in COPD. *Chest* **144**, 266-273 (2013).
8. Kanazawa, H., Tochino, Y., Asai, K. & Hirata, K. Simultaneous assessment of hepatocyte growth factor and vascular endothelial growth factor in epithelial lining fluid from patients with COPD. *Chest* **146**, 1159-1165 (2014).
9. Helan, M. *et al.* BDNF secretion by human pulmonary artery endothelial cells in response to hypoxia. *J Mol Cell Cardiol* **68**, 89-97 (2014).
10. Chen, B. *et al.* Autocrine activity of BDNF induced by the STAT3 signaling pathway causes prolonged TrkB activation and promotes human non-small-cell lung cancer proliferation. *Sci Rep* **6**, 30404 (2016).
11. Paris, A.J. *et al.* STAT3-BDNF-TrkB signalling promotes alveolar epithelial regeneration after lung injury. *Nat Cell Biol* **22**, 1197-1210 (2020).
12. El Kholtei, J., Codina-Tobias, M. & Schier, A.F. Nodal Signaling: A Paradigm for TGFbeta Signaling in Embryonic Development. *Annu Rev Cell Dev Biol* **41**, 45-88 (2025).
13. Jewer, M. *et al.* Translational control of breast cancer plasticity. *Nat Commun* **11**, 2498 (2020).

14. Lonardo, E. *et al.* Nodal/Activin signaling drives self-renewal and tumorigenicity of pancreatic cancer stem cells and provides a target for combined drug therapy. *Cell Stem Cell* **9**, 433-446 (2011).
15. De Silva, T. *et al.* Nodal promotes glioblastoma cell growth. *Front Endocrinol (Lausanne)* **3**, 59 (2012).
16. Hueng, D.Y. *et al.* Inhibition of Nodal suppresses angiogenesis and growth of human gliomas. *J Neurooncol* **104**, 21-31 (2011).
17. Topczewska, J.M. *et al.* Embryonic and tumorigenic pathways converge via Nodal signaling: role in melanoma aggressiveness. *Nat Med* **12**, 925-932 (2006).
18. Cui, Y. *et al.* Nodal mitigates cerebral ischemia-reperfusion injury via inhibiting oxidative stress and inflammation. *Eur Rev Med Pharmacol Sci* **23**, 5923-5933 (2019).
19. Heuchel, R. *et al.* Platelet-derived growth factor beta receptor regulates interstitial fluid homeostasis through phosphatidylinositol-3' kinase signaling. *Proc Natl Acad Sci U S A* **96**, 11410-11415 (1999).
20. Tallquist, M.D., French, W.J. & Soriano, P. Additive effects of PDGF receptor beta signaling pathways in vascular smooth muscle cell development. *PLoS Biol* **1**, E52 (2003).
21. Ayata, P. *et al.* Epigenetic regulation of brain region-specific microglia clearance activity. *Nat Neurosci* **21**, 1049-1060 (2018).
22. Ahmed, Z. *et al.* Actin-binding proteins coronin-1a and IBA-1 are effective microglial markers for immunohistochemistry. *J Histochem Cytochem* **55**, 687-700 (2007).
23. Kanazawa, H., Ohsawa, K., Sasaki, Y., Kohsaka, S. & Imai, Y. Macrophage/microglia-specific protein Iba1 enhances membrane ruffling and Rac activation via phospholipase C-gamma -dependent pathway. *J Biol Chem* **277**, 20026-20032 (2002).
24. Postler, E., Rimmer, A., Beschoner, R., Schluesener, H.J. & Meyermann, R. Allograft-inflammatory-factor-1 is upregulated in microglial cells in human cerebral infarctions. *J Neuroimmunol* **104**, 85-91 (2000).
25. Norden, D.M., Trojanowski, P.J., Villanueva, E., Navarro, E. & Godbout, J.P. Sequential activation of microglia and astrocyte cytokine expression precedes increased Iba-1 or GFAP immunoreactivity following systemic immune challenge. *Glia* **64**, 300-316 (2016).
26. Gottfried, E. *et al.* Expression of CD68 in non-myeloid cell types. *Scand J Immunol* **67**, 453-463 (2008).
27. Zoller, T. *et al.* Silencing of TGFbeta signalling in microglia results in impaired homeostasis. *Nat Commun* **9**, 4011 (2018).
28. Abutbul, S. *et al.* TGF-beta signaling through SMAD2/3 induces the quiescent microglial phenotype within the CNS environment. *Glia* **60**, 1160-1171 (2012).
29. Kato, T., Oka, K., Nakamura, T. & Ito, A. Decreased expression of Met during differentiation in rat lung. *Eur J Histochem* **60**, 2575 (2016).
30. Seedorf, G. *et al.* Hepatocyte growth factor as a downstream mediator of vascular endothelial growth factor-dependent preservation of growth in the developing lung. *Am J Physiol Lung Cell Mol Physiol* **310**, L1098-1110 (2016).
31. Shiratori, M. *et al.* Hepatocyte growth factor stimulates DNA synthesis in alveolar epithelial type II cells in vitro. *Am J Respir Cell Mol Biol* **12**, 171-180 (1995).
32. Sakamaki, Y. *et al.* Hepatocyte growth factor stimulates proliferation of respiratory epithelial cells during postpneumonectomy compensatory lung growth in mice. *Am J Respir Cell Mol Biol* **26**, 525-533 (2002).

33. Yamamoto, H. *et al.* Epithelial-vascular cross talk mediated by VEGF-A and HGF signaling directs primary septae formation during distal lung morphogenesis. *Dev Biol* **308**, 44-53 (2007).
34. Schmidt, A.M., Yan, S.D., Yan, S.F. & Stern, D.M. The multiligand receptor RAGE as a progression factor amplifying immune and inflammatory responses. *J Clin Invest* **108**, 949-955 (2001).
35. Luo, J. TGF-beta as a Key Modulator of Astrocyte Reactivity: Disease Relevance and Therapeutic Implications. *Biomedicines* **10** (2022).

Before we provide a detailed point-by-point response to all comments received on the first revision of our manuscript, we would like to thank the reviewers for their time and comments.

Reviewer #1 (Remarks to the Author):

The authors have addressed my previous comments very thoroughly. I have one minor suggestion: a recent study (PMID: 39806101) indicates that HIGD1B may serve as a pericyte marker. Could the authors conduct an analysis of this marker to determine whether it would distinguish pericytes from smooth muscle cells in their dataset?

We thank the reviewer for the positive feedback.

Regarding *Higd1b* expression, our analysis of our own but also previously published scRNA-seq datasets indicates that expression of this gene is largely restricted to pericytes in the lung, consistent with what has been reported by Klouda et al. (PMID: 39806101). In contrast, transcripts are found both in pericytes and in vascular smooth muscle cells from brain or heart (see below) but also in scRNA-seq data from heart (PMID: 36283392).

Reviewer Fig. 1: Analysis of *Higd1b* expression in different scRNA-seq dataset indicates that transcripts are largely confined to lung pericytes but expression by both pericytes and smooth muscle cells in the brain.

Reviewer #2 (Remarks to the Author):

The revised manuscript investigates the organ-specific roles of pericyte-derived factors in postnatal tissue morphogenesis, focusing on BDNF, HGF, and Nodal signaling in the lung and brain. The quality of the revised manuscript is very much improved. Still, several critical questions remain unanswered or insufficiently addressed. As a result, the authors'

conclusions do not appear to be solidly supported by the presented data. In order to further develop their work, the authors need to address the following points:

1. The previous comments of this reviewer about *Pdgfrb-CreERT2* lines have not been sufficiently addressed. The authors state that *Pdgfrb-CreERT2* does not induce similar lung or brain defects across different mutant backgrounds. However, this does not resolve the key concern raised in the original review that the *Pdgfrb-CreERT2* allele itself causes baseline developmental abnormalities as stated specifically for this line by the Jackson Lab (“Important Note: As of August 2017, it has been the experience at The Jackson Laboratory Repository that when breeding hemizygous mice to wildtype siblings or C57BL/6J inbred mice, the resulting of their findings, the authors should validate key findings using an alternative Cre line that does not exhibit baseline developmental abnormalities such as PDGFR β litters have less hemizygous offspring than expected, which is 20% instead of the expected 50%. Hemizygous mice are smaller than non-carrier siblings.”). Therefore, to ensure the validity -P2A-CreERT2 (The Jackson Laboratory, Strain #030201) and explicitly acknowledge the known developmental phenotype of *Pdgfrb-CreERT2* mice and its potential impact on data interpretation in the manuscript.

We politely but firmly disagree with the suggestion made by the reviewer that we “should validate key findings using an alternative Cre line that does not exhibit baseline developmental abnormalities”.

First of all, we would like to clarify that our *Pdgfrb-CreERT2* colony at the MPI for Molecular Biomedicine, which was generated by us and used for all the experimental work reported in the current manuscript, does not exhibit the features described by The Jackson Laboratory. We are, of course, aware of what people at The Jackson Laboratory had observed after the import and rederivation of our line, but we attribute these features either to changes in genetic background caused by the rederivation process or to specific properties of the transferred animals.

Second, as already mentioned in our reply to the first round of comments, we would like to emphasize that the absence of alterations in *Pdgfrb-CreERT2*-expressing mutants provides irrefutable evidence that the reported phenotypes are not caused by the sole presence or activity of the *Pdgfrb-CreERT2* allele. Specifically, the presence of *Pdgfrb-CreERT2* does not induce defects in *Nodal* mutant lungs and pulmonary development in these animals is indistinguishable from Cre-negative controls (Supplementary Fig. 10). In the brain, the presence of *Pdgfrb-CreERT2* in *Bdnf* or *Hgf* mutants does not affect glial cell responses or vascular development compared to the respective Cre-negative control littermates (Supplementary Fig. 4, 5, 6 and 7). Taken together, there is no credible evidence indicating that the reported phenotypes are unspecific or indirect.

Finally, it should be noted that the genetic experiments suggested by the reviewer require the import of a new CreERT2 line into our animal facility, crossbreeding with various loxP-flanked alleles and a large series of genetic experiments and phenotypic analyses, which would altogether would take about two years. Given that there is no scientific justification for such a large series of additional genetic experiments apart from non-peer-reviewed

information posted at The Jackson Laboratory website, which is not even relevant for our own *Pdgfrb-CreERT2* colony, the additional genetic experiments would be incompatible with basic ethical principles of animal experimentation and the 3R framework for replacement, reduction and refinement.

2. The authors' response did not adequately address the issue of unjustified overinterpretation of the results. First, the statement in the abstract that "inactivation of the HGF gene impairs alveologenesis due to defective interaction with AT2 epithelial cells" has not been validated. The revised manuscript does not provide experimental evidence supporting an instructive role of AT2 cells in this process. Immunostaining of c-Met expression in AT2 cells only indicates receptor presence but does not demonstrate functional interaction. Although the authors cite published studies showing that AT2 cells specific c-Met deletion impairs alveologenesis (PMID: 17583691), this raises questions about the novelty of the current findings and these studies should also be included in the manuscript. To substantiate the proposed mechanism, direct manipulation of AT2 cells in the present study is still required. Therefore, the claim that HGF loss impairs alveologenesis through defective AT2 interaction should be revised or removed from the abstract unless experimentally validated. Second, the authors state that "the reduced vascularization could be secondary to the defects in alveolar epithelium observed in postnatal Hgf iPCKO lung" (revised line 273). However, simply demonstrating the absence of C-Met in endothelial cells does not address whether the vascular defect is secondary to epithelial abnormalities. To support this claim, experiments elucidating the temporal sequence of epithelial versus vascular defects are still required. Without such data, the conclusion that the vascularization defect is secondary remains unsupported.

Thank you for this comment. First of all, there is a solid body of literature demonstrating that signaling of HGF through its receptor c-Met controls AT2 cell proliferation and thereby lung development and growth (PMID: 22245998, PMID: 7532419, PMID: 15994466). Furthermore, it has been established that the AT2 epithelial cell-specific inactivation of the *Met* gene impairs airspace morphogenesis but also pulmonary vascular development (see below Fig. 1G, H from PMID: 23459311). This result confirms that pulmonary epithelial and vascular growth are coupled, consistent with earlier results (PMID: 17583691).

[FIGURE REDACTED]

Reviewer Fig. 2 (taken from Calvi et al. 2013, PMID: 23459311): **G.** Representative thrombomodulin immunohistochemical staining of the microvascular bed in the lung parenchyma of *Met* deficient mice compared with controls. Inset shows reduced staining in the alveolar epithelial walls. 40× magnification, inset 100×. N=5–7 mice per genotype. **H.** Quantitative immunohistochemistry of thrombomodulin staining of *Met*-deficient mice and controls. **p<0.01.

Furthermore, our previous work has already established that *Pdgfrb-CreERT2*-induced inactivation of the Hippo pathway components *Yap1* and *Taz* reduced *Hgf* expression in pulmonary pericytes, strongly diminished c-Met activation (tyrosine phosphorylation) in P12 whole lung lysates, and compromised the proliferation of c-Met expressing AT2 epithelial cells and alveologenesis (PMID: 29934496 and Fig. 3 below).

[FIGURE REDACTED]

Reviewer Fig. 3 (taken from Kato et al. 2013, PMID: 29934496): **e**, RT-qPCR analysis of *Hgf* and *Met* expression in freshly sorted GFP+, CD31+ or EpCAM+ cells from P7 *Pdgfrb-CreERT2* R26-mT/mG lungs. Data represents mean \pm s.e.m. (n = 4 mice). **f**, High magnification images of P12 lungs stained for SFTPC (green), c-Met (red), and RAGE (blue). Arrows indicate SFTPC+ c-Met+ cells. Scale bar, 15 μ m. **g**, *Hgf* expression in freshly sorted PDGFR β + or CD31+ cells from P7 *Yap1, Wwtr1*iPCKO and littermate control lungs. Data represents mean \pm s.e.m. (n = 5 mice; NS not significant, Welch's t-test or two-tailed unpaired t-test). **h**, Western blot analysis of total and phosphorylated c-Met (pMet) in P12 *Yap1, Wwtr1*iPCKO and control total lung lysates (n = 2 for controls and 4 for mutant mice). Molecular weight marker (kDa) is indicated.

The same study also reported that *Met* transcript and protein expression is largely confined to EpCAM+ epithelial cells and, within this population, to SFTPC+ AT2 cells (see above and PMID: 29934496). These findings were independently confirmed by new scRNA-seq data in the current manuscript (see Suppl. Fig. 9c-h). Likewise, data from Hurskainen et al. 2021 (PMID: 33692365) confirms that *Met* expression is confined to epithelial cells and is very low or undetectable in endothelial cells (see below).

Analysis of published lung scRNA-seq data from PMID: 33692365

Reviewer Fig. 4: Reanalysis of lung scRNA-seq data from Hurskainen et al. 2021 (PMID: 33692365) confirms that *Met* expression is confined to epithelial cells.

Consistent with the low/absent expression of *Met* in lung endothelial cells, EC-specific inactivation of the gene did not impair embryonic development and resulting mutants were also able to reach adulthood (PMID: 27043280). The authors of this study concluded that these results argue for “a dispensable role of c-Met in developmental angiogenesis”,

contrasting its involvement in pathological angiogenesis in the context of cancer (PMID: 27043280, PMID: 20952508, PMID: 21613405).

Given that there is compelling evidence that c-Met controls lung development through its function in epithelial cells and since it has already been shown with a cell type-specific genetic approach that Met inactivation in AT2 epithelial cells impairs alveologenesis as well as pulmonary vascular development (PMID: 23459311), additional genetic experiments are not sufficiently justified. Such experiments can only confirm what is already known, would not provide any new scientific insight, and require the import of a new CreERT2 line and the generation of a new set of mutants. Likewise, there is not enough evidence justifying the inactivation of the *Met* gene in endothelial cells.

As outlined in the context of the *Pdgfrb-CreERT2* earlier, the additional genetic experiments would be incompatible with basic ethical principles of animal experimentation and the 3R framework for replacement, reduction and refinement.

To address the concern raised by the reviewer, we have changed the wording of the abstract and of relevant text passages elsewhere in the manuscript to indicate that we propose a signaling axis involving pericyte-derived HGF and c-Met in AT epithelial cells. This wording makes clear that the conclusion is based on our interpretation of the data, whereas we recognize that our findings might not be fully sufficient to convince the reviewer.

3. The authors argue that co-culture of pericytes and astrocytes is not feasible and therefore rely on rhNodal treatment to study the effect of pericyte-derived Nodal on astrocytes. However, the treatment with recombinant protein does not recapitulate the effects of pericyte-derived Nodal and therefore cannot support the claim that “pericyte-derived Nodal might directly regulate astrocyte behavior” (revised line 368). To directly address this point, treating astrocytes with concentrated pericyte-conditioned medium would solve the culture-compatibility issues and provide more physiologically relevant evidence.

As mentioned in our reply to the first round of comments, we found that cultured brain pericytes do not maintain the expression of *Nodal* seen *in vivo*. To support this statement, we are now providing below unpublished RiboTag sequencing data based on methodology published by Jeong et al. 2017 (PMID: 28959057). This data was generated via *Pdgfrb-CreERT2*-controlled expression of an epitope-tagged Rpl22 protein (PMID: 19666516) enabling the pull-down of ribosome-associated transcripts. We found that the recombined cells in the brain cortex show *Nodal* expression, which was lost in *ex vivo* cell culture (see Reviewer Fig. 5a below). While other publicly available scRNA-seq data (see Reviewer Fig. 5b below) independently confirm *Nodal* expression by murine and human brain pericytes, it is clear that it is not feasible to use supernatants from cultured pericytes for stimulation experiments.

Approaches with cell culture supernatants/conditioned medium would be also compromised by another critical issue. It was previously shown that rat brain pericyte-derived TGF β 1 *in vitro* can alter the barrier properties of immortalized mouse brain capillary endothelial (MBEC4) cells (PMID: 15757636). This indicates that conditioned medium from cultured pericytes might contain other ligands of the TGFβ superfamily, which would easily lead to results that are complex and rather difficult to interpret.

All these issues are avoided by the use of recombinant Nodal protein, which has allowed us to study the direct effects of the TGFβ superfamily ligand on cultured astrocytes and microglial cells in the current study.

**Reviewer Fig. 5**: **a**, RiboTag expression analysis of *Nodal* expression in PDGFRβ+ cells from the brain cortex in vivo and in culture. **b**, Data from the human protein atlas confirms prominent *NODAL* expression in human brain pericytes compared to various other cell populations.

4. There is still no consensus as to the relationship between vascular leakage and astrogliosis. While the authors propose that “vascular leakage might contribute to astrogliosis, ”their newly added data of brains at P8 in supplementary Fig. 13 show that no vascular leakage is detected whereas GFAP+ astrocytes are already increased. This contradicts the proposed conclusion and indicates that astrogliosis can occur independently of barrier defects. Therefore, the authors’ interpretation is not sufficiently supported by the data and requires additional experimental validation.

Thank you for this comment. We agree with the reviewer that the new data establish that astrogliosis in the Nodal mutant brain is initiated before the emergence of vascular leakage. This is fully consistent with our main conclusion that pericyte-derived signals act on other cell populations in the periphery of blood vessels. Thus, the new data generates no contradiction even though we cannot rule out that the vascular leakage emerging at later stages will further enhance astrogliosis.

To address the concern raised by the reviewer, we have revised relevant passages in the text. A version of the manuscript with tracked changes is provided as part of this submission.

Reviewer #3 (Remarks to the Author):

The authors have made substantial improvements to the overall paper. They provided stronger evidence that manipulation of Hgf, BDNF, and Nodal in Pdgfrb-expressing pericytes is likely limited to pericytes in the lung or brain and not other Pdgfrb-expressing cell types. But they acknowledge this cannot be ruled out completely. The added data of ligand-receptor interactions is helpful and clarification of RAGE and the addition of other lung parameters strengthens their conclusions there. Finally the addition of quantifying microglia morphology in the Nodal-PC KOs helps make their case. I still have a few remaining comments that need to be addressed.

1) I appreciate that the authors acknowledge the developmental differences between the brain and lung and the addition of the P8 assessment as well. I still do not feel like they set the stage for the developmental/homeostatic timeframe each organ is in when they administer tamoxifen and knock-out the various genes at P1. A simple description of the developmental phase the lung and brain are in during deletion of these various genes would be helpful. Additionally, it would be extremely helpful to add in the age labels to all the figures, this is done in the supplement nicely but not in the main figures.

Thank you for your feedback, which is most appreciated. We have now added a short paragraph to start of the Results section providing justification the experimental timeframe in the context of lung and brain development. As suggested, age labels have been added to the main figures and we agree that these labels are very useful.

2) Still not sure of the EM images since there is no quantification and lack of similar looking/types of blood vessels here. The authors describe the endothelial protrusions as abnormal, when they have been observed before in the healthy brain (PMID: 26604921). Also, still not sure where they are referencing to the perivascular space in Fig. 4j. I wonder if the authors are mixing up enlargement of the perivascular space with potential thickening of the vascular basement membranes which are two different things and difficult to make out here. And it is unclear how many areas and animals per genotype were looked at for these studies to make these conclusions.

Thank you for this comment. We have noted your concerns regarding the interpretation of the EM data in Fig. 4j and the lack of quantification for the observed findings. We have looked at 3-4 brain samples for each group, but since certain alterations, such as hemorrhaging, are very focal, it is proving difficult to conclusively show that endothelial cell junctions are intact even at sites of leakage. Since it is also unclear how the loss of Nodal in PDGFR β + cells leads to increased vascular leakage and focal bleeding, which is discussed in detail in our reply to your next question, we have decided to remove the data in Fig 4j because it does not add any conclusive or mechanistic insight.

3) The authors still do not provide or discuss a clear mechanism for BBB leakage and vessel rupture. The lack of vessel integrity in the Nodal-PC KOs seem to still ride on the electron micrographs, but junctions appear to be fine. The authors state that the basement membrane

is enlarged (and perivascular spaces?) and there are endothelial protrusions, however how these features relate to vessel integrity is not discussed. The reviewer panel showing transcytosis helps and could explain BBB leakage but not vessel rupture. Further, the activation of microglia and astrocytes in the areas lacking active leakage is interesting but this does not rule out historical bleeds or leakage points that have been cleaned up and thus no presence of Ter119+ RBCs/IgG.

Thank you very much for your comment. We appreciate that we have not resolved how the loss of Nodal leads to increased vascular leakage. This, in part, reflects that the main focus of our study concentrated on the paracrine/angiocrine function of pericytes in lung and brain as model organs. We have therefore chosen experiments, such as the stimulation of cultured astrocytes and microglial cells, that resolve the effect of Nodal on cell types in the proximity of brain capillaries. In contrast, the regulation of vascular integrity and barrier function, while unquestionably very important, was a more peripheral issue that we have admittedly not addressed in detail.

Furthermore, there is already some literature on the regulation of vascular growth and barrier function in the central nervous system by TGF β signaling, which is obviously highly relevant in the context of Nodal. Work by the group of Pat D'Amore has established that adenoviral expression of soluble endoglin (sEng) impaired TGF β signaling (measured by Smad2 phosphorylation), reduced vascular perfusion in the retina and compromised blood-retinal-barrier function (PMID: 19340291). Global inactivation of many TGF β pathway components and EC-specific inactivation of Smad4, the central signal transducer downstream of TGF β and BMP ligands, resulted in severely impaired angiogenesis, disrupted vascular integrity and embryonic or perinatal lethality (PMID: 17724086, PMID: 21397841). Notably, EC-specific Smad4 mutants showed pronounced intracranial hemorrhaging and BBB breakdown, which was accompanied by reduced mural cell coverage (PMID: 21397841). These findings highlight the importance of TGF β function but also the necessity to use inducible and cell type-specific genetic approaches for functional studies in postnatal and adult mice. To this end, work by the group of Jeremy Nathans has shown that EC-specific loss of the TGF β receptors TGFBR1 and/or TGFBR2 impaired retinal vascularization, induced local increases in vascular permeability and triggered the infiltration of immune cells (PMID: 41432545). Pericyte-derived TGF β 1 was also shown to increase barrier properties of immortalized mouse brain capillary endothelial (MBEC4) cells in co-culture with rat brain pericytes *in vitro* (PMID: 15757636).

Taken together, it is clear that TGF β family ligands and their receptors control fundamental aspects of vascular growth and integrity (reviewed in PMID: 19114994). The current study adds Nodal as an additional important player and demonstrates the importance of pericytes as a source of this ligand in the brain. We have now incorporated the additional references and discuss the role of TGF β signaling in the regulation of CNS vascular integrity and BBB function in the revised manuscript. A version of the manuscript with tracked changes is provided as part of this submission.

Reviewer #4 (Remarks to the Author):

The manuscript by Rasouli et al. entitled "Pericytes are organ-specific regulators of tissue morphogenesis" addressed now all our major concerns, particularly the analysis of vascular leakage and astrocyte proliferation/activation in Nodal mutants. The manuscript is now suitable for publication.

Thank you very much for your feedback. We are delighted to learn that all your questions and concerns have been addressed.

Before we provide a detailed point-by-point response to all comments received on the first revision of our manuscript, we would like to thank the reviewers for their time and comments.

Reviewer #1 (Remarks to the Author):

The authors have addressed my 2nd round comments well and I have no more questions. The manuscript is now suitable for publication.

The thank the reviewer for this assessment and the helpful feedback throughout the review process.

Reviewer #2 (Remarks to the Author):

The authors have addressed most concerns and have adequately rephrased several of their conclusions and interpretations. The major concern regarding the *Pdgfrb-CreERT* line was only partially clarified. The authors state that their *Pdgfrb-CreERT* line does not exhibit developmental defects as reported by The Jackson Laboratory. However, no experimental evidence is provided in support of this claim. It is therefore suggested that the authors include comparative data of body weights and fertility-related parameters (e.g., litter size at birth). If no differences are observed, then this would settle the case. This is not a semantic issue. Given that the Jackson lab deposited line is tainted, solid clarification is necessary, first, to corroborate the robustness auf the authors' findings and second, to alert the readers that there may be reproducibility issues when referring to publicly available reagents, such as the Jackson lab strain in this case.

As requested by the reviewer, we are now providing several lines of evidence that the presence of the *Tg(Pdgfrb-cre/ERT2)6096Rha/J* allele (termed *Pdgfrb-CreERT2* in our manuscript and hereafter) alone does not cause major growth retardation.

In Fig. 7c of Eilken et al. 2017 (PMID: 29146905), we have provided comparisons of body weight and outgrowth of the retinal vasculature, which is easily impaired by general growth retardation, for mutant and Cre-negative control pups at P6. No significant differences were obtained for *Pdgfrb-CreERT2*-positive *Flt1* mutants and Cre-negative littermate controls.

[FIGURE REDACTED]

Eilken et al. 2017, Fig. 7c: Quantitation of body weight and radial outgrowth of the retinal vasculature in control, *Flt1*^{iPC/+} and *Flt1*^{iPC} P6 pups. Error bars, s.e.m. p-values, one-way ANOVA. NS, not statistically significant.

In Fig. 3A and B of Yao et al. 2024 (PMID: 38509584), the authors compare the body weight of *Pdgfrb-CreERT2*-generated *Epas1* mutants and control animals, which shows no difference at the onset of the experiment at 8 weeks. The same study also provides a comparison of *Pdgfrb-CreERT2*-generated *Hif1a* mutants (Fig. 2D), again showing no difference in body weight at the onset of the experiment at 8 weeks.

[FIGURE REDACTED]

Yao et al. 2024, Fig. 3A and B: No apparent systemic effects caused by PDGFR β + cell HIF2 α inactivation in diet-induced obesity. **A** Monitoring of weekly body weights in control and *Epas1*-bKO mice during 8 weeks of HFD feeding. n = 6 per group. **B** Body composition (% of fat mass and lean mass) of control and *Epas1*-bKO mice after HFD feeding. n = 6 per group.

In Fig. 6C of Goodwin et al. 2023 (PMID: 37102682), the authors report that there is no weight loss of *Pdgfrb-CreERT2*-induced *Gnaq* mutants in a *Gna11*^{-/-} background relative to control littermates with other genotypes after tamoxifen induction starting at P49. While this figure only provides only relative and not absolute numbers, the authors state that “Tamoxifen-naïve *Pdgfrb-Cre/ERT2*^{+/-};*Gnaq*^{fl/fl};*Gna11*^{-/-} mice were born at the expected frequency”, which “indicates that having the *Pdgfrb-Cre/ERT2*^{+/-};*Gnaq*^{fl/fl};*Gna11*^{-/-} genotype, without administration of tamoxifen, does not cause any gross developmental defects.”

[FIGURE REDACTED]

Goodwin et al. 2023, Fig. 6C:
Weights of *Pdgfrb-Cre/ERT2*^{+/-};*Gnaq*^{fl/fl};*Gna11*^{-/-} mice (red) and littermates of all other genotypes (blue) during 21 days of tamoxifen administration.

In addition, we have measured the body weight of recently born *Pdgfrb-CreERT2* pups in our animal facility at the MPI. A comparison of Cre- and Cre+ from three different genetic backgrounds shows no significant difference in body weight at postnatal day 20 (see below).

[FIGURE REDACTED]

Unpublished data: Body weight of *Pdgfrb-Cre/ERT2*^{+/-} pups (w/o tamoxifen treatment) relative to Cre- littermates at P20.

Finally, we got in touch with staff at The Jackson Laboratory, who were extremely supportive, to clarify the issues raised by the reviewer. According to the Technical Information Scientist handling our request, it turns out that hemizygous and wild-type littermates of the line B6.Cg-Tg(*Pdgfrb-cre/ERT2*)6096Rha/J (stock# 029684) “are the same size”. The corresponding public strain datasheet at www.jax.org has been updated accordingly.

Taken together, we trust that this case is hereby settled, as suggested by the reviewer.

Reviewer #3 (Remarks to the Author):

The authors have fully addressed all of my comments and concerns. I believe the manuscript is in good form for publication.

The thank the reviewer for this assessment and the constructive feedback throughout the review process.